# Memory-Statistics Tradeoff in Continual Learning with Structural Regularization

**Haoran Li** [*]
School of Artificial Intelligence
Shenzhen University
Shenzhen, Guangdong, China
`lihr@szu.edu.cn`

**Jingfeng Wu**
Simons Institute
UC Berkeley
Berkeley, CA 94720, USA
`uuujf@berkeley.edu`

**Vladimir Braverman**
Department of Computer Science
Johns Hopkins University
Baltimore, MD 21218, USA
`vova@cs.jhu.edu`

## Abstract

We study the statistical performance of a continual learning problem with two linear regression tasks in a well-specified random design setting. We consider a structural regularization algorithm that incorporates a generalized $\ell_2$-regularization tailored to the Hessian of the previous task for mitigating catastrophic forgetting. We establish upper and lower bounds on the joint excess risk for this algorithm. Our analysis reveals a fundamental trade-off between memory complexity and statistical efficiency, where memory complexity is measured by the number of vectors needed to define the structural regularization. Specifically, increasing the number of vectors in structural regularization leads to a worse memory complexity but an improved excess risk, and vice versa. Furthermore, our theory suggests that naive continual learning without regularization suffers from catastrophic forgetting, while structural regularization mitigates this issue. Notably, structural regularization achieves comparable performance to joint training with access to both tasks simultaneously. These results highlight the critical role of curvature-aware regularization for continual learning.

## 1 Introduction

Continual learning (CL) is a machine learning setting where multiple distinct tasks are presented to a learning agent sequentially. As new tasks are encountered, it is expected that the agent learns both the old and the new tasks as the number of tasks increases. However, due to limited long-term memory, the CL agent cannot retain all past information. This makes CL significantly more challenging than single-task learning, as it cannot perform joint training on all available samples (Parisi et al., 2019). On the other hand, without using exceedingly large long-term memory, we can view a CL problem as an online multi-task problem and sequentially fit a CL model with the data from each task. Unfortunately, the final model obtained via such an online approach could suffer from *catastrophic forgetting* (McCloskey & Cohen, 1989; Goodfellow et al., 2013), where performance on earlier tasks degrades after adapting to new ones due to loss of knowledge from previous tasks.

Many works in CL focus on developing heuristic algorithms that record information about previous tasks to reduce forgetting in later tasks. One effective category of methods mitigates catastrophic forgetting by applying *structural regularization* (Kirkpatrick et al., 2017; Aljundi et al., 2018; Chaudhry et al., 2018; Kolouri et al., 2020; Li et al., 2021). From a geometric perspective (Chaudhry et al., 2018; Li et al., 2021), structural regularization methods store a PSD importance matrix that estimates

---

[*]Work done while H.L. was at Rice University.

the significance of model parameters for previous tasks. When learning new tasks, a quadratic regularizer based on this importance matrix is applied to prevent significant deviation of important parameters from the previous ones (Kirkpatrick et al., 2017; Aljundi et al., 2018).

While various effective importance matrices have been proposed, practitioners have observed that the full importance matrix can require prohibitively large memory, scaling as $\mathcal{O}\left(d^2\right)$ for a neural network with $d$ parameters. To address this, several approximation strategies have been applied to reduce the memory cost, including diagonal approximations (Kirkpatrick et al., 2017; Aljundi et al., 2018), K-FAC (Ritter et al., 2018), and sketching (Li et al., 2023). Empirical results suggest that more accurate approximations, which require higher memory costs, often lead to better CL performance (Ritter et al., 2018; Li et al., 2021).

In contrast to these practical advancements, the theoretical understanding of CL algorithms remains in its early stages. Several theoretical works have analyzed CL and particularly regularization-based CL methods on linear regression (e.g., Evron et al. (2022); Li et al. (2023); Zhao et al. (2024)); however, these studies focus on specific regularizers with fixed memory costs, and none explicitly link memory usage to the statistical performance of the CL algorithm. Additionally, existing works often impose strong assumptions on input data, limiting their forgetting analysis to optimization settings or scenarios with fixed input data or transforms.

**Contributions.** In this paper, we theoretically study the memory-statistics trade-off in continual learning algorithms with structural regularization within the linear regression setting. Specifically, we consider two-task linear regression under covariate shift (see Definitions 1 and 2) in random design. We study the generalized $\ell_2$-regularized CL algorithm (GRCL), where a user-defined quadratic regularizer is applied during the second task, controlling the extent to which the model deviates from the learned first task (see (3)). Our contributions are as follows:

- We provide sharp risk bounds for the GRCL algorithm in the one-hot random design setting. These bounds reveal a provable trade-off between CL performance and memory cost, governed by the regularization matrix: a well-designed regularizer enhances CL performance at the cost of more memory usage.

- We show that without regularization, catastrophic forgetting occurs when there is a significant difference in a small subset of dominant features in the one-hot setting. Conversely, by selecting an appropriate regularizer with higher memory usage, the GRCL algorithm can prevent catastrophic forgetting and achieve the error rate of joint learning.

- We extend the risk bounds without regularization to the Gaussian distribution setting as a technical advancement. We show that, in addition to differences in dominant features, catastrophic forgetting can also occur in the Gaussian design when there exists a slight difference in the tail features.

The paper is organized as follows. Section 2 covers the most related works to our paper; the others are deferred to Appendix A. In Section 3, we set up the theoretical setting of the 2-task CL problem. We present our main results and messages under this setting with supporting numerical experiments in Section 4; additional details of numerical and practical experiments are deferred to Appendix E. We extend our main results and messages into broader settings including Gaussian design, multi-task CL and general neural networks in the NTK regime in Section 5. Finally, we conclude our work in Section 6. All proofs are deferred to Appendices B, C, and D.

## 2 RELATED WORKS

Over the past decade, a long list of practical CL methods has been proposed to address the catastrophic forgetting problem of neural networks. These CL methods can be roughly divided into four categories, utilizing regularization, replay, architecture expansion and projection respectively. One can refer to Parisi et al. (2019); Wang et al. (2024) or Appendix A for a comprehensive overview.

**Memory-performance relationship beyond regularization.** The relationship between memory and CL performance has also been observed in other CL paradigms. In replay-based methods, the performance of ER (Chaudhry et al., 2019) rises with the size of the episodic memory. Similarly, in

projection-based approaches like GPM (Saha et al., 2021), the rank of the projection subspaces directly influences CL accuracy. While a theoretical analysis of the memory-performance relationship in these settings is an exciting open direction, it is beyond the scope of this work.

**CL theory.** Several theoretical works have emerged recently on understanding forgetting in CL. We discuss those immediately related to our paper; the others are deferred to Appendix A.

Evron et al. (2022) analyzes the ordinary continual learning in the setting with fixed input data and labels, limiting their results to only the optimization aspects of CL. The work by Li et al. (2023) considers the $\ell_2$-regularized CL performance on the two-task fixed design setting; the work by Zhao et al. (2024) also considers the fixed-design setting and evaluates the statistical performance with GRCL on the multi-task linear regression. In comparison, we consider the GRCL in the random design, in which the impact of input data randomness is taken into account. Moreover, all these works consider specific regularizers with fixed memory costs, while our work is the first to theoretically reveal the memory-statistics relation of CL.

## 3    PROBLEM SETUP

**Linear regression under covariate shift.** We set up a 2-task CL setting with two linear regression problems - our results can be extended to CL with more than two tasks without additional technical hurdles. We use $\boldsymbol{x} \in \mathbb{H}$ and $y \in \mathbb{R}$ to denote an input data vector in a Hilbert space (with a finite or a countably infinite dimension $d \leq \infty$ ) and a label variable, respectively. Consider two data distributions $\mathcal{D}^{(1)}$ and $\mathcal{D}^{(2)}$. In the problem of continual learning, we are first given $n$ pairs of data vectors and label variables independently sampled from the first task distribution, and then another $n$ data that are drawn independently from the second distribution, which is denoted by

$$(\boldsymbol{x}_i^{(1)}, y_i^{(1)})_{i=1}^n \sim \mathcal{D}^{(1)}, \quad (\boldsymbol{x}_i^{(2)}, y_i^{(2)})_{i=1}^n \sim \mathcal{D}^{(2)}.$$

For simplicity, we use the infinite-dimensional matrix notation for the dataset: $\boldsymbol{X}^{(t)}$ denotes the linear map from $\mathbb{H}$ to $\mathbb{R}^n$ corresponding to $\boldsymbol{x}_1^{(t)}, \ldots, \boldsymbol{x}_n^{(t)} \in \mathbb{H}$ for tasks $t = 1, 2$. We consider the CL problem under the covariate shift setting, defined by Definition 1 and 2.

**Definition 1** (Covariance conditions). Assume that all entries and the trace of the data covariance matrices of both tasks are finite. Denote the data covariance matrices of the two tasks by

$$\boldsymbol{G} := \mathbb{E}_{\mathcal{D}^{(1)}}[\boldsymbol{x}\boldsymbol{x}^\top], \quad \boldsymbol{H} := \mathbb{E}_{\mathcal{D}^{(2)}}[\boldsymbol{x}\boldsymbol{x}^\top],$$

and denote their eigenvalues respectively by $(\mu_i)_{i \geq 1}$ and $(\lambda_i)_{i \geq 1}$. For convenience, assume that both $\boldsymbol{G}$ and $\boldsymbol{H}$ are strictly positive definite.

**Definition 2** (Model conditions). For each model parameter $\boldsymbol{w}$, define the population risks for the two tasks by

$$\mathcal{R}_1(\boldsymbol{w}) := \mathbb{E}_{\mathcal{D}^{(1)}}(y - \boldsymbol{w}^\top \boldsymbol{x})^2, \qquad \mathcal{R}_2(\boldsymbol{w}) := \mathbb{E}_{\mathcal{D}^{(2)}}(y - \boldsymbol{w}^\top \boldsymbol{x})^2.$$

Assume that a shared optimal parameter exists for both tasks, i.e.,

$$\arg\min \mathcal{R}_1(\boldsymbol{w}) \cap \arg\min \mathcal{R}_2(\boldsymbol{w}) \neq \emptyset.$$

Denote $\boldsymbol{w}^*$ as the minimal-$\ell_2$-norm solution that simultaneously minimizes both tasks, i.e., $\boldsymbol{w}^*$ is the unique solution of the following program:

$$\min \|\boldsymbol{w}\|_2, \quad \text{s.t. } \boldsymbol{w} \in \arg\min \mathcal{R}_1(\boldsymbol{w}) \cap \arg\min \mathcal{R}_2(\boldsymbol{w}).$$

The shared optimal parameter assumption is common in theoretical CL literature (Zhao et al., 2024; Li et al., 2023; Evron et al., 2022), allowing us to focus on the effect of covariate shift. This assumption is mild, since an overparameterized neural network can solve multiple tasks together in practice, demonstrating the existence of shared optimal parameters. Furthermore, our theory is ready to be generalized to allow different optimal parameters by a standard application of the triangle inequality.

**Continual learning.** The goal of CL is to learn a model to minimize the *joint population risk*:

$$\mathcal{R}(\boldsymbol{w}) = \mathcal{R}_1(\boldsymbol{w}) + \mathcal{R}_2(\boldsymbol{w}).$$

Unlike multi-task learning problems, CL is under the constraint that the sampled datasets for the CL tasks are accessed in a sequential manner. Specifically, a CL agent on two tasks has two learning phases where the agent draws samples first from $\mathcal{D}^{(1)}$ and then from $\mathcal{D}^{(2)}$. At the end of the second phase, the agent generates a model parameter that aims to achieve a small joint population risk. To achieve this goal, information from $\mathcal{D}^{(1)}$ needs to be transmitted to the learning phase of $\mathcal{D}^{(2)}$. Therefore, the memory consolidation phase is introduced between the learning phases (Kirkpatrick et al., 2017; Zenke et al., 2017), in which the CL agent retains only limited information from learning $\mathcal{D}^{(1)}$ and feeds it to the second learning phase.

**Ordinary and $\ell_2$-regularized continual learning.** Two basic CL algorithms that are well studied in the literature are referred to by us as *ordinary continual learning* (OCL)(Evron et al., 2022; Li et al., 2023; Zhao et al., 2024) and *$\ell_2$-regularized continual learning* ($\ell_2$-RCL)(Li et al., 2023). In the first learning phase, both OCL and $\ell_2$-RCL perform ordinary least squares with the first dataset. In the memory consolidation phase, they transmit the obtained minimum norm estimator $\boldsymbol{w}^{(1)}$ for the first task. In the second learning phase, OCL finds the parameter $\boldsymbol{w}^{(2)}$ that fits the second dataset while minimizing the $\ell_2$-distance to $\boldsymbol{w}^{(1)}$. The information consolidated and transmitted between the two learning phases is minimal, consisting of only one vector of dimension $d$ (the estimator $\boldsymbol{w}^{(1)}$). However, it is known that this algorithm suffers from *catastrophic forgetting* (Evron et al., 2022; Li et al., 2023). The parameters generated by this training procedure are specified by:

$$\boldsymbol{w}^{(1)} = \left(\boldsymbol{X}^{(1)^\top}\boldsymbol{X}^{(1)}\right)^{-1}\boldsymbol{X}^{(1)^\top}\boldsymbol{y}^{(1)}; \; \boldsymbol{w}^{(2)} = \boldsymbol{w}^{(1)} + \left(\boldsymbol{X}^{(2)^\top}\boldsymbol{X}^{(2)}\right)^{-1}\boldsymbol{X}^{(2)^\top}\left(\boldsymbol{y}^{(2)} - \boldsymbol{X}^{(2)}\boldsymbol{w}^{(1)}\right). \tag{1}$$

With a slight sophistication, $\ell_2$-RCL computes a model parameter $\boldsymbol{w}^{(2)}$ that fits the second dataset under an *isotropic $\ell_2$-penalty* from the previous parameter, with its intensity quantified by $\gamma$:

$$\boldsymbol{w}^{(1)} = \left(\boldsymbol{X}^{(1)^\top}\boldsymbol{X}^{(1)}\right)^{-1}\boldsymbol{X}^{(1)^\top}\boldsymbol{y}^{(1)}; \quad \boldsymbol{w}^{(2)} = \arg\min_{\boldsymbol{w}} \frac{1}{n}\|\boldsymbol{y}^{(2)} - \boldsymbol{X}^{(2)}\boldsymbol{w}\|_2^2 + \gamma\|\boldsymbol{w} - \boldsymbol{w}^{(1)}\|_2^2. \tag{2}$$

Compared to OCL, the additional information saved and transmitted between the two learning phases is only a scalar (the regularization parameter $\gamma$). This shows that $\ell_2$-RCL is still a low-memory-cost algorithm. However, it is proved in Li et al. (2023) that no choice of $\gamma$ can temper the catastrophic forgetting for certain two-task linear regression problems. Therefore, better algorithms involving more complicated memory consolidation need to be considered for better CL performance.

**Generalized $\ell_2$-regularized continual learning.** The primary CL algorithm in our analysis is the *generalized $\ell_2$-regularized continual learning* (GRCL). The first learning phase of GRCL is identical to the OCL. In the memory consolidation phase, GRCL transmits the obtained parameter $\boldsymbol{w}^{(1)}$, as well as a semidefinite matrix $\boldsymbol{\Sigma} \in \mathbb{R}^{d \times d}$. In the second learning phase, the algorithm finds the parameter that fits the second dataset under a quadratic penalty of distance from the previous model parameter, quantified by the metric $\boldsymbol{\Sigma}$. Specifically, GRCL outputs $\boldsymbol{w}^{(2)}$ such that:

$$\boldsymbol{w}^{(1)} = \left(\boldsymbol{X}^{(1)^\top}\boldsymbol{X}^{(1)}\right)^{-1}\boldsymbol{X}^{(1)^\top}\boldsymbol{y}^{(1)}; \quad \boldsymbol{w}^{(2)} = \arg\min_{\boldsymbol{w}} \frac{1}{n}\|\boldsymbol{y}^{(2)} - \boldsymbol{X}^{(2)}\boldsymbol{w}\|_2^2 + \|\boldsymbol{w} - \boldsymbol{w}^{(1)}\|_{\boldsymbol{\Sigma}}^2. \tag{3}$$

The additional regularization matrix $\boldsymbol{\Sigma}$ is usually low-rank and can be stored in the form $\boldsymbol{\Sigma} = \boldsymbol{W}^\top\boldsymbol{W}$, where $\boldsymbol{W} \in \mathbb{R}^{k \times d}$. Compared to OCL, GRCL takes an additional memory of $\boldsymbol{\Sigma}$, which can be expressed with $k$ vectors of dimension $d$. The choice of $\boldsymbol{\Sigma}$ determines the balance between the memory cost and the statistical performance of the GRCL algorithm.

We note that GRCL covers several commonly studied CL algorithms as special cases. Specifically, GRCL becomes OCL when $\boldsymbol{\Sigma} \to \boldsymbol{0}$; when $\boldsymbol{\Sigma} = \gamma\boldsymbol{I}$ for $\gamma > 0$, GRCL becomes $\ell_2$-RCL (Li et al., 2023). The EWC algorithm (Kirkpatrick et al., 2017) is a special case of GRCL when $\boldsymbol{\Sigma}$ is a diagonal matrix. We also note that GRCL has been studied in Zhao et al. (2024); however, they only focus on the optimality of the algorithm where the size of the regularization $\boldsymbol{\Sigma}$ is unlimited, while we focus on the different choice of $\boldsymbol{\Sigma}$ that affects the balance of the memory-statistics trade-off.

**Evaluation metric.** The output parameter $\boldsymbol{w}^{(2)}$ of the CL algorithms is evaluated by the *joint excess risk*, defined by the excess risk of the algorithm compared to the optimal performance:

$$\Delta(\boldsymbol{w}^{(2)}) = \mathcal{R}(\boldsymbol{w}^{(2)}) - \min \mathcal{R}(\cdot).$$

We present a set of assumptions that are used in our analysis.

**Assumption 1** (Well-specified noise). Assume that for the distributions for both tasks $t = 1, 2$, the label variable is given by

$$y^{(t)} = \boldsymbol{x}^{(t)^\top} \boldsymbol{w}^* + \varepsilon^{(t)}, \quad \varepsilon^{(t)} \sim \mathcal{N}(0, \sigma^2),$$

where $\varepsilon^{(t)}$ is independent of $\boldsymbol{x}^{(t)}$ and the shared optimal parameter $\boldsymbol{w}^*$ is defined in Definition 2.

For conciseness, we assume that the two tasks have the same noise level $\sigma^2$. This is not restrictive, and our results can be directly generalized to allow different noise levels as well.

**Assumption 2** (Commutable covariance matrices). Assume that the two data covariance matrices $\boldsymbol{G}$ and $\boldsymbol{H}$ are commutable, namely $\boldsymbol{GH} = \boldsymbol{HG}$. Without loss of generality, we further assume that both $\boldsymbol{G}$ and $\boldsymbol{H}$ are diagonal matrices.

We note that the diagonality of covariance matrices $\boldsymbol{G}$ and $\boldsymbol{H}$ are only for convenience. Our results also hold for commutable covariance matrices since the CL problem is rotation invariant. These assumptions are commonly seen in the literature for the covariate shift scenario (Lei et al., 2021; Wu et al., 2022; Zhao et al., 2024).

**Notation.** For two positive-value functions $f(x)$ and $g(x)$, we write $f(x) \lesssim g(x)$ or $f(x) \gtrsim g(x)$ if $f(x) \leq cg(x)$ or $f(x) \geq cg(x)$ respectively for some absolute constant $c$; and we write $f(x) \eqsim g(x)$ if $f(x) \lesssim g(x) \lesssim f(x)$. For two matrices $\boldsymbol{G}$ and $\boldsymbol{H}$, we denote $\boldsymbol{G} \preceq \boldsymbol{H}$ or $\boldsymbol{G} \succeq \boldsymbol{H}$ if $\boldsymbol{G} - \boldsymbol{H}$ or $\boldsymbol{H} - \boldsymbol{G}$ are PSD matrices respectively. Additional notations are deferred to Appendix B.

## 4 MAIN RESULTS

We investigate the performance of the CL algorithms in the one-hot setting. The one-hot setting is a type of random design setting (Hsu et al., 2012) that is well-studied in previous literature (Zou et al., 2021), where input vectors are sampled from the set of natural bases. Compared to the fixed design in Li et al. (2023), the random design considers the impact of the input distribution randomness.

**Assumption 3** (One-hot setting). Let $(\boldsymbol{e}_i)_{i=1}^d$ be the orthogonal bases for $\mathbb{H}$. Assume that $\mathbb{P}(\boldsymbol{x}^{(1)} = \boldsymbol{e}_i) = \mu_i$ and that $\mathbb{P}(\boldsymbol{x}^{(2)} = \boldsymbol{e}_i) = \lambda_i$ for $i \geq 1$, where $\mu_i, \lambda_i \geq 0$, $\sum_i \mu_i = 1$ and $\sum_i \lambda_i = 1$.

**Risk bounds for joint learning.** When all information in the first learning phase can be accessed in learning the second task, one can solve the two-task CL problem by *joint learning*, in which the output is determined by

$$\boldsymbol{w}_{\texttt{joint}} = \operatorname*{arg\,min}_{\boldsymbol{w}: \boldsymbol{y}^{(1)} = \boldsymbol{X}^{(1)}\boldsymbol{w}, \boldsymbol{y}^{(2)} = \boldsymbol{X}^{(2)}\boldsymbol{w}} \|\boldsymbol{w}\|_2^2. \tag{4}$$

Since joint learning transmits more memory than any other CL algorithm, it is usually used as or an upper bound in evaluating the performance of CL algorithms in the literature (Farajtabar et al., 2020; Saha et al., 2021). We expect that a well-designed CL algorithm can match the performance of joint learning. To this end, the following proposition presents a risk bound for solving the two linear regression tasks with joint learning in the one-hot setting.

**Proposition 1** (Joint learning bound). *Suppose Assumptions 1, 2 and 3 hold. Denote $\mathbb{J}, \mathbb{K}$ such that $\mathbb{J} = \{i : \mu_i > \frac{1}{n}\}$, and $\mathbb{K} = \{i : \lambda_i > \frac{1}{n}\}$. Then for $\boldsymbol{w}_{\texttt{joint}}$ given by (4),*

$$\mathbb{E}\Delta(\boldsymbol{w}_{\texttt{joint}}) = \texttt{bias} + \texttt{variance},$$

*where*

$$\texttt{bias} = \sum_i (1 - \mu_i)^n (1 - \lambda_i)^n (\mu_i + \lambda_i) w_i^{*2}, \quad \texttt{variance} \eqsim \frac{\sigma^2}{n}\Big(|\mathbb{J} \cup \mathbb{K}| + n^2 \sum_{i \in \mathbb{J}^c \cap \mathbb{K}^c} (\mu_i + \lambda_i)^2\Big).$$

We are particularly interested in the CL problems that are *jointly learnable*, i.e., having an $o(1)$ excess risk with joint learning. From Proposition 1, we can see that a necessary (and also sufficient) condition to be jointly learnable is that the "head" of the distributions (i.e., the number of large eigenvalues $|\mathbb{J}|, |\mathbb{K}|$) is $o(n)$, and that the "tail" of the distributions satisfies the following conditions:

$$\sum_{i \in \mathbb{J}^c} \mu_i^2 = o(1/n), \quad \sum_{i \in \mathbb{K}^c} \lambda_i^2 = o(1/n). \tag{5}$$

### 4.1 RISK BOUNDS FOR CONTINUAL LEARNING

Our main result, Theorem 2, provides general risk bounds for GRCL with any regularization matrix $\boldsymbol{\Sigma} \succeq \boldsymbol{0}$. In particular, GRCL reduces to OCL when $\boldsymbol{\Sigma} \to \boldsymbol{0}$ and to $\ell_2$-RCL when $\boldsymbol{\Sigma} = \gamma \boldsymbol{I}$ for $\gamma > 0$. By establishing matching upper and lower bounds for the joint excess risk of GRCL as a function of $\boldsymbol{\Sigma}$, we are able to analyze the impact of regularization on different CL algorithms.

**Theorem 2.** *Suppose Assumptions 1, 2, and 3 hold. Denote $\mathbb{J}, \mathbb{K}$ such that $\mathbb{J} = \left\{ i : \mu_i \geq \frac{1}{n} \right\}$, and $\mathbb{K} = \left\{ i : \lambda_i \geq \frac{1}{n} \right\}$. Consider the PSD regularization matrix $\boldsymbol{\Sigma} = \mathrm{diag}(\gamma_i)_{i=1}^p$, where $\boldsymbol{\Sigma}$ is commutable with the data covariance matrices $\boldsymbol{G}, \boldsymbol{H}$. Then for the GRCL output (3), it holds that*

$$\mathbb{E}\Delta(\boldsymbol{w}^{(2)}) = \texttt{bias} + \texttt{variance},$$

*where the expectations of* `bias` *and* `variance` *with respect $\boldsymbol{x}^{(1)}$ and $\boldsymbol{x}^{(2)}$ satisfy*

$$\texttt{bias} \asymp \langle (\boldsymbol{G} + \boldsymbol{H})(\boldsymbol{I} - \boldsymbol{G})^n (\boldsymbol{\Sigma}^2(\boldsymbol{\Sigma} + \boldsymbol{H})^{-2} + (\boldsymbol{I} - \boldsymbol{H})^n), \boldsymbol{w}^* \boldsymbol{w}^{*\top} \rangle,$$

$$\texttt{variance} \asymp \sigma^2 \big( \langle (\boldsymbol{G} + \boldsymbol{H})(\boldsymbol{\Sigma}^2(\boldsymbol{\Sigma} + \boldsymbol{H})^{-2} + (\boldsymbol{I} - \boldsymbol{H})^n), \frac{1}{n} \boldsymbol{G}_{\mathbb{J}}^{-1} + n \boldsymbol{G}_{\mathbb{J}^c} \rangle \tag{6}$$

$$+ \langle \boldsymbol{G} + \boldsymbol{H}, \frac{1}{n}(\boldsymbol{H}_{\mathbb{K}} + \boldsymbol{\Sigma}_{\mathbb{K}})^{-2} \boldsymbol{H}_{\mathbb{K}} + n(\boldsymbol{I}_{\mathbb{K}^c} + n\boldsymbol{\Sigma}_{\mathbb{K}^c})^{-2} \boldsymbol{H}_{\mathbb{K}^c} \rangle \big).$$

Theorem 2 tightly characterizes the risk of GRCL in the one-hot setting. Specifically, the CL risk consists of a bias term and a variance term, and we give matching upper and lower bounds on both parts. These bounds are functions of the CL task distribution parameters $(\boldsymbol{G}, \boldsymbol{H}, \sigma^2, n)$ and the regularization matrix $\boldsymbol{\Sigma}$. In the following, we use this result to analyze catastrophic forgetting in OCL and $\ell_2$-RCL and demonstrate how GRCL mitigates forgetting with arbitrary memory $\boldsymbol{\Sigma}$.

**Catastrophic forgetting of OCL and $\ell_2$-RCL.** Compared to GRCL, OCL retains minimal memory carried over to the second learning phase. As a result, its statistical performance is expected to be inferior to joint learning. The following corollary quantifies this difference:

**Corollary 3.** *Under the conditions of Theorem 2, for the OCL output (1), we have*

$$\mathbb{E}\Delta(\boldsymbol{w}^{(2)}) \asymp \mathbb{E}\Delta(\boldsymbol{w}_{\texttt{joint}}) + \frac{\sigma^2}{n} \left( \sum_{i \in \mathbb{K}} \frac{\mu_i}{\lambda_i} + n^2 \sum_{i \in \mathbb{J} \cap \mathbb{K}^c} \mu_i \lambda_i \right). \tag{7}$$

As indicated by Corollary 3, for OCL to achieve an $o(1)$ CL excess risk in a jointly learnable problem, it is sufficient and necessary to have

$$\sum_{i \in \mathbb{K}} \frac{\mu_i}{\lambda_i} + n^2 \sum_{i \in \mathbb{J} \cap \mathbb{K}^c} \mu_i \lambda_i = o(n). \tag{8}$$

The expression in (8) quantifies the statistical gap between OCL and joint learning. To minimize this gap, the intermediate eigenvalue space of $\boldsymbol{H}$ near $1/n$ must align with the small eigenvalue space of $\boldsymbol{G}$. When (8) is not satisfied, OCL experiences a constant statistical underperformance due to catastrophic forgetting. Similarly, $\ell_2$-RCL is also a low-memory CL algorithm, carrying only an additional memory of the regularization parameter $\gamma$ compared to OCL. The following corollary helps us understand the performance of $\ell_2$-RCL.

**Corollary 4.** *Under the conditions of Theorem 2, for the $\ell_2$-RCL output (2), we have*

$$\mathbb{E}\Delta(\boldsymbol{w}^{(2)}) \lesssim \mathbb{E}\Delta(\boldsymbol{w}_{\texttt{joint}}) + (\gamma + \tfrac{1}{n})\|\boldsymbol{w}^*\|_2^2 + \frac{\sigma^2}{n} \sum_{i \in \mathbb{J} \cup \mathbb{K}} \left( \frac{\mu_i}{\lambda_i + \frac{1}{n} + \gamma} + \frac{\gamma}{\mu_i + \frac{1}{n}} \right). \tag{9}$$

Corollary 4 shows that for $\ell_2$-RCL to achieve an $o(1)$ risk in jointly learnable problem, it suffices that

$$\gamma = o(1), \quad \sum_{i \in \mathbb{J} \cup \mathbb{K}} \left( \frac{\mu_i}{\lambda_i + \frac{1}{n} + \gamma} + \frac{\gamma}{\mu_i + \frac{1}{n}} \right) = o(n), \tag{10}$$

holds for some $\gamma > 0$. It is clear that (10) is a weaker set of requirements than (8) as OCL is a special case of $\ell_2$-RCL when $\gamma \to 0$. However, there still exist CL tasks that are jointly solvable yet not solvable by $\ell_2$-RCL. To demonstrate the failure of OCL and $\ell_2$-RCL, we show two examples of CL problem: the first is not solvable by OCL, and the second is further not solvable by $\ell_2$-RCL.

**Example 5.** *Under the conditions of Theorem 2, suppose $\sigma^2 = 1$ and $\|\boldsymbol{w}\|_2^2 = 1$. Then, for $\boldsymbol{G} = \mathrm{diag}(\mu_i)_{i=1}^d$ and $\boldsymbol{H} = \mathrm{diag}(\lambda_i)_{i=1}^d$:*

- *If $\mu_1 = 1$ and $\lambda_1 = 1/n$, then the OCL excess risk satisfies $\mathbb{E}\Delta(\boldsymbol{w}^{(2)}) = \Omega(1)$.*

- *Suppose that for $i < \frac{1}{3}n$, $\mu_i = 1$ and $\lambda_i = \frac{1}{n}$, while for $\frac{1}{3}n \le i < \frac{2}{3}n$, $\mu_i = \frac{1}{n}$ and $\lambda_i = 1$. For all other $i \ge \frac{2}{3}n$, $\mu_i = \lambda_i = 0$. Then for any regularization parameter $\gamma > 0$, the $\ell_2$-RCL excess risk satisfies $\mathbb{E}\Delta(\boldsymbol{w}^{(2)}) = \Omega(1)$.*

Thus, both OCL and $\ell_2$-RCL cannot match the performance of joint learning across all CL tasks.

**GRCL avoids forgetting with full memory.** To enhance CL performance, GRCL introduces the regularization matrix $\boldsymbol{\Sigma}$ to transfer information from the first task to the training phase of the second task. The following corollary shows that with sufficient memory and appropriate $\boldsymbol{\Sigma}$, GRCL can achieve performance comparable to joint learning.

**Corollary 6.** *Under the same conditions as Theorem 2, consider the regularization $\boldsymbol{\Sigma} = \mathrm{diag}(\gamma_i)_{i=1}^d$ with $\gamma_i = \mu_i$ for $\mu_i \ge 1/n$ and $\gamma_i = 0$ otherwise. Then, for the GRCL output (6), we have*

$$\mathbb{E}\Delta(\boldsymbol{w}^{(2)}) \lesssim \mathbb{E}\Delta(\boldsymbol{w}_{\mathtt{joint}}).$$

Corollary 6 demonstrates that by selecting a regularization matrix of size $|\mathbb{J}|$ that captures all the top eigenvalues of $\boldsymbol{G}$ greater than $1/n$, GRCL can attain joint learning performance and eliminate catastrophic forgetting from OCL and $\ell_2$-RCL.

## 4.2 MEMORY-STATISTICS TRADE-OFF OF GRCL

In this part, we further investigate the constrained memory setting to analyze the trade-off between memory consumption and statistical performance in GRCL. The memory cost of GRCL is measured by the number of distinct eigenvectors in the regularization matrix $\boldsymbol{\Sigma}$. Our results demonstrate that the GRCL excess risk decreases as the memory allocated to regularization increases, and vice versa.

**Catastrophic forgetting in low-memory settings.** Our analysis in Section 4.1 reveals that both OCL and $\ell_2$-RCL exhibit catastrophic forgetting with limited memory. We extend this understanding to GRCL through the following example:

**Example 7.** *Under the conditions of Theorem 2, suppose $\sigma^2 = 1$ and $\|\boldsymbol{w}\|_2^2 = 1$. Then, for $\boldsymbol{G} = \mathrm{diag}(\mu_i)_{i=1}^d$ and $\boldsymbol{H} = \mathrm{diag}(\lambda_i)_{i=1}^d$,*

- *For any positive constant $k << N$, if $\mu_i = 1/(k+1)$ and $\lambda_i = 1/n$ for $1 \le i \le k+1$, then for any regularizer $\boldsymbol{\Sigma}$ of size $k$, the GRCL excess risk satisfies $\mathbb{E}\Delta(\boldsymbol{w}^{(2)}) = \Omega(1)$.*

This constructed scenario generalizes the OCL failure on Example 5 to GRCL: recall that in the previous example, there exists one eigenvalue mismatch on the dominant eigenspace ($\mu_1 = 1$, $\lambda_1 = 1/n$). Since OCL lacks the memory to store this mismatched eigenvalue, catastrophic forgetting occurs. Similarly, in Example 7, the $k + 1$ dominant features in $\boldsymbol{G}$ coupled with intermediate $\boldsymbol{H}$ eigenvalues create an insurmountable memory bottleneck – no $k$-dimensional regularization in GRCL can mitigate forgetting across $k + 1$ orthogonal directions.

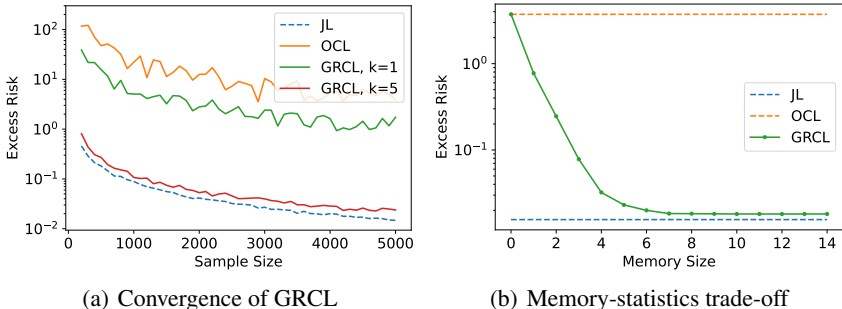

(a) Convergence of GRCL  (b) Memory-statistics trade-off

Figure 1: Expected excess risk vs. (a) sample size $n$, and (b) memory size $k$ for generalized $\ell_2$-regularized continual learning (GRCL), compared with joint learning (JL) and ordinary continual learning (OCL). For each point in each curve, the y-axis represents the expected CL excess risk (in the logarithmic scale). The dimension of the task is $d = 200$. The sample size is fixed at $n = 5000$ in (b). The excess risk is computed by taking an empirical average over 20 independent runs.

**Tempered forgetting in moderate memory settings.** In Corollary 6 and Example 7, we showed that GRCL can suffer catastrophic forgetting when memory is limited, and that GRCL performance matches joint learning with sufficient memory. The following example illustrates GRCL performance dynamics with intermediate memory.

**Example 8.** *Under the same conditions in Theorem 2, let $G = \mathrm{diag}(\mu_i)_{i=1}^\infty$, where $\mu_i = i^{-\alpha}$ with $\alpha > 1$. Consider the top-$k$ regularization $\Sigma = \mathrm{diag}(\gamma_i)_{i=1}^\infty$ with $\gamma_i = \mu_i$ for $i \leq k$ and $\gamma_i = 0$ otherwise, where $1 \leq k \leq \sqrt[\alpha]{n}$. Then, for the GRCL output (6), we have*

$$\mathbb{E}\Delta(\boldsymbol{w}^{(2)}) \lesssim \mathbb{E}\Delta(\boldsymbol{w}_{\mathtt{joint}}) \cdot \left(1 + \frac{n}{k^\alpha}\right). \tag{11}$$

In Example 8, we observe that as the memory size $k$ of GRCL increases, the ratio of GRCL's excess risk to the joint learning risk decreases at a rate of $\frac{n}{k^\alpha}$, approaching the joint learning performance at $k = \sqrt[\alpha]{n}$. This example demonstrates that GRCL with an intermediate memory constraint can partially alleviate catastrophic forgetting.

**Numerical Experiments.** We conduct numerical experiments in Gaussian design to verify our theoretical results; additional experiments on practical CL datasets are deferred to Appendix E.2. Gaussian design is a more widely used random design setting in the statistical learning literature (Hsu et al., 2012; Bartlett et al., 2020).

**Assumption 4** (Gaussian design). Assume that $\boldsymbol{x}^{(1)} = \boldsymbol{G}^{1/2}\boldsymbol{z}^{(1)}$ and that $\boldsymbol{x}^{(2)} = \boldsymbol{H}^{1/2}\boldsymbol{z}^{(2)}$, where $\boldsymbol{z}^{(1)}, \boldsymbol{z}^{(2)} \sim \mathcal{N}(0, \boldsymbol{I})$.

We consider a CL problem adapted from Wu et al. (2022) and Li et al. (2023)(see Appendix E.1 for details). As shown in Figure 1(a), OCL suffers from a constant excess risk due to catastrophic forgetting. In contrast, GRCL's performance improves with memory size $k$, matching the convergence rate of joint learning at $k = 5$. To further verify the memory-statistics trade-off of GRCL, we vary the memory size $k$ in Figure 1(b), which shows that GRCL reduces excess risk with larger $k$ and reaches joint learning performance with $k \leq 15$ for the given CL problem.

# 5 EXTENSIONS

In this section, we discuss extensions of our results and messages to some more general CL settings.

**OCL in Gaussian design.** From the numerical experiments, we observe that our main results for the one-hot setting in Section 4 also empirically hold for the Gaussian design. However, our main theoretical results on GRCL cannot be directly generalized to the Gaussian design due to technical hurdles. As a preliminary step, we provide the lower and upper bounds of the OCL excess risk in Theorems 18 and 19 in Appendix C.

We would like to highlight the different behaviors in Gaussian design from the one-hot setting: the following example shows the additional source of forgetting in Gaussian design.

**Example 9.** *Under the same condition in Theorem 18, suppose that $\sigma^2 = 1$, $\|\boldsymbol{w}\|_2^2 = 1$. Then for $(\mu_i)_{i=1}^n$ and $(\lambda_i)_{i=1}^n$,*

- *If $\mu_1 = 1$ and $\lambda_1 = 1/n$, then the OCL excess risk and the forgetting is $\Omega(1)$.*

- *If $\mu_i = i^{-1} \log^{-\alpha}(i+1)$ and $\lambda_i = i^{-1} \log^{-\beta}(i+1)$ for some constants $\alpha, \beta > 1$ that satisfies $\beta \leq \alpha + 1$, then the OCL excess risk and the forgetting is $\Omega(\log^{\beta-\alpha-1} n)$.*

The first problem in Example 9 illustrates how a single difference in essential features can result in forgetting in the Gaussian case, which is also described in the one-hot case in Example 5. However, the second problem in Example 9 shows that forgetting can also result from differences in the less significant features on the tail.

**CL in the NTK regime.** We study the extension of our main messages in linear regression to the NTK regime of neural networks. Consider a 2-task CL setting in which the goal is to learn a neural network model $f_{\boldsymbol{w}}(\boldsymbol{x})$ specified by parameter $\boldsymbol{w} \in \mathcal{R}^d$. Lee et al. (2019) proved that under the NTK regime, neural networks evolve as a linear model:

$$f_{\boldsymbol{w}}(\boldsymbol{x}) = f^{(0)}(\boldsymbol{x}) + \langle \nabla_{\boldsymbol{w}} f^{(0)}(x), \boldsymbol{w} - \boldsymbol{w}^{(0)} \rangle, \tag{12}$$

where $f^{(0)}(\cdot)$ denotes the model with $\boldsymbol{w}^{(0)} = \boldsymbol{0}$ as the initial weight. Under this setting, we consider the *convergence of GRCL with gradient descent (GD)*. We can reduce the GRCL in the NTK setting to the linear algorithm in (3), providing that we give the following inputs $\boldsymbol{x}', y'$ to the linear model:

$$\boldsymbol{x}'^{(1)} = \phi(\boldsymbol{x}^{(1)}), \ \boldsymbol{x}'^{(2)} = \phi(\boldsymbol{x}^{(2)}), \ y'^{(1)} = y^{(1)} - f^{(0)}(\boldsymbol{x}^{(1)}), \ y'^{(2)} = y^{(2)} - f^{(1)}(\boldsymbol{x}^{(2)}). \tag{13}$$

With the covariance matrices $\boldsymbol{G} := \mathbb{E}_{\mathcal{D}^{(1)}}[\phi(\boldsymbol{x}^{(1)})\phi(\boldsymbol{x}^{(1)})^\top]$, $\boldsymbol{H} := \mathbb{E}_{\mathcal{D}^{(2)}}[\phi(\boldsymbol{x}^{(2)})\phi(\boldsymbol{x}^{(2)})^\top]$ defined accordingly, which actually represents the Hessian of the model, we have our main results:

**Theorem 10** (Theorem 2 in the NTK regime). *Suppose Assumption 5 holds. Denote $\boldsymbol{G} := \mathbb{E}_{\mathcal{D}^{(1)}}[\phi(\boldsymbol{x}^{(1)})\phi(\boldsymbol{x}^{(1)})^\top]$, $\boldsymbol{H} := \mathbb{E}_{\mathcal{D}^{(2)}}[\phi(\boldsymbol{x}^{(2)})\phi(\boldsymbol{x}^{(2)})^\top]$. Then for the GRCL output $\boldsymbol{w}^{(2)}$,*

$$\mathbb{E}\Delta(\boldsymbol{w}^{(2)}) = \texttt{bias} + \texttt{variance},$$

*where the bias and variance satisfy* (6) *defined in Theorem 2 with $\boldsymbol{G}, \boldsymbol{H}$ defined as above.*

Therefore, our main results in Theorem 2 as well as its collollaries and messages still hold in the NTK regime and can be applied in general neural networks.

**Multi-task CL.** We study the extension of our main messages to the multi-task CL setting. For tasks $t = 1, \ldots, T$ where $T \geq 3$, the goal is to learn a model $\boldsymbol{w}$ to minimize the joint population risk $\mathcal{R}(\boldsymbol{w}) = \sum_t \mathcal{R}_t(\boldsymbol{w})$. Corollary 11 extends the message that with sufficient memory and appropriate regularization, GRCL can match the performance of joint training in multi-task CL. For details, please refer to Appendix D.2.

**Corollary 11** (Corollary 6 in the multi-task setting). *Suppose Assumptions 1", 2", 3" hold. Consider the regularization $\boldsymbol{\Sigma}^{(t)} = \mathrm{diag}(\gamma_i^{(t)})_{i=1}^d$ for $t = 1, \ldots, T$, with $\gamma_i^{(t)} = \mu_i^{(t-1)}$ for $\mu_i^{(t-1)} \geq 1/n$ and $\gamma_i^{(t)} = 0$ otherwise. Then, for the GRCL output $\boldsymbol{w}^{(T)}$, we have*

$$\mathbb{E}\Delta(\boldsymbol{w}^{(T)}) \lesssim \mathbb{E}\Delta(\boldsymbol{w}_{\texttt{joint}}). \tag{14}$$

Combining with the existing Example 7, where low-memory CL behaves poorly, we deliver the message that there is a provable memory-statistics trade-off in multi-task CL.

**Practical algorithms.** We now examine how to generate the regularization matrix $\boldsymbol{\Sigma}$ from Corollary 6, which achieves joint learning performance in GRCL. Recall that $\boldsymbol{\Sigma}$ is computed after the first training phase when we have access to the sampled training data $\boldsymbol{X}^{(1)}$, and the goal is to memorize the top eigenvalues and their corresponding eigenvectors of $\boldsymbol{G}$. In the one-hot setting, we can simply store all distinct $\boldsymbol{x}_i^{(1)}$ as the eigenvectors and use their frequencies as an unbiased estimation of

eigenvalues. This approach has a 0.999 probability of capturing the eigenvectors with eigenvalues $\mu_i > 10/n$. As a result, this makes GRCL a practical algorithm without requiring prior knowledge of the true covariance parameter. In the case of Gaussian distributions or practical datasets, linear sketching methods such as CountSketch (Charikar et al., 2002) can be used to specify the memory utilized in GRCL, where the GRCL algorithm corresponds to the Sketched Structural Regularization method proposed by Li et al. (2021).

## 6 CONCLUSIONS

We consider the generalized $\ell_2$-regularized continual learning algorithm with two linear regression tasks in the random design setting. We derive lower and upper bounds for the algorithm. We show a provable trade-off between the memory size and the statistical performance of the algorithm, which can be adjusted by the regularization matrix. We show that catastrophic forgetting occurs when no regularization is added and that a well-designed structural regularization can fully mitigate this issue. We demonstrate the memory-statistics trade-off of generalized $\ell_2$-regularized continual learning in random design with concrete examples and experiments.

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

## A ADDITIONAL RELATED WORKS

**CL practice.** Over the past decade, continual learning has found applications in computer vision (Qu et al., 2021) and, more recently, in large language models (Wu et al., 2024). A long list of practical CL methods has been proposed to address the catastrophic forgetting problem of neural networks. These CL methods can be roughly divided into four categories: regularization-based methods (e.g., Kirkpatrick et al. (2017)), replay-based methods (e.g., Chaudhry et al. (2019)), architecture-based methods (e.g., Serra et al. (2018)), and projection-based methods (e.g., Saha et al. (2021)). Among the regularization-based methods, early structural regularizers including EWC (Kirkpatrick et al., 2017) and MAS (Aljundi et al., 2018) are widely used since they take only $\mathcal{O}\left(d\right)$ memory by applying diagonal approximation to the importance matrix. More complex structural regularizers with higher memory costs are proposed later for better CL performance, including Structured Laplacian Approximation (Ritter et al., 2018) and Sketched Structural Regularization (Li et al., 2021). Beyond structural regularization, replay-based methods reduce forgetting by retaining the memory of old data and replaying them in training new tasks, and projection-based methods reduce forgetting by projecting the gradient update into a memorized subspace. One can refer to Parisi et al. (2019); Wang et al. (2024) for a comprehensive overview.

**CL theory.** In addition to the works by Zhao et al. (2024); Li et al. (2023); Evron et al. (2022) introduced in Section 2, in this section we cover additional CL theory works in the literature.

The theoretical effect of full Hessian regularization on CL has been mentioned in several papers to different extents, including Peng et al. (2023) and Evron et al. (2023). Peng et al. (2023) mainly focuses on the optimal solution space of general CL problems and how different CL paradigms (including regularization-based methods) approximately solve CL by approaching that solution space, and Evron et al. (2023) mainly studies the continual linear classification scenario. In these two works, it is briefly mentioned and proved in optimization scenarios that a full Hessian regularization can fully mitigate forgetting when measured on the input dataset. Compared to these papers, we investigate further by characterizing the memory-performance trade-off as well as how much memory is required to prevent forgetting, since the Hessian is usually too large to store in the GPU memory.

The works by Goldfarb & Hand (2023); Goldfarb et al. (2024) consider CL settings in which the first task input dataset is considered random. However, in both works, the second dataset is dependent on the first dataset, whereas in our work, the two tasks are independent. As a result, their similarity metric is transition-dependent while ours is task-dependent. Similar to Evron et al. (2022), Goldfarb et al. (2024) also only considers the optimization error.

The works by Heckel (2022); Lin et al. (2023) consider the statistical error in CL. However, they additionally assumed that the data covariance matrices are identity matrices for all tasks, while the optimal parameter for each task is different, limiting their generalization results to the underparameterized regime. This reflects the task-incremental CL setting (Van de Ven & Tolias, 2019), while we focus on the domain-incremental CL setting where all tasks share the same optimal parameter, but the data covariance matrices change across different tasks. Therefore, our results are not directly comparable with theirs.

The rest of the papers are not directly connected to ours. In particular, Doan et al. (2021); Bennani et al. (2020) studies the forgetting and the generalization error of the Orthogonal Gradient Descent method (Farajtabar et al., 2020) in the NTK regime (Jacot et al., 2018).

## B PROOF OF ONE-HOT GRCL

**Additional notations.** For any index set $\mathbb{K}$, we denote its complement by $\mathbb{K}^c$. For any index set $\mathbb{K}$ and any matrix $M \in \mathcal{R}^{n \times d}$, denote $M_{\mathbb{K}}$ as the matrix comprising the $i$-th columns that satisfy $i \in \mathbb{K}$; additionally, for any matrix $M \in \mathcal{R}^{d \times d}$, denote $M_{\mathbb{K}}$ as the submatrix of $M$ comprising the $i$-th rows and columns that satisfy $i \in \mathbb{K}$. We note that all matrix inverses used in this paper are Moore-Penrose pseudoinverses.

### B.1 PRELIMINARIES

**Computing $\boldsymbol{w}^{(1)}$.** We first compute $\boldsymbol{w}^{(1)}$. By definition, we have

$$
\begin{aligned}
\boldsymbol{w}^{(1)} &= \big(\boldsymbol{X}^{(1)\top}\boldsymbol{X}^{(1)}\big)^{-1}\boldsymbol{X}^{(1)\top}\boldsymbol{y} \\
&= \big(\boldsymbol{X}^{(1)\top}\boldsymbol{X}^{(1)}\big)^{-1}\boldsymbol{X}^{(1)\top}\boldsymbol{X}^{(1)}\cdot\boldsymbol{w}^* + \big(\boldsymbol{X}^{(1)\top}\boldsymbol{X}^{(1)}\big)^{-1}\boldsymbol{X}^{(1)\top}\cdot\boldsymbol{\varepsilon}^{(1)}.
\end{aligned}
$$

Denote $\mathcal{P}^{(1)} := \boldsymbol{I} - \big(\boldsymbol{X}^{(1)\top}\boldsymbol{X}^{(1)}\big)^{-1}\boldsymbol{X}^{(1)\top}\boldsymbol{X}^{(1)}$. Therefore

$$
\boldsymbol{w}^{(1)} - \boldsymbol{w}^* = -\mathcal{P}^{(1)}\cdot\boldsymbol{w}^* + \big(\boldsymbol{X}^{(1)\top}\boldsymbol{X}^{(1)}\big)^{-1}\boldsymbol{X}^{(1)\top}\cdot\boldsymbol{\varepsilon}^{(1)}.
$$

Now noticing that

$$
\mathbb{E}\boldsymbol{\varepsilon}^{(1)} = 0, \quad \mathbb{E}\boldsymbol{\varepsilon}^{(1)}\boldsymbol{\varepsilon}^{(1)\top} = \sigma^2\boldsymbol{I},
$$

so the covariance of $\boldsymbol{w}^{(1)} - \boldsymbol{w}^*$ is

$$
\mathbb{E}_\varepsilon(\boldsymbol{w}^{(1)} - \boldsymbol{w}^*)(\boldsymbol{w}^{(1)} - \boldsymbol{w}^*)^\top = \mathcal{P}^{(1)}\cdot\boldsymbol{w}^*\boldsymbol{w}^{*\top}\cdot\mathcal{P}^{(1)} + \sigma^2\big(\boldsymbol{X}^{(1)\top}\boldsymbol{X}^{(1)}\big)^{-1}\boldsymbol{X}^{(1)\top}\boldsymbol{X}^{(1)}\big(\boldsymbol{X}^{(1)\top}\boldsymbol{X}^{(1)}\big)^{-1}
$$

$$
\tag{15}
$$

$$
= \mathcal{P}^{(1)}\cdot\boldsymbol{w}^*\boldsymbol{w}^{*\top}\cdot\mathcal{P}^{(1)} + \sigma^2\big(\boldsymbol{X}^{(1)\top}\boldsymbol{X}^{(1)}\big)^{-1}. \tag{16}
$$

**Computing $\boldsymbol{w}^{(2)}$.** Then we compute $\boldsymbol{w}^{(2)}$. By the first-order optimality condition, we have

$$
\frac{1}{n}\boldsymbol{X}^{(2)\top}\big(\boldsymbol{X}^{(2)}\boldsymbol{w}^{(2)} - \boldsymbol{y}^{(2)}\big) + \boldsymbol{\Sigma}(\boldsymbol{w}^{(2)} - \boldsymbol{w}^{(1)}) = 0,
$$

which implies

$$
\begin{aligned}
\boldsymbol{w}^{(2)} &= \big(\boldsymbol{X}^{(2)\top}\boldsymbol{X}^{(2)} + n\boldsymbol{\Sigma}\big)^{-1}\big(\boldsymbol{X}^{(2)\top}\boldsymbol{y}^{(2)} + n\boldsymbol{\Sigma}\boldsymbol{w}^{(1)}\big) \\
&= \big(\boldsymbol{X}^{(2)\top}\boldsymbol{X}^{(2)} + n\boldsymbol{\Sigma}\big)^{-1}\big(\boldsymbol{X}^{(2)\top}\boldsymbol{X}^{(2)}\boldsymbol{w}^* + \boldsymbol{X}^{(2)\top}\boldsymbol{\epsilon}^{(2)} + n\boldsymbol{\Sigma}\boldsymbol{w}^{(1)}\big).
\end{aligned}
$$

Then we have

$$
\begin{aligned}
\boldsymbol{w}^{(2)} - \boldsymbol{w}^* &= \big(\boldsymbol{X}^{(2)\top}\boldsymbol{X}^{(2)} + n\boldsymbol{\Sigma}\big)^{-1} \\
&\quad \cdot \big(\boldsymbol{X}^{(2)\top}\boldsymbol{X}^{(2)}\boldsymbol{w}^* + \boldsymbol{X}^{(2)\top}\boldsymbol{\epsilon}^{(2)} + n\boldsymbol{\Sigma}\boldsymbol{w}^{(1)} - (\boldsymbol{X}^{(2)\top}\boldsymbol{X}^{(2)} + n\boldsymbol{\Sigma})\boldsymbol{w}^*\big) \\
&= \big(\boldsymbol{X}^{(2)\top}\boldsymbol{X}^{(2)} + n\boldsymbol{\Sigma}\big)^{-1}\big(n\boldsymbol{\Sigma}(\boldsymbol{w}^{(1)} - \boldsymbol{w}^*) + \boldsymbol{X}^{(2)\top}\boldsymbol{\epsilon}^{(2)}\big).
\end{aligned}
$$

Similarly, notice that

$$
\mathbb{E}\boldsymbol{\epsilon}^{(2)} = 0, \quad \mathbb{E}\boldsymbol{\epsilon}^{(2)}\boldsymbol{\epsilon}^{(2)\top} = \sigma^2\boldsymbol{I},
$$

so the covariance of $\boldsymbol{w}^{(2)} - \boldsymbol{w}^*$ is

$$
\begin{aligned}
&\mathbb{E}(\boldsymbol{w}^{(2)} - \boldsymbol{w}^*)(\boldsymbol{w}^{(2)} - \boldsymbol{w}^*)^\top \\
&= \big(\boldsymbol{X}^{(2)\top}\boldsymbol{X}^{(2)} + n\boldsymbol{\Sigma}\big)^{-1}n\boldsymbol{\Sigma}\cdot\mathbb{E}(\boldsymbol{w}^{(1)} - \boldsymbol{w}^*)(\boldsymbol{w}^{(1)} - \boldsymbol{w}^*)^\top\cdot n\boldsymbol{\Sigma}\big(\boldsymbol{X}^{(2)\top}\boldsymbol{X}^{(2)} + n\boldsymbol{\Sigma}\big)^{-1} \\
&\quad + \sigma^2\big(\boldsymbol{X}^{(2)\top}\boldsymbol{X}^{(2)} + n\boldsymbol{\Sigma}\big)^{-1}\cdot\boldsymbol{X}^{(2)\top}\boldsymbol{X}^{(2)}\cdot\big(\boldsymbol{X}^{(2)\top}\boldsymbol{X}^{(2)} + n\boldsymbol{\Sigma}\big)^{-1}.
\end{aligned}
$$

Denote $\mathcal{P}_{\boldsymbol{\Sigma}}^{(2)} = \big(\boldsymbol{X}^{(2)\top}\boldsymbol{X}^{(2)} + n\boldsymbol{\Sigma}\big)^{-1}n\boldsymbol{\Sigma}$. Therefore

$$
\begin{aligned}
&\mathbb{E}(\boldsymbol{w}^{(2)} - \boldsymbol{w}^*)(\boldsymbol{w}^{(2)} - \boldsymbol{w}^*)^\top \\
&= \mathcal{P}_{\boldsymbol{\Sigma}}^{(2)}\cdot\mathbb{E}(\boldsymbol{w}^{(1)} - \boldsymbol{w}^*)(\boldsymbol{w}^{(1)} - \boldsymbol{w}^*)^\top\cdot\mathcal{P}_{\boldsymbol{\Sigma}}^{(2)} \\
&\quad + \sigma^2\big(\boldsymbol{X}^{(2)\top}\boldsymbol{X}^{(2)} + n\boldsymbol{\Sigma}\big)^{-1}\cdot\boldsymbol{X}^{(2)\top}\boldsymbol{X}^{(2)}\cdot\big(\boldsymbol{X}^{(2)\top}\boldsymbol{X}^{(2)} + n\boldsymbol{\Sigma}\big)^{-1}. \tag{17}
\end{aligned}
$$

**Risk decomposition.** According to the risk definition and the assumption on the noise, we have

$$
\begin{aligned}
\mathcal{R}_1(\boldsymbol{w}) &= \frac{1}{n}\mathbb{E}\big\|\boldsymbol{X}^{(1)}\boldsymbol{w} - \boldsymbol{y}^{(1)}\big\|_2^2 \\
&= \frac{1}{n}\mathbb{E}\big\|\boldsymbol{X}^{(1)}\boldsymbol{w} - \boldsymbol{X}^{(1)}\boldsymbol{w}^* - \varepsilon^{(1)}\big\|_2^2 \\
&= (\boldsymbol{w} - \boldsymbol{w}^*)^\top \boldsymbol{G}(\boldsymbol{w} - \boldsymbol{w}^*) + \sigma^2 \\
&= \langle \boldsymbol{G},\ (\boldsymbol{w} - \boldsymbol{w}^*)(\boldsymbol{w} - \boldsymbol{w}^*)^\top \rangle + \sigma^2.
\end{aligned}
\tag{18}
$$

Similarly, the risk for the second task is

$$
\mathcal{R}_2(\boldsymbol{w}) = \langle \boldsymbol{H},\ (\boldsymbol{w} - \boldsymbol{w}^*)(\boldsymbol{w} - \boldsymbol{w}^*)^\top \rangle + \sigma^2,
\tag{19}
$$

and the joint population risk is

$$
\mathcal{R}(\boldsymbol{w}) = \langle \boldsymbol{G} + \boldsymbol{H},\ (\boldsymbol{w} - \boldsymbol{w}^*)(\boldsymbol{w} - \boldsymbol{w}^*)^\top \rangle + 2\sigma^2.
\tag{20}
$$

## B.2 Proof of Theorem 2

To compute the joint population risk and the forgetting, we give several useful lemmas.

**Lemma 12** (Bound on the head of $\big(\boldsymbol{X}^{(t)\top}\boldsymbol{X}^{(t)}\big)^{-1}$). *Suppose Assumptions 1 and 3 hold. Denote* $\mathbb{K}$ *such that* $\mathbb{K} = \big\{i : \lambda_i \geq \frac{1}{n}\big\}$. *Then, for* $n \geq 2$,

$$
\frac{1}{2n}\boldsymbol{H}_{\mathbb{K}}^{-1} \preceq \mathbb{E}\big(\boldsymbol{X}_{\mathbb{K}}^{(2)\top}\boldsymbol{X}_{\mathbb{K}}^{(2)}\big)^{-1} \preceq \frac{12}{n}\boldsymbol{H}_{\mathbb{K}}^{-1}.
$$

*Similarly, denote* $\mathbb{J}$ *such that* $\mathbb{J} = \big\{i : \mu_i \geq \frac{1}{n}\big\}$. *Then, for* $n \geq 2$,

$$
\frac{1}{4n}\boldsymbol{G}_{\mathbb{J}}^{-1} \preceq \mathbb{E}\big(\boldsymbol{X}_{\mathbb{J}}^{(1)\top}\boldsymbol{X}_{\mathbb{J}}^{(1)}\big)^{-1} \preceq \frac{12}{n}\boldsymbol{G}_{\mathbb{J}}^{-1}.
$$

*Proof.* We will give the proof for $\big(\boldsymbol{X}_{\mathbb{K}}^{(2)\top}\boldsymbol{X}_{\mathbb{K}}^{(2)}\big)^{-1}$. The proof is the same for $\big(\boldsymbol{X}_{\mathbb{J}}^{(1)\top}\boldsymbol{X}_{\mathbb{J}}^{(1)}\big)^{-1}$.

For the upper bound, consider each index $i \in \mathbb{K}$ in $\mathrm{diag}\big(\boldsymbol{X}_{\mathbb{K}}^{(2)\top}\boldsymbol{X}_{\mathbb{K}}^{(2)}\big)^{-1}$, which is $\big(\boldsymbol{X}^{(2)\top}\boldsymbol{X}^{(2)}\big)_{ii}$. Notice that according to Assumption 3, for every $i \in [d]$, $\big(\boldsymbol{X}^{(2)\top}\boldsymbol{X}^{(2)}\big)_{ii}$ follows the binomial distribution $B(n, \lambda_i)$. Also notice that for $n \geq 2$ and $1 \leq j \leq \lceil n/2 \rceil$,

$$
\begin{aligned}
\frac{1}{j}\binom{n}{j}\lambda_i^j(1-\lambda_i)^{n-j} &= \frac{1}{j} \cdot \frac{j+1}{n-j} \cdot \binom{n}{j+1} \cdot \frac{1-\lambda_i}{\lambda_i} \cdot \lambda_i^{j+1}(1-\lambda_i)^{n-j-1} \\
&= \frac{j+1}{j} \cdot \frac{n}{n-j} \cdot (1-\lambda_i) \cdot \frac{1}{n\lambda_i}\binom{n}{j+1}\lambda_i^{j+1}(1-\lambda_i)^{n-j-1} \\
&\leq 6 \cdot \frac{1}{n\lambda_i}\binom{n}{j+1}\lambda_i^{j+1}(1-\lambda_i)^{n-j-1}.
\end{aligned}
$$

Since the zero values do not count in the expectation of Moore-Penrose pseudoinverse,

$$
\begin{aligned}
\mathbb{E}\big(\boldsymbol{X}^{(2)\top}\boldsymbol{X}^{(2)}\big)_{ii}^{-1} &= \sum_{j=1}^{n} \frac{1}{j}\binom{n}{j}\lambda_i^j(1-\lambda_i)^{n-j} \\
&\leq 2\sum_{j=1}^{\lceil n/2 \rceil} \frac{1}{j}\binom{n}{j}\lambda_i^j(1-\lambda_i)^{n-j} \\
&\leq 12\sum_{j=1}^{\lceil n/2 \rceil} \frac{1}{n\lambda_i}\binom{n}{j+1}\lambda_i^{j+1}(1-\lambda_i)^{n-j-1} \\
&= \frac{12}{n\lambda_i}\sum_{j=2}^{\lceil n/2 \rceil+1}\binom{n}{j}\lambda_i^j(1-\lambda_i)^{n-j} \\
&\leq \frac{12}{n\lambda_i}.
\end{aligned}
$$

For the lower bound, consider each index $i \in \mathbb{K}$ in $\mathrm{diag}\big(\boldsymbol{X}_{\mathbb{K}}^{(2)\top}\boldsymbol{X}_{\mathbb{K}}^{(2)}\big)^{-1}$:

$$
\begin{aligned}
\mathbb{E}\big(\boldsymbol{X}^{(2)\top}\boldsymbol{X}^{(2)}\big)_{ii}^{-1} &= \mathbb{P}\big(\big(\boldsymbol{X}^{(2)\top}\boldsymbol{X}^{(2)}\big)_{ii} > 0\big)\mathbb{E}\big[\big(\boldsymbol{X}^{(2)\top}\boldsymbol{X}^{(2)}\big)_{ii}^{-1}\big|\big(\boldsymbol{X}^{(2)\top}\boldsymbol{X}^{(2)}\big)_{ii} > 0\big] \\
&\geq \mathbb{P}\big(\big(\boldsymbol{X}^{(2)\top}\boldsymbol{X}^{(2)}\big)_{ii} > 0\big)\mathbb{E}\big[\big(\boldsymbol{X}^{(2)\top}\boldsymbol{X}^{(2)}\big)_{ii}\big|\big(\boldsymbol{X}^{(2)\top}\boldsymbol{X}^{(2)}\big)_{ii} > 0\big]^{-1} \\
&= \mathbb{P}\big(\big(\boldsymbol{X}^{(2)\top}\boldsymbol{X}^{(2)}\big)_{ii} > 0\big)\cdot\big(\mathbb{E}\big[\big(\boldsymbol{X}^{(2)\top}\boldsymbol{X}^{(2)}\big)_{ii}\big]/\mathbb{P}\big(\big(\boldsymbol{X}^{(2)\top}\boldsymbol{X}^{(2)}\big)_{ii} > 0\big)\big)^{-1} \\
&= \mathbb{P}\big(\big(\boldsymbol{X}^{(2)\top}\boldsymbol{X}^{(2)}\big)_{ii} > 0\big)^2\mathbb{E}\big[\big(\boldsymbol{X}^{(2)\top}\boldsymbol{X}^{(2)}\big)_{ii}\big]^{-1} \\
&= \frac{1}{n\lambda_i}\mathbb{P}\big(\big(\boldsymbol{X}^{(2)\top}\boldsymbol{X}^{(2)}\big)_{ii} > 0\big)^2 \\
&\geq \frac{1}{4n\lambda_i}.
\end{aligned}
$$

The second line is by Jensen's inequality, and the last line is because

$$
\mathbb{P}\big(\big(\boldsymbol{X}^{(2)\top}\boldsymbol{X}^{(2)}\big)_{ii} > 0\big) = 1 - \big(1 - \lambda_i\big)^n \geq 1 - e^{-1} \geq 1/2
$$

by noticing that for all $i \leq k$, $\lambda_i > \lambda_k \geq \frac{1}{n}$. $\qquad\square$

**Lemma 13** (Bound on the tail of $\big(\boldsymbol{X}^{(t)\top}\boldsymbol{X}^{(t)}\big)^{-1}$). *Suppose Assumptions 1 and 3 hold. Denote $\mathbb{J}, \mathbb{K}$ such that $\mathbb{J} = \big\{i : \mu_i \geq \frac{1}{n}\big\}$, and $\mathbb{K} = \big\{i : \lambda_i \geq \frac{1}{n}\big\}$. Then*

$$
\frac{n}{e}\boldsymbol{G}_{\mathbb{J}^c} \preceq \mathbb{E}\big(\boldsymbol{X}_{\mathbb{J}^c}^{(1)\top}\boldsymbol{X}_{\mathbb{J}^c}^{(1)}\big)^{-1} \preceq n\boldsymbol{G}_{\mathbb{J}^c},
$$

$$
\frac{n}{e}\boldsymbol{H}_{\mathbb{K}^c} \preceq \mathbb{E}\big(\boldsymbol{X}_{\mathbb{K}^c}^{(2)\top}\boldsymbol{X}_{\mathbb{K}^c}^{(2)}\big)^{-1} \preceq n\boldsymbol{H}_{\mathbb{K}^c}.
$$

*Proof.* We will give the proof for $\big(\boldsymbol{X}_{\mathbb{K}}^{(2)\top}\boldsymbol{X}_{\mathbb{K}}^{(2)}\big)^{-1}$. The proof is the same for $\big(\boldsymbol{X}_{\mathbb{J}}^{(1)\top}\boldsymbol{X}_{\mathbb{J}}^{(1)}\big)^{-1}$. Recall that for every index $i$, $\big(\boldsymbol{X}^{(2)\top}\boldsymbol{X}^{(2)}\big)_{ii}$ follows the binomial distribution $B(n, \lambda_i)$. Therefore,

$$
\mathbb{E}\big(\boldsymbol{X}^{(2)\top}\boldsymbol{X}^{(2)}\big)_{ii}^{-1} \leq \mathbb{E}\big(\boldsymbol{X}^{(2)\top}\boldsymbol{X}^{(2)}\big)_{ii} = n\lambda_i.
$$

Also, for $i \in \mathbb{K}^c$,

$$
\begin{aligned}
\mathbb{E}\big(\boldsymbol{X}^{(2)\top}\boldsymbol{X}^{(2)}\big)_{ii}^{-1} &\geq \mathbb{E}\mathbb{1}\big[\big(\boldsymbol{X}^{(2)\top}\boldsymbol{X}^{(2)}\big)_{ii} = 1\big] \\
&= n\lambda_i(1 - \lambda_i)^{n-1} \\
&\geq \frac{n}{e}\lambda_i.
\end{aligned}
$$

The last line is by $\lambda_i \leq \frac{1}{n}$ for all $i \in \mathbb{K}^c$. $\qquad\square$

**Lemma 14** (Expectation on $\mathcal{P}^{(2)}$). *Suppose Assumptions 1 and 3 hold. Then*

$$
\begin{aligned}
\mathbb{E}\mathcal{P}^{(1)} &= (\boldsymbol{I} - \boldsymbol{G})^n, \\
\mathbb{E}\mathcal{P}^{(2)} &= (\boldsymbol{I} - \boldsymbol{H})^n.
\end{aligned}
$$

*Proof.* We will give the proof for $\mathcal{P}^{(2)}$. For any index $i$,

$$
\mathbb{E}\mathcal{P}_{ii}^{(2)} = \mathbb{E}\mathbb{1}\big[\big(\boldsymbol{X}^{(2)\top}\boldsymbol{X}^{(2)}\big)_{ii} = 0\big] = (1 - \lambda_i)^n.
$$

The same holds for $\mathcal{P}^{(1)}$. $\qquad\square$

**Lemma 15.** *Suppose Assumptions 1 and 3 hold. Then, for all index $i$ that satisfies $\lambda_i \geq 1/n, \gamma_i \neq 0$ and for $n \geq 2$,*

$$
\frac{1}{2}\left(\frac{1}{n^2(\lambda_i + \gamma_i)^2} + \frac{(1 - \lambda_i)^n}{n^2\gamma_i^2}\right) \leq \mathbb{E}\big(\boldsymbol{X}^{(2)\top}\boldsymbol{X}^{(2)} + n\boldsymbol{\Sigma}\big)_{ii}^{-2} \leq 144\left(\frac{1}{n^2(\lambda_i + \gamma_i)^2} + \frac{(1 - \lambda_i)^n}{n^2\gamma_i^2}\right).
$$

*Proof.* For the upper bound, notice that according to Assumption 3, for every $i \in [d]$, $\left(\boldsymbol{X}^{(2)\top}\boldsymbol{X}^{(2)}\right)_{ii}$ follows the binomial distribution $B(n, \lambda_i)$. Also notice that for $n \geq 2$ and $1 \leq j \leq \lceil n/2 \rceil$,

$$\binom{n}{j}\lambda_i^j(1-\lambda_i)^{n-j} \leq 3 \cdot \frac{j+1}{n\lambda_i}\binom{n}{j+1}\lambda_i^{j+1}(1-\lambda_i)^{n-j-1}.$$

Therefore, we examine the following quantity:

$$\mathbb{E}\left(\boldsymbol{X}^{(2)\top}\boldsymbol{X}^{(2)} + n\boldsymbol{\Sigma}\right)_{ii}^{-2} - \frac{2}{n^2\gamma_i^2}(1-\lambda_i)^n = (*).$$

Notice that

$$(*) \leq \sum_{j=1}^{n-1} \frac{1}{(j+n\gamma_i)^2}\binom{n}{j}\lambda_i^j(1-\lambda_i)^{n-j}$$

$$\leq 2 \sum_{j=1}^{\lceil n/2 \rceil} \frac{1}{(j+n\gamma_i)^2}\binom{n}{j}\lambda_i^j(1-\lambda_i)^{n-j}$$

$$\leq 6 \sum_{j=1}^{\lceil n/2 \rceil} \frac{1}{n\lambda_i}\frac{(j+1)}{(j+n\gamma_i)^2}\binom{n}{j+1}\lambda_i^{j+1}(1-\lambda_i)^{n-j-1}$$

$$\leq \frac{12}{n\lambda_i} \sum_{j=2}^{\lceil n/2 \rceil+1} \frac{j}{(j+n\gamma_i)^2}\binom{n}{j}\lambda_i^j(1-\lambda_i)^{n-j}.$$

Therefore,

$$(n\lambda_i + n\gamma_i) \cdot (*) \leq 12 \sum_{j=2}^{\lceil n/2 \rceil+1} \frac{j}{(j+n\gamma_i)^2}\binom{n}{j}\lambda_i^j(1-\lambda_i)^{n-j} + \sum_{j=1}^{n-1} \frac{n\gamma_i}{(j+n\gamma_i)^2}\binom{n}{j}\lambda_i^j(1-\lambda_i)^{n-j}$$

$$\leq 12 \cdot \sum_{j=1}^{n-1} \frac{1}{j+n\gamma_i}\binom{n}{j}\lambda_i^j(1-\lambda_i)^{n-j}.$$

Use the same technique and we can get

$$(n\lambda_i + n\gamma_i) \cdot \sum_{j=1}^{n-1} \frac{1}{j+n\gamma_i}\binom{n}{j}\lambda_i^j(1-\lambda_i)^{n-j} \leq 12 \cdot \sum_{j=1}^{n-1}\binom{n}{j}\lambda_i^j(1-\lambda_i)^{n-j}$$

$$\leq 12.$$

As a result, $(*) \leq \frac{144}{n^2(\lambda_i+\gamma_i)^2}$, and

$$\mathbb{E}\left(\boldsymbol{X}^{(2)\top}\boldsymbol{X}^{(2)} + n\boldsymbol{\Sigma}\right)_{ii}^{-2} \leq 144 \cdot \left(\frac{1}{n^2(\lambda_i+\gamma_i)^2} + \frac{1}{n^2\gamma_i^2}(1-\lambda_i)^n\right).$$

For the lower bound, firstly, by Jensen's inequality,

$$\mathbb{E}\left(\boldsymbol{X}^{(2)\top}\boldsymbol{X}^{(2)} + n\boldsymbol{\Sigma}\right)_{ii}^{-2} \geq \left(\mathbb{E}\boldsymbol{X}^{(2)\top}\boldsymbol{X}^{(2)} + n\boldsymbol{\Sigma}\right)_{ii}^{-2}$$

$$= (n\lambda_i + n\gamma_i)^{-2}.$$

Secondly, by only selecting the $j = 0$ term in the sum,

$$\mathbb{E}\left(\boldsymbol{X}^{(2)\top}\boldsymbol{X}^{(2)} + n\boldsymbol{\Sigma}\right)_{ii}^{-1} = \sum_{j=0}^{n} \frac{1}{(j+n\gamma_i)^2}\binom{n}{j}\lambda_i^j(1-\lambda_i)^{n-j}$$

$$\geq \frac{1}{n^2\gamma_i^2}(1-\lambda_i)^n.$$

Combine them and we have

$$\mathbb{E}\left(\boldsymbol{X}^{(2)\top}\boldsymbol{X}^{(2)} + n\boldsymbol{\Sigma}\right)_{ii}^{-2} \geq \frac{1}{2} \cdot \left(\frac{1}{n^2(\lambda_i+\gamma_i)^2} + \frac{1}{n^2\gamma_i^2}(1-\lambda_i)^n\right).$$

$\square$

**Lemma 16.** *Suppose Assumptions 1 and 3 hold. Then, for all index i that satisfies $\lambda_i \geq 1/n, \gamma_i \neq 0$ and for $n \geq 2$,*

$$\frac{1}{48} \cdot \frac{\lambda_i}{n(\lambda_i + \gamma_i)^2} \leq \mathbb{E}[\left(\boldsymbol{X}^{(2)\top}\boldsymbol{X}^{(2)} + n\boldsymbol{\Sigma}\right)^{-2}\boldsymbol{X}^{(2)\top}\boldsymbol{X}^{(2)}]_{ii} \leq 144 \cdot \frac{\lambda_i}{n(\lambda_i + \gamma_i)^2}.$$

*Proof.* For the upper bound, notice that according to Assumption 3, for every $i \in [d]$, $\left(\boldsymbol{X}^{(2)\top}\boldsymbol{X}^{(2)}\right)_{ii}$ follows the binomial distribution $B(n, \lambda_i)$. Also notice that for $n \geq 2$ and $1 \leq j \leq \lceil n/2 \rceil$,

$$\binom{n}{j}\lambda_i^j(1 - \lambda_i)^{n-j} \leq 3 \cdot \frac{j+1}{n\lambda_i}\binom{n}{j+1}\lambda_i^{j+1}(1 - \lambda_i)^{n-j-1}.$$

Therefore, we examine the following quantity:

$$\mathbb{E}[\left(\boldsymbol{X}^{(2)\top}\boldsymbol{X}^{(2)} + n\boldsymbol{\Sigma}\right)^{-2}\boldsymbol{X}^{(2)\top}\boldsymbol{X}^{(2)}]_{ii} = (*).$$

Notice that

$$(*) \leq \sum_{j=1}^{n-1}\frac{j}{(j + n\gamma_i)^2}\binom{n}{j}\lambda_i^j(1 - \lambda_i)^{n-j}$$

$$\leq 2\sum_{j=1}^{\lceil n/2 \rceil}\frac{j}{(j + n\gamma_i)^2}\binom{n}{j}\lambda_i^j(1 - \lambda_i)^{n-j}$$

$$\leq 6\sum_{j=1}^{\lceil n/2 \rceil}\frac{1}{n\lambda_i}\frac{j(j+1)}{(j + n\gamma_i)^2}\binom{n}{j+1}\lambda_i^{j+1}(1 - \lambda_i)^{n-j-1}$$

$$\leq \frac{12}{n\lambda_i}\sum_{j=2}^{\lceil n/2 \rceil+1}\frac{j^2}{(j + n\gamma_i)^2}\binom{n}{j}\lambda_i^j(1 - \lambda_i)^{n-j}.$$

Therefore,

$$(n\lambda_i + n\gamma_i) \cdot (*) \leq 12\sum_{j=2}^{\lceil n/2 \rceil+1}\frac{j^2}{(j + n\gamma_i)^2}\binom{n}{j}\lambda_i^j(1 - \lambda_i)^{n-j} + \sum_{j=1}^{n-1}\frac{jn\gamma_i}{(j + n\gamma_i)^2}\binom{n}{j}\lambda_i^j(1 - \lambda_i)^{n-j}$$

$$\leq 12 \cdot \sum_{j=1}^{n-1}\frac{j}{j + n\gamma_i}\binom{n}{j}\lambda_i^j(1 - \lambda_i)^{n-j}.$$

Use the same technique and we can get

$$(n\lambda_i + n\gamma_i) \cdot \sum_{j=1}^{n-1}\frac{j}{j + n\gamma_i}\binom{n}{j}\lambda_i^j(1 - \lambda_i)^{n-j} \leq 12 \cdot \sum_{j=1}^{n-1}j\binom{n}{j}\lambda_i^j(1 - \lambda_i)^{n-j}$$

$$\leq 12n\lambda_i.$$

As a result,

$$(*) = \mathbb{E}[\left(\boldsymbol{X}^{(2)\top}\boldsymbol{X}^{(2)} + n\boldsymbol{\Sigma}\right)^{-2}\boldsymbol{X}^{(2)\top}\boldsymbol{X}^{(2)}]_{ii} \leq 144 \cdot \frac{\lambda_i}{n(\lambda_i + \gamma_i)^2}.$$

For the lower bound, by applying Jensen's inequality on the positive values of $\big(\boldsymbol{X}^{(2)^\top}\boldsymbol{X}^{(2)} + n\boldsymbol{\Sigma}\big)^{-2}\boldsymbol{X}^{(2)^\top}\boldsymbol{X}^{(2)}$ we have:

$$
\mathbb{E}[\big(\boldsymbol{X}^{(2)^\top}\boldsymbol{X}^{(2)} + n\boldsymbol{\Sigma}\big)^{-2}\boldsymbol{X}^{(2)^\top}\boldsymbol{X}^{(2)}]_{ii}
$$

$$
\geq \mathbb{P}\big(\big(\boldsymbol{X}^{(2)^\top}\boldsymbol{X}^{(2)}\big)_{ii} > 0\big)^2 \big[\mathbb{E}\big(\boldsymbol{X}^{(2)^\top}\boldsymbol{X}^{(2)} + n\boldsymbol{\Sigma}\big)^2(\boldsymbol{X}^{(2)^\top}\boldsymbol{X}^{(2)})^{-1}]_{ii}^{-1}
$$

$$
= \mathbb{P}\big(\big(\boldsymbol{X}^{(2)^\top}\boldsymbol{X}^{(2)}\big)_{ii} > 0\big)^2 \big[n\lambda_i + 2n\gamma_i + n^2\gamma^2\mathbb{E}(\boldsymbol{X}^{(2)^\top}\boldsymbol{X}^{(2)})_{ii}^{-1}\big]^{-1}
$$

$$
\geq \mathbb{P}\big(\big(\boldsymbol{X}^{(2)^\top}\boldsymbol{X}^{(2)}\big)_{ii} > 0\big)^2 \big[n\lambda_i + 2n\gamma_i + 12n\frac{\gamma^2}{\lambda_i}\big]^{-1}
$$

$$
= \frac{\lambda_i}{12n(\lambda_i + \gamma_i)^2}\mathbb{P}\big(\big(\boldsymbol{X}^{(2)^\top}\boldsymbol{X}^{(2)}\big)_{ii} > 0\big)^2
$$

$$
\geq \frac{\lambda_i}{48n(\lambda_i + \gamma_i)^2}.
$$

The last line is because

$$
\mathbb{P}\big(\big(\boldsymbol{X}^{(2)^\top}\boldsymbol{X}^{(2)}\big)_{ii} > 0\big) = 1 - \big(1 - \lambda_i\big)^n \geq 1 - e^{-1} \geq 1/2
$$

by noticing that $\lambda_i \geq \frac{1}{n}$. $\qquad\square$

**Lemma 17.** *Suppose Assumptions 1 and 3 hold. Then, for all index $i$ that satisfies $0 < \lambda_i \leq 1/n$ and for $n \geq 2$, Then*

$$
\frac{n\lambda_i}{e(1 + n\gamma_i)^2} \leq \mathbb{E}[\big(\boldsymbol{X}^{(2)^\top}\boldsymbol{X}^{(2)} + n\boldsymbol{\Sigma}\big)^{-2}\boldsymbol{X}^{(2)^\top}\boldsymbol{X}^{(2)}]_{ii} \leq \frac{n\lambda_i}{(1 + n\gamma_i)^2}.
$$

*Proof.* Recall that for every index $i$, $\big(\boldsymbol{X}^{(2)^\top}\boldsymbol{X}^{(2)}\big)_{ii}$ follows the binomial distribution $B(n, \lambda_i)$. Therefore,

$$
\mathbb{E}[\big(\boldsymbol{X}^{(2)^\top}\boldsymbol{X}^{(2)} + n\boldsymbol{\Sigma}\big)^{-2}\boldsymbol{X}^{(2)^\top}\boldsymbol{X}^{(2)}]_{ii} \leq \frac{1}{(1 + n\gamma_i)^2}\mathbb{E}\big(\boldsymbol{X}^{(2)^\top}\boldsymbol{X}^{(2)}\big)_{ii}
$$

$$
= \frac{n\lambda_i}{(1 + n\gamma_i)^2}.
$$

Also,

$$
\mathbb{E}[\big(\boldsymbol{X}^{(2)^\top}\boldsymbol{X}^{(2)} + n\boldsymbol{\Sigma}\big)^{-2}\boldsymbol{X}^{(2)^\top}\boldsymbol{X}^{(2)}]_{ii} \geq \frac{1}{(1 + n\gamma_i)^2}\mathbb{P}\big[\big(\boldsymbol{X}^{(2)^\top}\boldsymbol{X}^{(2)}\big)_{ii} = 1\big]
$$

$$
= \frac{1}{(1 + n\gamma_i)^2}n\lambda_i(1 - \lambda_i)^{n-1}
$$

$$
\geq \frac{n\lambda_i}{e(1 + n\gamma_i)^2}.
$$

The last line is by $\lambda_i \leq \frac{1}{n}$. $\qquad\square$

Now we can give the risk in the one-hot setting in Theorem 2.

*Proof of Theorem 2.* Recall that $\mathcal{P}_{\boldsymbol{\Sigma}}^{(2)} = \big(\boldsymbol{X}^{(2)^\top}\boldsymbol{X}^{(2)} + n\boldsymbol{\Sigma}\big)^{-1}n\boldsymbol{\Sigma}$. According to (17) and (20),

$$
\mathbb{E}_\varepsilon[\mathcal{R}(\boldsymbol{w}^{(2)}) - \min\mathcal{R}] = \mathbb{E}\langle\boldsymbol{G} + \boldsymbol{H}, \; \mathbb{E}_\varepsilon(\boldsymbol{w}^{(2)} - \boldsymbol{w}^*)(\boldsymbol{w}^{(2)} - \boldsymbol{w}^*)^\top\rangle
$$

$$
= \langle\boldsymbol{G} + \boldsymbol{H}, \; \mathcal{P}_{\boldsymbol{\Sigma}}^{(2)}\mathbb{E}_\varepsilon(\boldsymbol{w}^{(1)} - \boldsymbol{w}^*)(\boldsymbol{w}^{(1)} - \boldsymbol{w}^*)^\top\mathcal{P}_{\boldsymbol{\Sigma}}^{(2)}\rangle
$$

$$
+ \sigma^2\langle\boldsymbol{G} + \boldsymbol{H}, \; \big(\boldsymbol{X}^{(2)^\top}\boldsymbol{X}^{(2)} + n\boldsymbol{\Sigma}\big)^{-2}\boldsymbol{X}^{(2)^\top}\boldsymbol{X}^{(2)}\rangle
$$

$$
= \langle\boldsymbol{G} + \boldsymbol{H}, \; \mathcal{P}_{\boldsymbol{\Sigma}}^{(2)}\mathcal{P}^{(1)}\boldsymbol{w}^*\boldsymbol{w}^{*\top}\mathcal{P}^{(1)}\mathcal{P}_{\boldsymbol{\Sigma}}^{(2)}\rangle
$$

$$
+ \sigma^2\big(\langle\boldsymbol{G} + \boldsymbol{H}, \; \mathcal{P}_{\boldsymbol{\Sigma}}^{(2)}\big(\boldsymbol{X}^{(1)^\top}\boldsymbol{X}^{(1)}\big)^{-1}\mathcal{P}_{\boldsymbol{\Sigma}}^{(2)}\rangle
$$

$$
+ \langle\boldsymbol{G} + \boldsymbol{H}, \; \big(\boldsymbol{X}^{(2)^\top}\boldsymbol{X}^{(2)} + n\boldsymbol{\Sigma}\big)^{-2}\boldsymbol{X}^{(2)^\top}\boldsymbol{X}^{(2)}\rangle\big)
$$

$$
= \texttt{bias} + \texttt{variance}.
$$

where the bias term satisfies

$$\mathbb{E}\texttt{bias} = \mathbb{E}\langle \boldsymbol{G} + \boldsymbol{H},\ \mathcal{P}_{\boldsymbol{\Sigma}}^{(2)}\mathcal{P}^{(1)}\boldsymbol{w}^*\boldsymbol{w}^{*\top}\mathcal{P}^{(1)}\mathcal{P}_{\boldsymbol{\Sigma}}^{(2)}\rangle$$
$$\approx \langle (\boldsymbol{G} + \boldsymbol{H})(\boldsymbol{I} - \boldsymbol{G})^n(\boldsymbol{\Sigma}^2(\boldsymbol{\Sigma} + \boldsymbol{H})^{-2} + (\boldsymbol{I} - \boldsymbol{H})^n),\ \boldsymbol{w}^*\boldsymbol{w}^{*\top}\rangle$$

and the variance term satisfies

$$\mathbb{E}\texttt{variance} = \sigma^2\mathbb{E}\big(\langle \boldsymbol{G} + \boldsymbol{H},\ \mathcal{P}_{\boldsymbol{\Sigma}}^{(2)}\big(\boldsymbol{X}^{(1)^\top}\boldsymbol{X}^{(1)}\big)^{-1}\mathcal{P}_{\boldsymbol{\Sigma}}^{(2)}\rangle$$
$$+ \langle \boldsymbol{G} + \boldsymbol{H},\ \big(\boldsymbol{X}^{(2)^\top}\boldsymbol{X}^{(2)} + n\boldsymbol{\Sigma}\big)^{-2}\boldsymbol{X}^{(2)^\top}\boldsymbol{X}^{(2)}\rangle\big)$$
$$\approx \sigma^2\big(\langle (\boldsymbol{G} + \boldsymbol{H})(\boldsymbol{\Sigma}^2(\boldsymbol{\Sigma} + \boldsymbol{H})^{-2} + (\boldsymbol{I} - \boldsymbol{H})^n),\ \mathbb{E}\big(\boldsymbol{X}^{(1)^\top}\boldsymbol{X}^{(1)}\big)^{-1}\rangle$$
$$+ \langle \boldsymbol{G} + \boldsymbol{H},\ \mathbb{E}\big(\boldsymbol{X}^{(2)^\top}\boldsymbol{X}^{(2)} + n\boldsymbol{\Sigma}\big)^{-2}\boldsymbol{X}^{(2)^\top}\boldsymbol{X}^{(2)}\rangle\big)$$
$$\approx \sigma^2\big(\langle (\boldsymbol{G} + \boldsymbol{H})(\boldsymbol{\Sigma}^2(\boldsymbol{\Sigma} + \boldsymbol{H})^{-2} + (\boldsymbol{I} - \boldsymbol{H})^n),\ \frac{1}{n}\boldsymbol{G}_{\mathbb{J}}^{-1} + n\boldsymbol{G}_{\mathbb{J}^c}\rangle$$
$$+ \langle \boldsymbol{G} + \boldsymbol{H},\ \frac{1}{n}(\boldsymbol{H}_{\mathbb{K}} + \boldsymbol{\Sigma}_{\mathbb{K}})^{-2}\boldsymbol{H}_{\mathbb{K}} + n(\boldsymbol{I}_{\mathbb{K}^c} + n\boldsymbol{\Sigma}_{\mathbb{K}^c})^{-2}\boldsymbol{H}_{\mathbb{K}^c}\rangle\big).$$

We have finished the proof. □

### B.3 PROOF OF PROPOSITION 1

*Proof of Proposition 1.* Define the joint data and the joint label noise by

$$\boldsymbol{X} := \begin{pmatrix} \boldsymbol{X}^{(1)} \\ \boldsymbol{X}^{(2)} \end{pmatrix}, \quad \boldsymbol{y} := \begin{pmatrix} \boldsymbol{y}^{(1)} \\ \boldsymbol{y}^{(2)} \end{pmatrix} \in \mathbb{R}^{2n}, \quad \boldsymbol{\varepsilon} := \begin{pmatrix} \boldsymbol{\varepsilon}^{(1)} \\ \boldsymbol{\varepsilon}^{(2)} \end{pmatrix} \in \mathbb{R}^{2n},$$

then

$$\mathbb{E}\boldsymbol{\varepsilon} = 0, \quad \mathbb{E}\boldsymbol{\varepsilon}\boldsymbol{\varepsilon}^\top = \sigma^2\boldsymbol{I}.$$

By the definition of (4), we obtain

$$\boldsymbol{w}_{\texttt{joint}} = \big(\boldsymbol{X}^\top\boldsymbol{X}\big)^{-1}\boldsymbol{X}^\top\boldsymbol{y}$$
$$= \big(\boldsymbol{X}^\top\boldsymbol{X}\big)^{-1}\boldsymbol{X}^\top\boldsymbol{X}\boldsymbol{w}^* + \big(\boldsymbol{X}^\top\boldsymbol{X}\big)^{-1}\boldsymbol{X}^\top\boldsymbol{\varepsilon},$$

which implies that

$$\boldsymbol{w}_{\texttt{joint}} - \boldsymbol{w}^* = \big(\big(\boldsymbol{X}^\top\boldsymbol{X}\big)^{-1}\boldsymbol{X}^\top\boldsymbol{X} - \boldsymbol{I}\big)\cdot\boldsymbol{w}^* + \big(\boldsymbol{X}^\top\boldsymbol{X}\big)^{-1}\boldsymbol{X}^\top\boldsymbol{\varepsilon}.$$

Denote $\mathcal{P}_{\texttt{joint}} = \boldsymbol{I} - \big(\boldsymbol{X}^\top\boldsymbol{X}\big)^{-1}\boldsymbol{X}^\top\boldsymbol{X}$. Therefore, it holds that

$$\mathbb{E}_\varepsilon(\boldsymbol{w}_{\texttt{joint}} - \boldsymbol{w}^*)(\boldsymbol{w}_{\texttt{joint}} - \boldsymbol{w}^*)^\top = \mathcal{P}_{\texttt{joint}}\cdot\boldsymbol{w}^*\boldsymbol{w}^{*\top}\cdot\mathcal{P}_{\texttt{joint}} + \sigma^2\cdot\big(\boldsymbol{X}^\top\boldsymbol{X}\big)^{-1}. \quad (21)$$

Moreover, the total risk can be reformulated to

$$\mathcal{R}(\boldsymbol{w}) - \min\mathcal{R} = \mathcal{R}_1(\boldsymbol{w}) - \min\mathcal{R}_1 + \mathcal{R}_2(\boldsymbol{w}) - \min\mathcal{R}_2$$
$$= \langle \boldsymbol{G},\ (\boldsymbol{w} - \boldsymbol{w}^*)(\boldsymbol{w} - \boldsymbol{w}^*)^\top\rangle + \langle \boldsymbol{H},\ (\boldsymbol{w} - \boldsymbol{w}^*)(\boldsymbol{w} - \boldsymbol{w}^*)^\top\rangle$$
$$= \langle \boldsymbol{G} + \boldsymbol{H},\ (\boldsymbol{w} - \boldsymbol{w}^*)(\boldsymbol{w} - \boldsymbol{w}^*)^\top\rangle.$$

Bringing (21) into the above, we obtain

$$\mathbb{E}_\varepsilon\mathcal{R}(\boldsymbol{w}_{\texttt{joint}}) - \min\mathcal{R} = \langle \boldsymbol{G} + \boldsymbol{H},\ \mathbb{E}_\varepsilon(\boldsymbol{w}_{\texttt{joint}} - \boldsymbol{w}^*)(\boldsymbol{w}_{\texttt{joint}} - \boldsymbol{w}^*)^\top\rangle$$
$$= \langle \boldsymbol{G} + \boldsymbol{H},\ \mathcal{P}_{\texttt{joint}}\cdot\boldsymbol{w}^*\boldsymbol{w}^{*\top}\cdot\mathcal{P}_{\texttt{joint}}\rangle + \sigma^2\cdot\langle \boldsymbol{G} + \boldsymbol{H},\ \big(\boldsymbol{X}^\top\boldsymbol{X}\big)^{-1}\rangle$$
$$= \texttt{bias} + \texttt{variance}.$$

For the bias part, notice that $\mathcal{P}_{\texttt{joint}} = \mathcal{P}_1\mathcal{P}_2$ in the one-hot setting. Therefore,

$$\mathbb{E}\texttt{bias} = \mathbb{E}\langle \boldsymbol{G} + \boldsymbol{H},\ \mathcal{P}_{\texttt{joint}}\cdot\boldsymbol{w}^*\boldsymbol{w}^{*\top}\cdot\mathcal{P}_{\texttt{joint}}\rangle +$$
$$= \langle (\boldsymbol{G} + \boldsymbol{H})(\boldsymbol{I} - \boldsymbol{G})^n(\boldsymbol{I} - \boldsymbol{H})^n,\ \boldsymbol{w}^*\boldsymbol{w}^{*\top}\rangle$$
$$= \sum_i (\mu_i + \lambda_i)(1 - \mu_i)^n(1 - \lambda_i)^n w_i^{*2}.$$

For the variance part, notice that $(\boldsymbol{X}^\top \boldsymbol{X})_{ii} = (\boldsymbol{X^{(1)}}^\top \boldsymbol{X^{(1)}})_{ii} + (\boldsymbol{X^{(2)}}^\top \boldsymbol{X^{(2)}})_{ii}$ for every index $i$, where $(\boldsymbol{X^{(1)}}^\top \boldsymbol{X^{(1)}})_{ii}$ follows the binomial distribution $B(n, \mu_i)$, and $(\boldsymbol{X^{(2)}}^\top \boldsymbol{X^{(2)}})_{ii}$ follows the binomial distribution $B(n, \lambda_i)$. As such, consider a random variable $s_i$, which is the sum of $n$ independent trials, where each trial returns 1 iff at least one trial of the binomials $(\boldsymbol{X^{(1)}}^\top \boldsymbol{X^{(1)}})_{ii}$ and $(\boldsymbol{X^{(2)}}^\top \boldsymbol{X^{(2)}})_{ii}$ return 1. Then for every index $i$, $(\boldsymbol{X}^\top \boldsymbol{X})_{ii}$ satisfies

$$s_i \leq (\boldsymbol{X}^\top \boldsymbol{X})_{ii} \leq 2s_i,$$

and $s_i$ follows the binomial distribution $B(n, \mu_i + \lambda_i - \mu_i \lambda_i)$. Therefore, according to Lemma 12, for each index $i$ that satisfies $\mu_i + \lambda_i - \mu_i \lambda_i \geq \frac{c}{n}$ for constant $c > 0$,

$$\frac{1}{n}\frac{1}{\mu_i + \lambda_i - \mu_i \lambda_i} \lesssim \frac{1}{2}\mathbb{E}s_i^{-1} \leq \mathbb{E}(\boldsymbol{X}^\top \boldsymbol{X})_{ii}^{-1} \leq \mathbb{E}s_i^{-1} \lesssim \frac{1}{n}\frac{1}{\mu_i + \lambda_i - \mu_i \lambda_i}.$$

Also, according to Lemma 13, for each index $i$ that satisfies $\mu_i + \lambda_i - \mu_i \lambda_i \leq \frac{c}{n}$ for constant $c > 0$,

$$n(\mu_i + \lambda_i - \mu_i \lambda_i) \lesssim \frac{1}{2}\mathbb{E}s_i^{-1} \leq \mathbb{E}(\boldsymbol{X}^\top \boldsymbol{X})_{ii}^{-1} \leq \mathbb{E}s_i^{-1} \lesssim n(\mu_i + \lambda_i - \mu_i \lambda_i).$$

Notice that for the index sets $\mathbb{J} = \left\{i : \mu_i \geq \frac{1}{n}\right\}$, and $\mathbb{K} = \left\{i : \lambda_i \geq \frac{1}{n}\right\}$, for $i \in \mathbb{J} \cup \mathbb{K}$, $\mu_i + \lambda_i - \mu_i \lambda_i \geq \frac{1}{n}$; also, for $i \in \mathbb{J}^c \cap \mathbb{K}^c$, $\mu_i + \lambda_i - \mu_i \lambda_i \leq \frac{2}{n}$. Also notice that

$$0 \leq \mu_i \lambda_i \leq \frac{1}{4}(\mu_i + \lambda_i)^2 \leq \frac{1}{2}(\mu_i + \lambda_i).$$

Therefore,

$$\frac{1}{n}\frac{1}{\mu_i + \lambda_i} \lesssim \mathbb{E}(\boldsymbol{X}^\top \boldsymbol{X})_{ii}^{-1} \lesssim \frac{1}{n}\frac{1}{\mu_i + \lambda_i}, \quad i \in \mathbb{J} \cup \mathbb{K};$$

$$n(\mu_i + \lambda_i) \lesssim \mathbb{E}(\boldsymbol{X}^\top \boldsymbol{X})_{ii}^{-1} \lesssim n(\mu_i + \lambda_i), \quad i \in \mathbb{J}^c \cap \mathbb{K}^c.$$

As a result,

$$\mathbb{E}(\boldsymbol{X}^\top \boldsymbol{X})^{-1} \asymp \frac{1}{n}(\boldsymbol{G} + \boldsymbol{H})_{\mathbb{J} \cup \mathbb{K}}^{-1} + n(\boldsymbol{G} + \boldsymbol{H})_{\mathbb{J}^c \cap \mathbb{K}^c}.$$

Now we can bound the variance by

$$\mathbb{E}\texttt{variance} = \sigma^2 \cdot \langle \boldsymbol{G} + \boldsymbol{H}, \; \mathbb{E}(\boldsymbol{X}^\top \boldsymbol{X})^{-1} \rangle$$

$$\asymp \sigma^2 \cdot \langle \boldsymbol{G} + \boldsymbol{H}, \; \frac{1}{n}(\boldsymbol{G} + \boldsymbol{H})_{\mathbb{J} \cup \mathbb{K}}^{-1} + n(\boldsymbol{G} + \boldsymbol{H})_{\mathbb{J}^c \cap \mathbb{K}^c} \rangle$$

$$= \frac{\sigma^2}{n}\left(|\mathbb{J} \cup \mathbb{K}| + n^2 \sum_i (\mu_i + \lambda_i)^2\right).$$

We have finished the proof. $\qquad\square$

### B.4 Proof of Corollary 3

*Proof of Corollary 3.* Recall from Theorem 2 that

$$\mathbb{E}\Delta(\boldsymbol{w}^{(2)}) = \texttt{bias} + \texttt{variance},$$

where

$$\texttt{bias} \asymp \langle (\boldsymbol{G} + \boldsymbol{H})(\boldsymbol{I} - \boldsymbol{G})^n (\boldsymbol{I} - \boldsymbol{H})^n, \; \boldsymbol{w}^* \boldsymbol{w}^{*\top} \rangle$$

$$= \sum_i (1 - \mu_i)^n (1 - \lambda_i)^n (\mu_i + \lambda_i) w_i^{*2},$$

$$\texttt{variance} \asymp \sigma^2 \left( \langle (\boldsymbol{G} + \boldsymbol{H})(\boldsymbol{I} - \boldsymbol{H})^n, \frac{1}{n}\boldsymbol{G}_{\mathbb{J}}^{-1} + n\boldsymbol{G}_{\mathbb{J}^c} \rangle + \langle \boldsymbol{G} + \boldsymbol{H}, \frac{1}{n}\boldsymbol{H}_{\mathbb{K}}^{-1} + n\boldsymbol{H}_{\mathbb{K}^c} \rangle \right)$$

$$\lesssim \frac{\sigma^2}{n}\left( \sum_{i \in \mathbb{K}} \frac{\mu_i}{\lambda_i} + n^2 \sum_{i \in \mathbb{K}^c} \mu_i \lambda_i + \sum_{i \in \mathbb{J}} (1 - \lambda_i)^n + n^2 \sum_{i \in \mathbb{J}^c} (1 - \lambda_i)^n \mu_i^2 \right.$$

$$\left. + |\mathbb{K}| + n^2 \sum_{i \in \mathbb{K}^c} \lambda_i^2 + \sum_{i \in \mathbb{J}} \lambda_i (1 - \lambda_i)^n \mu_i^{-1} + n^2 \sum_{i \in \mathbb{J}^c} \lambda_i (1 - \lambda_i)^n \mu_i \right)$$

Also recall that the joint learning parameter $\boldsymbol{w}_{\text{joint}}$ satisfy

$$\mathbb{E}\Delta(\boldsymbol{w}_{\text{joint}}) = \texttt{bias}_{\text{joint}} + \texttt{variance}_{\text{joint}},$$

where

$$\texttt{bias}_{\text{joint}} = \sum_i (1 - \mu_i)^n (1 - \lambda_i)^n (\mu_i + \lambda_i) w_i^{*2},$$

$$\texttt{variance}_{\text{joint}} \asymp \frac{\sigma^2}{n}\Big(|\mathbb{J} \cup \mathbb{K}| + n^2 \sum_{i \in \mathbb{J}^c \cap \mathbb{K}^c} (\mu_i + \lambda_i)^2\Big).$$

Notice that:

- The bias matches the joint learning bias;

- $\frac{\sigma^2}{n}\big(|\mathbb{K}| + n^2 \sum_{i \in \mathbb{K}^c} \lambda_i^2\big)$ is bounded by the joint learning variance;

- And that

$$\frac{\sigma^2}{n}\bigg( \sum_{i \in \mathbb{J}} \lambda_i (1 - \lambda_i)^n \mu_i^{-1} + n^2 \sum_{i \in \mathbb{J}^c} \lambda_i \big(1 - \lambda_i\big)^n \mu_i \bigg)$$

$$\leq \sigma^2 \bigg( \sum_i \lambda_i^2 (1 - \lambda_i)^{2n} \bigg)^{1/2} \bigg( \sum_{i \in \mathbb{J}} n^{-2} \mu_i^{-2} + \sum_{i \in \mathbb{J}^c} n^2 \mu_i^2 \bigg)^{1/2}$$

$$\leq \sigma^2 \bigg( \frac{e^2}{n^2} |\mathbb{K}| + \sum_{i \in \mathbb{K}^c} \lambda_i^2 \bigg)^{1/2} \bigg( |\mathbb{J}| + \sum_{i \in \mathbb{J}^c} n^2 \mu_i^2 \bigg)^{1/2}$$

is also bounded by the joint learning variance.

As a result,

$$\texttt{variance} \lesssim \texttt{variance}_{\text{joint}} + \sum_{i \in \mathbb{K}} \frac{\mu_i}{\lambda_i} + n^2 \sum_{i \in \mathbb{K}^c} \mu_i \lambda_i + \sum_{i \in \mathbb{J}} \big(1 - \lambda_i\big)^n + n^2 \sum_{i \in \mathbb{J}^c} \big(1 - \lambda_i\big)^n \mu_i^2$$

$$\leq \texttt{variance}_{\text{joint}} + \sum_{i \in \mathbb{K}} \frac{\mu_i}{\lambda_i} + n^2 \sum_{i \in \mathbb{J} \cap \mathbb{K}^c} \mu_i \lambda_i + n^2 \sum_{i \in \mathbb{J}^c \cap \mathbb{K}^c} \mu_i \lambda_i + |\mathbb{J}| + n^2 \sum_{i \in \mathbb{J}^c} \mu_i^2$$

$$\leq \texttt{variance}_{\text{joint}} + \sum_{i \in \mathbb{K}} \frac{\mu_i}{\lambda_i} + n^2 \sum_{i \in \mathbb{J} \cap \mathbb{K}^c} \mu_i \lambda_i + |\mathbb{J}| + 2n^2 \sum_{i \in \mathbb{J}^c} \mu_i^2 + n^2 \sum_{i \in \mathbb{K}^c} \lambda_i^2$$

$$\lesssim \texttt{variance}_{\text{joint}} + \sum_{i \in \mathbb{K}} \frac{\mu_i}{\lambda_i} + n^2 \sum_{i \in \mathbb{J} \cap \mathbb{K}^c} \mu_i \lambda_i.$$

Thus we finish the proof. $\qquad \square$

### B.5 PROOF OF COROLLARY 4

*Proof of Corollary 4.* Recall from Theorem 2 that

$$\mathbb{E}\Delta(\boldsymbol{w}^{(2)}) = \texttt{bias} + \texttt{variance},$$

where the bias satisfies

$$\texttt{bias} \asymp \langle (\boldsymbol{G} + \boldsymbol{H})(\boldsymbol{I} - \boldsymbol{G})^n (\boldsymbol{\Sigma}^2 (\boldsymbol{\Sigma} + \boldsymbol{H})^{-2} + (\boldsymbol{I} - \boldsymbol{H})^n), \ \boldsymbol{w}^* \boldsymbol{w}^{*\top} \rangle$$

$$= \texttt{bias}_{\text{joint}} + \langle (\boldsymbol{G} + \boldsymbol{H})(\boldsymbol{I} - \boldsymbol{G})^n \boldsymbol{\Sigma}^2 (\boldsymbol{\Sigma} + \boldsymbol{H})^{-2}, \ \boldsymbol{w}^* \boldsymbol{w}^{*\top} \rangle,$$

$$\lesssim \texttt{bias}_{\text{joint}} + \frac{1}{n}\langle \boldsymbol{\Sigma}^2 (\boldsymbol{\Sigma} + \boldsymbol{H})^{-2}, \ \boldsymbol{w}^* \boldsymbol{w}^{*\top} \rangle + \langle \boldsymbol{\Sigma}, \ \boldsymbol{w}^* \boldsymbol{w}^{*\top} \rangle,$$

$$\lesssim \texttt{bias}_{\text{joint}} + (\gamma + \tfrac{1}{n})\|\boldsymbol{w}^*\|_2^2.$$

For the variance,

$$\frac{1}{\sigma^2}\texttt{variance} \lesssim \langle (\boldsymbol{G} + \boldsymbol{H})(\boldsymbol{\Sigma}^2(\boldsymbol{\Sigma} + \boldsymbol{H})^{-2} + (\boldsymbol{I} - \boldsymbol{H})^n), \frac{1}{n}\boldsymbol{G}_{\mathbb{J}}^{-1} + n\boldsymbol{G}_{\mathbb{J}^c} \rangle$$

$$+ \langle \boldsymbol{G} + \boldsymbol{H}, \frac{1}{n}(\boldsymbol{H}_{\mathbb{K}} + \boldsymbol{\Sigma}_{\mathbb{K}})^{-2}\boldsymbol{H}_{\mathbb{K}} + n(\boldsymbol{I}_{\mathbb{K}^c} + n\boldsymbol{\Sigma}_{\mathbb{K}^c})^{-2}\boldsymbol{H}_{\mathbb{K}^c} \rangle$$

$$\lesssim \langle (\boldsymbol{G} + \boldsymbol{H})(\boldsymbol{I} - \boldsymbol{H})^n, \frac{1}{n}\boldsymbol{G}_{\mathbb{J}}^{-1} + n\boldsymbol{G}_{\mathbb{J}^c} \rangle + \langle \boldsymbol{G}, \frac{1}{n}\boldsymbol{G}_{\mathbb{J}}^{-1} + n\boldsymbol{G}_{\mathbb{J}^c} \rangle + \langle \boldsymbol{H}, \frac{1}{n}\boldsymbol{H}_{\mathbb{K}}^{-1} + n\boldsymbol{H}_{\mathbb{K}^c} \rangle$$

$$+ \langle \boldsymbol{H}\boldsymbol{\Sigma}^2(\boldsymbol{\Sigma} + \boldsymbol{H})^{-2}, \frac{1}{n}\boldsymbol{G}_{\mathbb{J}}^{-1} + n\boldsymbol{G}_{\mathbb{J}^c} \rangle + \langle \boldsymbol{G}, \frac{1}{n}(\boldsymbol{H}_{\mathbb{K}} + \boldsymbol{\Sigma}_{\mathbb{K}})^{-2}\boldsymbol{H}_{\mathbb{K}} + n(\boldsymbol{I}_{\mathbb{K}^c} + n\boldsymbol{\Sigma}_{\mathbb{K}^c})^{-2}\boldsymbol{H}_{\mathbb{K}^c} \rangle$$

$$\lesssim \frac{1}{\sigma^2}\texttt{variance}_{\texttt{joint}} + \langle \boldsymbol{\Sigma}, \frac{1}{n}\boldsymbol{G}_{\mathbb{J}}^{-1} + n\boldsymbol{G}_{\mathbb{J}^c} \rangle + \langle \boldsymbol{G}, \frac{1}{n}(\boldsymbol{H} + \boldsymbol{\Sigma} + \frac{1}{n}\boldsymbol{I})^{-2}\boldsymbol{H} \rangle$$

$$\lesssim \frac{1}{\sigma^2}\texttt{variance}_{\texttt{joint}} + \frac{1}{n}\sum_{i \in \mathbb{J} \cup \mathbb{K}} \left( \frac{\mu_i}{\lambda_i + \frac{1}{n} + \gamma} + \frac{\gamma}{\mu_i + \frac{1}{n}} \right).$$

We know from Corollary 3 that $\sigma^2 \langle (\boldsymbol{G} + \boldsymbol{H})(\boldsymbol{I} - \boldsymbol{H})^n, \frac{1}{n}\boldsymbol{G}_{\mathbb{J}}^{-1} + n\boldsymbol{G}_{\mathbb{J}^c} \rangle$ is bounded by the joint learning variance. The tail terms are also bounded by the joint learning variance. Thus we finish the proof. $\qquad \square$

### B.6 PROOF OF COROLLARY 6

*Proof of Corollary 6.* Recall that

$$\mathbb{E}\Delta(\boldsymbol{w}^{(2)}) = \texttt{bias} + \texttt{variance},$$

where bias and variance with respect $\boldsymbol{x}^{(1)}$ satisfy

$$\texttt{bias} \asymp \langle (\boldsymbol{G} + \boldsymbol{H})(\boldsymbol{I} - \boldsymbol{G})^n(\boldsymbol{\Sigma}^2(\boldsymbol{\Sigma} + \boldsymbol{H})^{-2} + (\boldsymbol{I} - \boldsymbol{H})^n), \boldsymbol{w}^* \boldsymbol{w}^{*\top} \rangle$$

$$= \sum_i (1 - \mu_i)^n \left( \frac{\gamma_i^2}{(\lambda_i + \gamma_i)^2} + (1 - \lambda_i)^n \right)(\mu_i + \lambda_i)w_i^{*2},$$

$$\texttt{variance} \lesssim \sigma^2 \big( \langle (\boldsymbol{G} + \boldsymbol{H})(\boldsymbol{\Sigma}^2(\boldsymbol{\Sigma} + \boldsymbol{H})^{-2} + (\boldsymbol{I} - \boldsymbol{H})^n), \frac{1}{n}\boldsymbol{G}_{\mathbb{J}}^{-1} + n\boldsymbol{G}_{\mathbb{J}^c} \rangle$$

$$+ \langle \boldsymbol{G} + \boldsymbol{H}, \frac{1}{n}(\boldsymbol{H}_{\mathbb{K}} + \boldsymbol{\Sigma}_{\mathbb{K}})^{-2}\boldsymbol{H}_{\mathbb{K}} + n(\boldsymbol{I}_{\mathbb{K}^c} + n\boldsymbol{\Sigma}_{\mathbb{K}^c})^{-2}\boldsymbol{H}_{\mathbb{K}^c} \rangle \big).$$

For the bias, notice that it satisfies that the quantity $\sum_i (1 - \mu_i)^n (1 - \lambda_i)^n (\mu_i + \lambda_i)w_i^{*2}$ is the joint learning bias. For the remaining part, when $\gamma_i = \mu_i$ for $\mu_i \geq 1/n$ and $\gamma_i = 0$ otherwise, we have

$$\sum_i (1 - \mu_i)^n \frac{\gamma_i^2}{(\lambda_i + \gamma_i)^2}(\mu_i + \lambda_i)w_i^{*2} = \sum_{i \in \mathbb{J}} (1 - \mu_i)^n \frac{\mu_i^2}{(\lambda_i + \mu_i)^2}(\mu_i + \lambda_i)w_i^{*2}$$

$$\leq \sum_{i \in \mathbb{J}} (1 - \mu_i)^n \mu_i w_i^{*2}$$

$$\leq \sum_{i \in \mathbb{J}} (1 - \mu_i - \lambda_i)^n (\mu_i + \lambda_i)w_i^{*2}$$

$$\leq \sum_{i \in \mathbb{J}} (1 - \mu_i)^n (1 - \lambda_i)^n (\mu_i + \lambda_i)w_i^{*2} \leq \texttt{bias}_{\texttt{joint}}.$$

The second last line is because the function $x(1 - x)^n$ decreases when $x \geq 1/n$. As a result, the whole bias is bounded by the joint learning bias.

For the variance,

$$\frac{1}{\sigma^2}\texttt{variance}$$

$$\lesssim \left(\langle (\boldsymbol{G}+\boldsymbol{H})(\boldsymbol{\Sigma}^2(\boldsymbol{\Sigma}+\boldsymbol{H})^{-2} + (\boldsymbol{I}-\boldsymbol{H})^n), \frac{1}{n}\boldsymbol{G}_{\mathbb{J}}^{-1} + n\boldsymbol{G}_{\mathbb{J}^c}\rangle\right.$$

$$\left.+ \langle \boldsymbol{G}+\boldsymbol{H}, \frac{1}{n}(\boldsymbol{H}_{\mathbb{K}}+\boldsymbol{\Sigma}_{\mathbb{K}})^{-2}\boldsymbol{H}_{\mathbb{K}} + n(\boldsymbol{I}_{\mathbb{K}^c}+n\boldsymbol{\Sigma}_{\mathbb{K}^c})^{-2}\boldsymbol{H}_{\mathbb{K}^c}\rangle\right)$$

$$= \langle (\boldsymbol{G}+\boldsymbol{H})(\boldsymbol{I}-\boldsymbol{H})^n, \frac{1}{n}\boldsymbol{G}_{\mathbb{J}}^{-1} + n\boldsymbol{G}_{\mathbb{J}^c}\rangle$$

$$+ \sum_{i\in\mathbb{J}\cap\mathbb{K}}(\mu_i+\lambda_i)\left(\frac{1}{n\mu_i}\frac{\gamma_i^2}{(\gamma_i+\lambda_i)^2} + \frac{\lambda_i}{n(\gamma_i+\lambda_i)^2}\right) + \sum_{i\in\mathbb{J}\cap\mathbb{K}^c}(\mu_i+\lambda_i)\left(\frac{1}{n\mu_i}\frac{\gamma_i^2}{(\gamma_i+\lambda_i)^2} + \frac{\lambda_i}{n(\gamma_i+1/n)^2}\right)$$

$$+ \sum_{i\in\mathbb{J}^c\cap\mathbb{K}}(\mu_i+\lambda_i)\left(n\mu_i\frac{\gamma_i^2}{(\gamma_i+\lambda_i)^2} + \frac{\lambda_i}{n(\gamma_i+\lambda_i)^2}\right) + \sum_{i\in\mathbb{J}^c\cap\mathbb{K}^c}(\mu_i+\lambda_i)\left(n\mu_i\frac{\gamma_i^2}{(\gamma_i+\lambda_i)^2} + \frac{\lambda_i}{n(\gamma_i+1/n)^2}\right)$$

We know from Corollary 3 that $\sigma^2\langle (\boldsymbol{G}+\boldsymbol{H})(\boldsymbol{I}-\boldsymbol{H})^n, \frac{1}{n}\boldsymbol{G}_{\mathbb{J}}^{-1} + n\boldsymbol{G}_{\mathbb{J}^c}\rangle$ is bounded by the joint learning variance. Thus the quantity of our focus is

$$(*) = \sum_{i\in\mathbb{J}\cap\mathbb{K}}(\mu_i+\lambda_i)\left(\frac{1}{n\mu_i}\frac{\gamma_i^2}{(\gamma_i+\lambda_i)^2} + \frac{\lambda_i}{n(\gamma_i+\lambda_i)^2}\right) + \sum_{i\in\mathbb{J}\cap\mathbb{K}^c}(\mu_i+\lambda_i)\left(\frac{1}{n\mu_i}\frac{\gamma_i^2}{(\gamma_i+\lambda_i)^2} + \frac{\lambda_i}{n(\gamma_i+1/n)^2}\right)$$

$$+ \sum_{i\in\mathbb{J}^c\cap\mathbb{K}}(\mu_i+\lambda_i)\left(n\mu_i\frac{\gamma_i^2}{(\gamma_i+\lambda_i)^2} + \frac{\lambda_i}{n(\gamma_i+\lambda_i)^2}\right) + \sum_{i\in\mathbb{J}^c\cap\mathbb{K}^c}(\mu_i+\lambda_i)\left(n\mu_i\frac{\gamma_i^2}{(\gamma_i+\lambda_i)^2} + \frac{\lambda_i}{n(\gamma_i+1/n)^2}\right).$$

When $\gamma_i = \mu_i$ for $\mu_i \geq 1/n$ and $\gamma_i = 0$ otherwise, the above quantity satisfies

$$(*) \leq \sum_{i\in\mathbb{J}\cap\mathbb{K}}(\mu_i+\lambda_i)\left(\frac{\mu_i^2}{n(\mu_i+\lambda_i)^2} + \frac{\lambda_i}{n(\mu_i+\lambda_i)^2}\right) + \sum_{i\in\mathbb{J}\cap\mathbb{K}^c}(\mu_i+\lambda_i)\left(\frac{\mu_i^2}{n(\mu_i+\lambda_i)^2} + \frac{\lambda_i}{n(\mu_i+\lambda_i)^2}\right)$$

$$+ \sum_{i\in\mathbb{J}^c\cap\mathbb{K}}(\mu_i+\lambda_i)\frac{1}{n\lambda_i} + \sum_{i\in\mathbb{J}^c\cap\mathbb{K}^c}(\mu_i+\lambda_i)\cdot n\lambda_i$$

$$\leq \sum_{i\in\mathbb{J}\cap\mathbb{K}}\frac{1}{n} + \sum_{i\in\mathbb{J}\cap\mathbb{K}^c}\frac{1}{n} + \sum_{i\in\mathbb{J}^c\cap\mathbb{K}}\frac{2}{n} + \sum_{i\in\mathbb{J}^c\cap\mathbb{K}^c}(n\mu_i\lambda_i + n\lambda_i^2) \lesssim \frac{1}{\sigma^2}\texttt{variance}_{\texttt{joint}}.$$

$$(22)$$

The last line holds because of Cauchy-Schwarz inequality. Thus we finish the proof. □

### B.7 PROOF OF COROLLARY 8

*Proof of Corollary 8.* Again, recall that

$$\mathbb{E}\Delta(\boldsymbol{w}^{(2)}) = \texttt{bias} + \texttt{variance},$$

where bias and variance with respect $\boldsymbol{x}^{(1)}$ satisfy

$$\texttt{bias} \asymp \langle (\boldsymbol{G}+\boldsymbol{H})(\boldsymbol{I}-\boldsymbol{G})^n(\boldsymbol{\Sigma}^2(\boldsymbol{\Sigma}+\boldsymbol{H})^{-2} + (\boldsymbol{I}-\boldsymbol{H})^n), \boldsymbol{w}^*\boldsymbol{w}^{*\top}\rangle$$

$$= \sum_i (1-\mu_i)^n\left(\frac{\gamma_i^2}{(\lambda_i+\gamma_i)^2} + (1-\lambda_i)^n\right)(\mu_i+\lambda_i)w_i^{*2},$$

$$\texttt{variance} \lesssim \sigma^2\left(\langle (\boldsymbol{G}+\boldsymbol{H})(\boldsymbol{\Sigma}^2(\boldsymbol{\Sigma}+\boldsymbol{H})^{-2} + (\boldsymbol{I}-\boldsymbol{H})^n), \frac{1}{n}\boldsymbol{G}_{\mathbb{J}}^{-1} + n\boldsymbol{G}_{\mathbb{J}^c}\rangle\right.$$

$$\left.+ \langle \boldsymbol{G}+\boldsymbol{H}, \frac{1}{n}(\boldsymbol{H}_{\mathbb{K}}+\boldsymbol{\Sigma}_{\mathbb{K}})^{-2}\boldsymbol{H}_{\mathbb{K}} + n(\boldsymbol{I}_{\mathbb{K}^c}+n\boldsymbol{\Sigma}_{\mathbb{K}^c})^{-2}\boldsymbol{H}_{\mathbb{K}^c}\rangle\right).$$

For the bias, notice that the quantity $\sum_i (1 - \mu_i)^n (1 - \lambda_i)^n (\mu_i + \lambda_i) w_i^{*2}$ is the joint learning bias. For the remaining part, when $\gamma_i = \mu_i$ for $i \leq k$ and $\gamma_i = 0$ otherwise, we have

$$
\begin{aligned}
\sum_i (1 - \mu_i)^n \frac{\gamma_i^2}{(\lambda_i + \gamma_i)^2} (\mu_i + \lambda_i) w_i^{*2} &= \sum_{i \leq k} (1 - \mu_i)^n \frac{\mu_i^2}{(\lambda_i + \mu_i)^2} (\mu_i + \lambda_i) w_i^{*2} \\
&\leq \sum_{i \leq k} (1 - \mu_i)^n \mu_i w_i^{*2} \\
&\leq \sum_{i \leq k} (1 - \mu_i - \lambda_i)^n (\mu_i + \lambda_i) w_i^{*2} \\
&\leq \sum_{i \leq k} (1 - \mu_i)^n (1 - \lambda_i)^n (\mu_i + \lambda_i) w_i^{*2} \leq \texttt{bias}_{\texttt{joint}}.
\end{aligned}
$$

The second last line is because the function $x(1-x)^n$ decreases when $x \geq 1/n$. As a result, the whole bias is bounded by the joint learning bias.

For the variance, similarly to the proof of Corollary 6, the quantity of our focus is

$$
\begin{aligned}
(*) = &\sum_{i \in \mathbb{J} \cap \mathbb{K}} (\mu_i + \lambda_i) \left( \frac{1}{n\mu_i} \frac{\gamma_i^2}{(\gamma_i + \lambda_i)^2} + \frac{\lambda_i}{n(\gamma_i + \lambda_i)^2} \right) + \sum_{i \in \mathbb{J} \cap \mathbb{K}^c} (\mu_i + \lambda_i) \left( \frac{1}{n\mu_i} \frac{\gamma_i^2}{(\gamma_i + \lambda_i)^2} + \frac{\lambda_i}{n(\gamma_i + 1/n)^2} \right) \\
&+ \sum_{i \in \mathbb{J}^c \cap \mathbb{K}} (\mu_i + \lambda_i) \left( n\mu_i \frac{\gamma_i^2}{(\gamma_i + \lambda_i)^2} + \frac{\lambda_i}{n(\gamma_i + \lambda_i)^2} \right) + \sum_{i \in \mathbb{J}^c \cap \mathbb{K}^c} (\mu_i + \lambda_i) \left( n\mu_i \frac{\gamma_i^2}{(\gamma_i + \lambda_i)^2} + \frac{\lambda_i}{n(\gamma_i + 1/n)^2} \right).
\end{aligned}
$$

When $\gamma_i = \mu_i$ for $i \leq k$ and $\gamma_i = 0$ otherwise, the above quantity satisfies

$$
\begin{aligned}
(*) \leq &\sum_{i \in \mathbb{K}, i \leq k} (\mu_i + \lambda_i) \left( \frac{\mu_i^2}{n(\mu_i + \lambda_i)^2} + \frac{\lambda_i}{n(\mu_i + \lambda_i)^2} \right) + \sum_{i \in \mathbb{K}^c, i \leq k} (\mu_i + \lambda_i) \left( \frac{\mu_i^2}{n(\mu_i + \lambda_i)^2} + \frac{\lambda_i}{n(\mu_i + \lambda_i)^2} \right) \\
&+ \sum_{i \in \mathbb{J} \cap \mathbb{K}, i > k} (\mu_i + \lambda_i) \frac{1}{n\lambda_i} + \sum_{i \in \mathbb{J} \cap \mathbb{K}^c, i > k} (\mu_i + \lambda_i) \cdot n\lambda_i + \sum_{i \in \mathbb{J}^c \cap \mathbb{K}} (\mu_i + \lambda_i) \frac{1}{n\lambda_i} + \sum_{i \in \mathbb{J}^c \cap \mathbb{K}^c} (\mu_i + \lambda_i) \cdot n\lambda_i \\
\leq &\sum_{i \in \mathbb{K}, i \leq k} \frac{1}{n} + \sum_{i \in \mathbb{K}^c, i \leq k} \frac{1}{n} + \sum_{i \in \mathbb{J} \cap \mathbb{K}, i > k} \frac{1 + \mu_i/\lambda_i}{n} + \sum_{i \in \mathbb{K}^c, i > k} (n\mu_i \lambda_i + n\lambda_i^2) + \sum_{i \in \mathbb{J}^c \cap \mathbb{K}} (n\mu_i \lambda_i + n\lambda_i^2) \\
\lesssim &\frac{1}{\sigma^2} \left( \texttt{variance}_{\texttt{joint}} + \frac{n}{k^\alpha} \texttt{variance}_{\texttt{joint}} \right).
\end{aligned}
$$

The last line holds because of Proposition 1 and Cauchy-Schwarz inequality; in particular, the fourth term is because that

$$
\begin{aligned}
\sum_{i \in \mathbb{K}^c, i > k} n\mu_i \lambda_i &\leq n \left( \sum_{i \in \mathbb{K}^c, i > k} \mu_i^2 \right)^{1/2} \left( \sum_{i \in \mathbb{K}^c, i > k} \lambda_i^2 \right)^{1/2} \\
&\leq n \left( n \cdot k^{-2\alpha} \cdot \frac{|\mathbb{J}|}{n} \right)^{1/2} \left( n^{-1} \cdot n \sum_{i \in \mathbb{K}^c} \lambda_i^2 \right)^{1/2} \lesssim \frac{n}{k^\alpha} \cdot \texttt{variance}_{\texttt{joint}}.
\end{aligned}
$$

Thus we finish the proof. $\qquad\square$

### B.8   PROOF OF EXAMPLES

*Proof of Example 5.* Recall that

$$
\mathbb{E}\Delta(\boldsymbol{w}^{(2)}) = \texttt{bias} + \texttt{variance},
$$

where `bias` and `variance` with respect $\boldsymbol{x}^{(1)}$ satisfy

$$\texttt{bias} \eqsim \langle (\boldsymbol{G}+\boldsymbol{H})(\boldsymbol{I}-\boldsymbol{G})^n(\boldsymbol{\Sigma}^2(\boldsymbol{\Sigma}+\boldsymbol{H})^{-2}+(\boldsymbol{I}-\boldsymbol{H})^n), \boldsymbol{w}^*\boldsymbol{w}^{*\top} \rangle$$

$$= \sum_i (1-\mu_i)^n \left( \frac{\gamma_i^2}{(\lambda_i+\gamma_i)^2} + (1-\lambda_i)^n \right)(\mu_i+\lambda_i)w_i^{*2},$$

$$\texttt{variance} \lesssim \sigma^2 \big( \langle (\boldsymbol{G}+\boldsymbol{H})(\boldsymbol{\Sigma}^2(\boldsymbol{\Sigma}+\boldsymbol{H})^{-2}+(\boldsymbol{I}-\boldsymbol{H})^n), \frac{1}{n}\boldsymbol{G}_{\mathbb{J}}^{-1} + n\boldsymbol{G}_{\mathbb{J}^c} \rangle$$

$$+ \langle \boldsymbol{G}+\boldsymbol{H}, \frac{1}{n}(\boldsymbol{H}_{\mathbb{K}}+\boldsymbol{\Sigma}_{\mathbb{K}})^{-2}\boldsymbol{H}_{\mathbb{K}} + n(\boldsymbol{I}_{\mathbb{K}^c}+n\boldsymbol{\Sigma}_{\mathbb{K}^c})^{-2}\boldsymbol{H}_{\mathbb{K}^c} \rangle \big).$$

1. By Corollary 3 we have

$$\mathbb{E}\Delta(\boldsymbol{w}^{(2)}) \gtrsim \frac{\sigma^2}{n} \left( \sum_{i \in \mathbb{K}} \frac{\mu_i}{\lambda_i} + n^2 \sum_{i \in \mathbb{J} \cap \mathbb{K}^c} \mu_i \lambda_i \right) = \Omega(1).$$

2. By Theorem 2 we have

$$\mathbb{E}\Delta(\boldsymbol{w}^{(2)}) \geq \langle \boldsymbol{H}\boldsymbol{\Sigma}^2(\boldsymbol{\Sigma}+\boldsymbol{H})^{-2}, \frac{1}{n}\boldsymbol{G}_{\mathbb{J}}^{-1} \rangle + \langle \boldsymbol{G}, \frac{1}{n}(\boldsymbol{H}_{\mathbb{K}}+\boldsymbol{\Sigma}_{\mathbb{K}})^{-2}\boldsymbol{H}_{\mathbb{K}} \rangle$$

$$= \sum_{i \in \mathbb{J}} \frac{\gamma^2}{\mu_i} \left( \frac{\lambda_i}{n(\gamma+\lambda_i)^2} \right) + \sum_{i \in \mathbb{K}} \mu_i \left( \frac{\lambda_i}{n(\gamma+\lambda_i)^2} \right)$$

$$\geq \frac{n}{3}\frac{\gamma^2}{(\gamma+1)^2} + \frac{n}{3}\frac{1}{n^2(\gamma+1/n)^2} = \Omega(1),$$

since $\frac{n}{3}\frac{\gamma^2}{(\gamma+1)^2} = \Omega(1)$ when $\gamma \geq n^{-1/2}$, and $\frac{n}{3}\frac{1}{n^2(\gamma+1/n)^2} = \Omega(1)$ when $\gamma \leq n^{-1/2}$.

$\square$

*Proof of Example 7.* Note that $\boldsymbol{\Sigma}$ has rank at most $k$. By Theorem 2 we have

$$\mathbb{E}\Delta(\boldsymbol{w}^{(2)}) \geq \langle \boldsymbol{G}, \frac{1}{n}(\boldsymbol{H}_{\mathbb{K}}+\boldsymbol{\Sigma}_{\mathbb{K}})^{-2}\boldsymbol{H}_{\mathbb{K}} + n(\boldsymbol{I}_{\mathbb{K}^c}+n\boldsymbol{\Sigma}_{\mathbb{K}^c})^{-2}\boldsymbol{H}_{\mathbb{K}^c} \rangle$$

$$= \sum_{i \in \mathbb{K}} \mu_i \left( \frac{\lambda_i}{n(\gamma_i+\lambda_i)^2} \right) + \sum_{i \in \mathbb{K}^c} \mu_i \left( \frac{\lambda_i}{n(\gamma_i+1/n)^2} \right)$$

$$\geq \frac{1}{k+1} \sum_{1 \leq i \leq k+1} \frac{1}{n^2(\gamma_i+1/n)^2} = \Omega(1),$$

since $k << n$ is a constant and there exists at least one $i$ such that $\gamma_i = 0$ for $1 \leq i \leq k+1$. $\square$

## C    PROOF OF GAUSSIAN OCL

In this section, we present the full analysis of OCL in Gaussian design as briefed in Section 5.

### C.1    UPPER AND LOWER BOUNDS

We present the lower and upper bounds of the OCL excess risk in the following theorems.

**Theorem 18** (Lower bound)**.** *There exist constants $b_1, b_2, c > 0$ for which the following holds. Denote index sets $\mathbb{J}, \mathbb{K}$ that are defined as follows. Let $\mathbb{J}_\mu = \{i : \mu_i > \mu\}$. Let $\mu^* = \max\{\mu : r_{\mathbb{J}_\mu^c}(\boldsymbol{G}) \geq b_2 n\}$, and define $\mathbb{J} := \mathbb{J}_{\mu^*}$. Similarly, let $\mathbb{K}_\lambda = \{i : \lambda_i > \lambda\}$, $\lambda^* = \max\{\lambda : r_{\mathbb{K}_\lambda^c}(\boldsymbol{H}) \geq b_2 n\}$, and define $\mathbb{K} := \mathbb{K}_{\lambda^*}$. Then if $|\mathbb{J}| \leq b_1 n$ and $|\mathbb{K}| \leq b_1 n$, for every $n > c$, for the OCL output* (1)*, it holds that*

$$\mathbb{E}[\mathcal{R}_1(\boldsymbol{w}^{(2)}) - \min \mathcal{R}_1] = \texttt{bias} + \texttt{variance}$$

*where*

$$\texttt{bias} \gtrsim \Big\|\Big(\frac{(\operatorname{tr}\boldsymbol{G}_{\mathbb{J}^c})^2}{n^2}\boldsymbol{G}_{\mathbb{J}}^{-2} + \boldsymbol{I}_{\mathbb{J}^c}\Big)^{\frac{1}{2}} \cdot \Big(\frac{(\operatorname{tr}\boldsymbol{H}_{\mathbb{K}^c})^2}{n^2}\boldsymbol{H}_{\mathbb{K}}^{-2} + \boldsymbol{I}_{\mathbb{K}^c}\Big)^{\frac{1}{2}}\boldsymbol{w}^*\Big\|_{\boldsymbol{G}}^2,$$

$$\texttt{variance} \gtrsim \frac{\sigma^2}{n}\Big\langle \boldsymbol{G},\ \boldsymbol{H}_{\mathbb{K}}^{-1} + \frac{n^2}{(\operatorname{tr}\boldsymbol{H}_{\mathbb{K}^c})^2}\boldsymbol{H}_{\mathbb{K}^c} + \Big(\frac{(\operatorname{tr}\boldsymbol{H}_{\mathbb{K}^c})^2}{n^2}\boldsymbol{H}_{\mathbb{K}}^{-2} + \boldsymbol{I}_{\mathbb{K}^c}\Big) \cdot \Big(\boldsymbol{G}_{\mathbb{J}}^{-1} + \frac{n^2}{(\operatorname{tr}\boldsymbol{G}_{\mathbb{J}^c})^2}\boldsymbol{G}_{\mathbb{J}^c}\Big)\Big\rangle.$$

The excess risk bound in Theorem 18 consists of both bias and variance errors. The bias error arises from the (incorrect) zero initialization, which deviates from the optimal parameter $\boldsymbol{w}^*$. The variance error is attributed to the additive noise $\boldsymbol{\varepsilon}^{(t)} = \boldsymbol{y}^{(t)} - \langle \boldsymbol{x}^{(t)}, \boldsymbol{w}^* \rangle$.

We complement the lower bound in Theorem 18 with an upper bound.

**Theorem 19** (Upper bound). *There exist constants $b_1, b_2, b_3 > 0$ such that the following holds. Suppose Assumptions 1, 2, and 4 hold. Denote two index sets $\mathbb{J}, \mathbb{K} \subset [d]$, which satisfy $|\mathbb{J}| \le b_1 n$, $r_{\mathbb{J}^c}(\boldsymbol{G}) \ge b_2 n$ and $|\mathbb{K}| \le b_1 n$, $r_{\mathbb{K}^c}(\boldsymbol{H}) \ge b_2 n$, where*

$$r_{\mathbb{J}^c}(\boldsymbol{G}) = \frac{\operatorname{tr}(\boldsymbol{G}_{\mathbb{J}^c})}{\|\boldsymbol{G}_{\mathbb{J}^c}\|_2}, \quad r_{\mathbb{K}^c}(\boldsymbol{H}) = \frac{\operatorname{tr}(\boldsymbol{H}_{\mathbb{K}^c})}{\|\boldsymbol{H}_{\mathbb{K}^c}\|_2}.$$

*Then, for the OCL output* (1), *it holds that*

$$\mathbb{E}[\mathcal{R}_1(\boldsymbol{w}^{(2)}) - \min \mathcal{R}_1] = \texttt{bias} + \texttt{variance},$$

*where* bias *and* variance *satisfy that with probability at least $1 - b_3 e^{-n/c}$,*

$$\texttt{bias} \lesssim \frac{(\operatorname{tr}\boldsymbol{G}_{\mathbb{J}^c})^2}{n^2}\|\boldsymbol{w}^*\|_{\boldsymbol{G}_{\mathbb{J}}^{-1}}^2 + \|\boldsymbol{w}^*\|_{\boldsymbol{G}_{\mathbb{J}^c}}^2,$$

$$\texttt{variance} \lesssim \frac{\sigma^2}{n}\Bigg( \operatorname{tr}(\boldsymbol{G}_{\mathbb{J}}\boldsymbol{H}_{\mathbb{K}}^{-1}) + n^2\frac{\operatorname{tr}(\boldsymbol{G}_{\mathbb{J}}\boldsymbol{H}_{\mathbb{K}^c})}{(\operatorname{tr}\boldsymbol{H}_{\mathbb{K}^c})^2} + |\mathbb{J} \cap \mathbb{K}| + n^2\frac{\operatorname{tr}(\boldsymbol{G}_{\mathbb{J}^c \cap \mathbb{K}^c}^2)}{(\operatorname{tr}\boldsymbol{G}_{\mathbb{J}^c})^2}$$

$$+ \Bigg( \|\boldsymbol{G}_{\mathbb{K}}\boldsymbol{H}_{\mathbb{K}}^{-1}\|_2 + \frac{n(\operatorname{tr}(\boldsymbol{G}_{\mathbb{K}^c}\boldsymbol{H}_{\mathbb{K}^c}) + n\|\boldsymbol{G}_{\mathbb{K}^c}\boldsymbol{H}_{\mathbb{K}^c}\|_2)}{\operatorname{tr}(\boldsymbol{H}_{\mathbb{K}^c})^2} + \frac{(\operatorname{tr}\boldsymbol{G}_{\mathbb{J}^c})^2}{(\operatorname{tr}\boldsymbol{H}_{\mathbb{K}^c})^2} \Bigg)$$

$$\cdot \operatorname{tr}\Bigg\{ \Big(\boldsymbol{G}_{\mathbb{J}}^{-1} + \frac{n^2}{\operatorname{tr}(\boldsymbol{G}_{\mathbb{J}^c})^2}\boldsymbol{G}_{\mathbb{J}^c}\Big)\Big(\frac{\operatorname{tr}(\boldsymbol{H}_{\mathbb{K}^c})^2}{n^2}\boldsymbol{H}_{\mathbb{K}}^{-1} + \boldsymbol{H}_{\mathbb{K}^c}\Big) \Bigg\}\Bigg).$$

Theorem 19 provides an upper bound for the risk and forgetting in OCL under the Gaussian distribution setting. It can be shown that when $\boldsymbol{G} = \boldsymbol{H}$, the upper bound reduces to the single-task linear regression bound in Bartlett et al. (2020), thus matching the lower bound. However, for the non-degenerate case, there exists at least a $\|\boldsymbol{G}_{\mathbb{K}}\boldsymbol{H}_{\mathbb{K}}^{-1}\|_2$ gap in the fifth term concerning the variance error. The gaps between the upper and lower bounds are due to technical challenges in obtaining an accurate variance bound under covariate shift in the Gaussian distribution. We leave the task of tightening these bounds for future work.

## C.2 PRELIMINARIES

For both task $t = 1, 2$, for any index $i \in [d]$ and index set $\mathbb{K} \subset [d]$ where $d \le \infty$, we denote

$$\boldsymbol{A}^{(t)} = \boldsymbol{X}^{(t)}\boldsymbol{X}^{(t)\top}, \quad \boldsymbol{A}_{-i}^{(t)} = \boldsymbol{X}^{(t)}\boldsymbol{X}^{(t)\top} - \boldsymbol{X}_i^{(t)}\boldsymbol{X}_i^{(t)\top}, \quad \boldsymbol{A}_{\mathbb{K}^c}^{(t)} = \boldsymbol{X}_{\mathbb{K}^c}^{(t)}\boldsymbol{X}_{\mathbb{K}^c}^{(t)\top}.$$

Recall that according to (15) and (17), the second moments of the output used in bounding the risks are specified by

$$\mathbb{E}_\varepsilon(\boldsymbol{w}^{(1)} - \boldsymbol{w}^*)(\boldsymbol{w}^{(1)} - \boldsymbol{w}^*)^\top = \mathcal{P}^{(1)} \cdot \boldsymbol{w}^*\boldsymbol{w}^{*\top} \cdot \mathcal{P}^{(1)} + \sigma^2\big(\boldsymbol{X}^{(1)\top}\boldsymbol{X}^{(1)}\big)^{-1}\boldsymbol{X}^{(1)\top}\boldsymbol{X}^{(1)}\big(\boldsymbol{X}^{(1)\top}\boldsymbol{X}^{(1)}\big)^{-1}$$

$$= \mathcal{P}^{(1)} \cdot \boldsymbol{w}^*\boldsymbol{w}^{*\top} \cdot \mathcal{P}^{(1)} + \sigma^2\boldsymbol{X}^{(1)\top}\boldsymbol{A}^{(1)-2}\boldsymbol{X}^{(1)}, \quad\quad (23)$$

$$\mathbb{E}_\varepsilon(\boldsymbol{w}^{(2)} - \boldsymbol{w}^*)(\boldsymbol{w}^{(2)} - \boldsymbol{w}^*)^\top = \mathcal{P}^{(2)} \cdot \mathbb{E}_\varepsilon(\boldsymbol{w}^{(1)} - \boldsymbol{w}^*)(\boldsymbol{w}^{(1)} - \boldsymbol{w}^*)^\top \cdot \mathcal{P}^{(2)}$$

$$+ \sigma^2\big(\boldsymbol{X}^{(2)\top}\boldsymbol{X}^{(2)}\big)^{-1}\boldsymbol{X}^{(2)\top}\boldsymbol{X}^{(1)}\big(\boldsymbol{X}^{(2)\top}\boldsymbol{X}^{(2)}\big)^{-1}$$

$$= \mathcal{P}^{(2)} \cdot \mathbb{E}_\varepsilon(\boldsymbol{w}^{(1)} - \boldsymbol{w}^*)(\boldsymbol{w}^{(1)} - \boldsymbol{w}^*)^\top \cdot \mathcal{P}^{(2)} + \sigma^2\boldsymbol{X}^{(2)\top}\boldsymbol{A}^{(2)-2}\boldsymbol{X}^{(2)},$$

$$(24)$$

where $\mathcal{P}^{(1)} = \boldsymbol{I} - \left(\boldsymbol{X}^{(1)\top}\boldsymbol{X}^{(1)}\right)^{-1}\boldsymbol{X}^{(1)\top}\boldsymbol{X}^{(1)}$ and $\mathcal{P}^{(2)} = \boldsymbol{I} - \left(\boldsymbol{X}^{(2)\top}\boldsymbol{X}^{(2)}\right)^{-1}\boldsymbol{X}^{(2)\top}\boldsymbol{X}^{(2)}$.
Therefore, the risk that we are bounding in Theorem 18 depends on terms including $\mathcal{P}^{(1)}, \mathcal{P}^{(2)}$,
$\boldsymbol{X}^{(1)\top}\boldsymbol{A}^{(1)-2}\boldsymbol{X}^{(1)}$ and $\boldsymbol{X}^{(2)\top}\boldsymbol{A}^{(2)-2}\boldsymbol{X}^{(2)}$.

## C.3 SPLIT THE COORDINATES

Motivated by the previous benign overfitting literature, including Bartlett et al. (2020) and Tsigler & Bartlett (2023), we split the coordinates of the covariance matrix of each task $\boldsymbol{G}$ and $\boldsymbol{H}$ into a "head" part and a "tail" part. The head part represents a low-dimensional space with relatively large eigenvalues, and the tail part represents a high-dimensional space with relatively small eigenvalues. The following lemma is an algebraic property in the same spirit of the one in Tsigler & Bartlett (2023), and is used in splitting the coordinates of $\boldsymbol{X}^{(1)\top}\boldsymbol{A}^{(1)-2}\boldsymbol{X}^{(1)}$ and $\boldsymbol{X}^{(2)\top}\boldsymbol{A}^{(2)-2}\boldsymbol{X}^{(2)}$.

**Lemma 20.** *For any nonempty index set $\mathbb{K} \subset [d]$,*

$$\boldsymbol{X}_{\mathbb{K}}^{(1)\top}\boldsymbol{A}^{(1)-1} = \left(\boldsymbol{I}_{\mathbb{K}} + \boldsymbol{X}_{\mathbb{K}}^{(1)\top}\boldsymbol{A}_{\mathbb{K}^c}^{(1)-1}\boldsymbol{X}_{\mathbb{K}}^{(1)}\right)^{-1}\boldsymbol{X}_{\mathbb{K}}^{(1)\top}\boldsymbol{A}_{\mathbb{K}^c}^{(1)-1},$$

$$\boldsymbol{X}_{\mathbb{K}}^{(2)\top}\boldsymbol{A}^{(2)-1} = \left(\boldsymbol{I}_{\mathbb{K}} + \boldsymbol{X}_{\mathbb{K}}^{(2)\top}\boldsymbol{A}_{\mathbb{K}^c}^{(2)-1}\boldsymbol{X}_{\mathbb{K}}^{(2)}\right)^{-1}\boldsymbol{X}_{\mathbb{K}}^{(2)\top}\boldsymbol{A}_{\mathbb{K}^c}^{(2)-1}.$$

*Proof.* We will give the proof for $\boldsymbol{X}_{\mathbb{K}}^{(2)\top}\boldsymbol{A}^{(2)-1}$. Notice that

$$\boldsymbol{A}^{(2)} = \boldsymbol{X}^{(2)}\boldsymbol{X}^{(2)\top} = \boldsymbol{X}_{\mathbb{K}}^{(2)}\boldsymbol{X}_{\mathbb{K}}^{(2)\top} + \boldsymbol{X}_{\mathbb{K}^c}^{(2)}\boldsymbol{X}_{\mathbb{K}^c}^{(2)\top} = \boldsymbol{A}_{\mathbb{K}^c}^{(2)} + \boldsymbol{X}_{\mathbb{K}}^{(2)}\boldsymbol{X}_{\mathbb{K}}^{(2)\top},$$

and according to the Sherman-Morrison-Woodbury formula,

$$\begin{aligned}
\boldsymbol{X}_{\mathbb{K}}^{(2)}\boldsymbol{A}^{(2)-1} &= \boldsymbol{X}_{\mathbb{K}}^{(2)}\left(\boldsymbol{A}_{\mathbb{K}^c}^{(2)} + \boldsymbol{X}_{\mathbb{K}}^{(2)}\boldsymbol{X}_{\mathbb{K}}^{(2)\top}\right)^{-1} \\
&= \boldsymbol{X}_{\mathbb{K}}^{(2)}\boldsymbol{A}_{\mathbb{K}^c}^{(2)-1}\left(\boldsymbol{I} + \boldsymbol{X}_{\mathbb{K}}^{(2)}\boldsymbol{X}_{\mathbb{K}}^{(2)\top}\boldsymbol{A}_{\mathbb{K}^c}^{(2)-1}\right)^{-1} \\
&= \boldsymbol{X}_{\mathbb{K}}^{(2)}\boldsymbol{A}_{\mathbb{K}^c}^{(2)-1}\left[\boldsymbol{I} - \boldsymbol{X}_{\mathbb{K}}^{(2)}\left(\boldsymbol{I}_{\mathbb{K}} + \boldsymbol{X}_{\mathbb{K}}^{(2)\top}\boldsymbol{A}_{\mathbb{K}^c}^{(2)-1}\boldsymbol{X}_{\mathbb{K}}^{(2)}\right)^{-1}\boldsymbol{X}_{\mathbb{K}}^{(2)\top}\boldsymbol{A}_{\mathbb{K}^c}^{(2)-1}\right] \\
&= \left[\boldsymbol{I}_{\mathbb{K}} - \boldsymbol{X}_{\mathbb{K}}^{(2)\top}\boldsymbol{A}_{\mathbb{K}^c}^{(2)-1}\boldsymbol{X}_{\mathbb{K}}^{(2)}\left(\boldsymbol{I}_{\mathbb{K}} + \boldsymbol{X}_{\mathbb{K}}^{(2)\top}\boldsymbol{A}_{\mathbb{K}^c}^{(2)-1}\boldsymbol{X}_{\mathbb{K}}^{(2)}\right)^{-1}\right]\boldsymbol{X}_{\mathbb{K}}^{(2)\top}\boldsymbol{A}_{\mathbb{K}^c}^{(2)-1} \\
&= \left(\boldsymbol{I}_{\mathbb{K}} + \boldsymbol{X}_{\mathbb{K}}^{(2)\top}\boldsymbol{A}_{\mathbb{K}^c}^{(2)-1}\boldsymbol{X}_{\mathbb{K}}^{(2)}\right)^{-1}\boldsymbol{X}_{\mathbb{K}}^{(2)\top}\boldsymbol{A}_{\mathbb{K}^c}^{(2)-1}.
\end{aligned}$$

The proof goes the same for the first task. $\square$

The next lemma splits $\mathcal{P}^{(1)}$ and $\mathcal{P}^{(2)}$ into submatrices with respect to the coordinate split in each task.

**Lemma 21.** *For $t = 1, 2$, for any nonempty index set $\mathbb{K}$,*

$$\mathcal{P}^{(t)} := \begin{pmatrix} \mathcal{P}_{\mathbb{K}}^{(t)} & -\mathcal{Q}^{(t)} \\ -\mathcal{Q}^{(t)\top} & \mathcal{P}_{\mathbb{K}^c}^{(t)} \end{pmatrix} = \begin{pmatrix} \left(\boldsymbol{I}_{\mathbb{K}} + \boldsymbol{X}_{\mathbb{K}}^{(t)\top}\boldsymbol{A}_{\mathbb{K}^c}^{(t)-1}\boldsymbol{X}_{\mathbb{K}}^{(t)}\right)^{-1} & -\boldsymbol{X}_{\mathbb{K}}^{(t)\top}\boldsymbol{A}^{(t)-1}\boldsymbol{X}_{\mathbb{K}^c}^{(t)} \\ -\boldsymbol{X}_{\mathbb{K}^c}^{(t)\top}\boldsymbol{A}^{(t)-1}\boldsymbol{X}_{\mathbb{K}}^{(t)} & \boldsymbol{I}_{\mathbb{K}^c} - \boldsymbol{X}_{\mathbb{K}^c}^{(t)\top}\boldsymbol{A}^{(t)-1}\boldsymbol{X}_{\mathbb{K}^c}^{(t)} \end{pmatrix}.$$

*Proof.* Notice that

$$\begin{aligned}
\mathcal{P}^{(t)} &= \boldsymbol{I} - \boldsymbol{X}^{\top}\boldsymbol{A}^{-1}\boldsymbol{X} \\
&= \boldsymbol{I} - \begin{pmatrix} \boldsymbol{X}_{\mathbb{K}}^{(t)\top}\boldsymbol{A}^{-1}\boldsymbol{X}_{\mathbb{K}}^{(t)} & \boldsymbol{X}_{\mathbb{K}}^{(t)\top}\boldsymbol{A}^{-1}\boldsymbol{X}_{\mathbb{K}^c}^{(t)} \\ \boldsymbol{X}_{\mathbb{K}^c}^{(t)\top}\boldsymbol{A}^{-1}\boldsymbol{X}_{\mathbb{K}}^{(t)} & \boldsymbol{X}_{\mathbb{K}^c}^{(t)\top}\boldsymbol{A}^{-1}\boldsymbol{X}_{\mathbb{K}^c}^{(t)} \end{pmatrix} \\
&= \begin{pmatrix} \boldsymbol{I}_{\mathbb{K}} - \boldsymbol{X}_{\mathbb{K}}^{(t)\top}\boldsymbol{A}^{-1}\boldsymbol{X}_{\mathbb{K}}^{(t)} & -\boldsymbol{X}_{\mathbb{K}}^{(t)\top}\boldsymbol{A}^{-1}\boldsymbol{X}_{\mathbb{K}^c}^{(t)} \\ -\boldsymbol{X}_{\mathbb{K}^c}^{(t)\top}\boldsymbol{A}^{-1}\boldsymbol{X}_{\mathbb{K}}^{(t)} & \boldsymbol{I}_{\mathbb{K}^c} - \boldsymbol{X}_{\mathbb{K}^c}^{(t)\top}\boldsymbol{A}^{-1}\boldsymbol{X}_{\mathbb{K}^c}^{(t)} \end{pmatrix}.
\end{aligned}$$

Therefore, by Lemma 20,

$$\begin{aligned}
\mathcal{P}_{\mathbb{K}}^{(t)} &= \boldsymbol{I}_{\mathbb{K}} - \boldsymbol{X}_{\mathbb{K}}^{(t)\top}\boldsymbol{A}^{-1}\boldsymbol{X}_{\mathbb{K}}^{(t)} \\
&= \boldsymbol{I}_{\mathbb{K}} - \left(\boldsymbol{I}_{\mathbb{K}} + \boldsymbol{X}_{\mathbb{K}}^{(t)\top}\boldsymbol{A}_{\mathbb{K}^c}^{(t)-1}\boldsymbol{X}_{\mathbb{K}}^{(t)}\right)^{-1}\boldsymbol{X}_{\mathbb{K}}^{(t)\top}\boldsymbol{A}_{\mathbb{K}^c}^{(t)-1}\boldsymbol{X}_{\mathbb{K}}^{(t)} \\
&= \left(\boldsymbol{I}_{\mathbb{K}} + \boldsymbol{X}_{\mathbb{K}}^{(t)\top}\boldsymbol{A}_{\mathbb{K}^c}^{(t)-1}\boldsymbol{X}_{\mathbb{K}}^{(t)}\right)^{-1}.
\end{aligned}$$

We have finished the proof. $\square$

## C.4 MATRIX CONCENTRATION INEQUALITIES

In the following lemmas, we will present several useful concentration inequalities on eigenvalues of several matrices used in our proof. The following lemma is from Lemma 9 in Bartlett et al. (2020) and we rewrite it in our notation.

**Lemma 22** (Bartlett et al. (2020)). *There exists a constant $c \geq 1$ such that for any PSD matrix $\boldsymbol{J}$, with probability at least $1 - 2e^{-n/c}$,*

$$\frac{1}{c} \operatorname{tr} \boldsymbol{J} - cn\|\boldsymbol{J}\|_2 \leq \mu_n(\boldsymbol{Z}\boldsymbol{J}\boldsymbol{Z}^\top) \leq \mu_1(\boldsymbol{Z}\boldsymbol{J}\boldsymbol{Z}^\top) \leq c(\operatorname{tr} \boldsymbol{J} + n\|\boldsymbol{J}\|_2).$$

Notice that one can verify that this is identical to the original lemma in Bartlett et al. (2020), since in their notation, $\boldsymbol{A}_k = \boldsymbol{Z}_k \boldsymbol{H}_k \boldsymbol{Z}_k^\top$ and $\boldsymbol{H}_k = \operatorname{diag}\{\lambda_i\}_{i>k}$.

**Lemma 23.** *There exist a constant $b$ such that with probability at least $1 - 2e^{-n/c}$, if $r_{\mathbb{J}^c}(\boldsymbol{G}) \geq bn$ and $r_{\mathbb{K}^c}(\boldsymbol{H}) \geq bn$, then*

$$\operatorname{tr} \boldsymbol{G}_{\mathbb{J}^c} \cdot \boldsymbol{I}_n \lesssim \boldsymbol{A}_{\mathbb{J}^c}^{(1)} \lesssim \operatorname{tr} \boldsymbol{G}_{\mathbb{J}^c} \cdot \boldsymbol{I}_n, \qquad \operatorname{tr} \boldsymbol{H}_{\mathbb{K}^c} \cdot \boldsymbol{I}_n \lesssim \boldsymbol{A}_{\mathbb{K}^c}^{(2)} \lesssim \operatorname{tr} \boldsymbol{H}_{\mathbb{K}^c} \cdot \boldsymbol{I}_n.$$

*Proof.* We will give the proof for the second task regarding $\boldsymbol{H}$ and $\boldsymbol{A}^{(2)}$. Recall that $\boldsymbol{A}_{\mathbb{K}^c}^{(2)} = \boldsymbol{X}_{\mathbb{K}^c}^{(2)} \boldsymbol{X}_{\mathbb{K}^c}^{(2)\top}$ Notice that since $r_{\mathbb{K}^c}(\boldsymbol{H}) = \operatorname{tr}(\boldsymbol{H}_{\mathbb{K}^c})/\|\boldsymbol{H}_{\mathbb{K}^c}\|_2 \geq bn$. Therefore,

$$\begin{aligned}
\mu_n(\boldsymbol{A}_{\mathbb{K}^c}^{(2)}) &= \mu_n(\boldsymbol{Z}_{\mathbb{K}^c}^{(2)} \boldsymbol{H}_{\mathbb{K}^c} \boldsymbol{Z}_{\mathbb{K}^c}^{(2)\top}) \\
&\geq \frac{1}{c} \operatorname{tr} \boldsymbol{H}_{\mathbb{K}^c} - cn\|\boldsymbol{H}_{\mathbb{K}^c}\|_2 \\
&\geq \left(\frac{1}{c} - \frac{c}{b}\right) \operatorname{tr} \boldsymbol{H}_{\mathbb{K}^c}.
\end{aligned}$$

Also,

$$\begin{aligned}
\mu_1(\boldsymbol{A}_{\mathbb{K}^c}^{(2)}) &= \mu_1(\boldsymbol{Z}_{\mathbb{K}^c}^{(2)} \boldsymbol{H}_{\mathbb{K}^c} \boldsymbol{Z}_{\mathbb{K}^c}^{(2)\top}) \\
&\leq c(\operatorname{tr} \boldsymbol{H}_{\mathbb{K}^c} + n\|\boldsymbol{H}_{\mathbb{K}^c}\|_2) \\
&\leq c\left(1 + \frac{1}{b}\right) \operatorname{tr} \boldsymbol{H}_{\mathbb{K}^c}.
\end{aligned}$$

The proof remains the same for the first task regarding $\boldsymbol{G}$ and $\boldsymbol{A}^{(1)}$. $\qquad\square$

**Lemma 24.** *There exist constants $b_1, b_2, c > 0$ such that for any nonempty index set $\mathbb{K}$ that satisfies $|\mathbb{K}| =: k \leq b_1 n$, then with probability at least $1 - 2e^{-n/c}$,*

$$\operatorname{tr}(\boldsymbol{H}_{\mathbb{J}^c}) \lesssim \mu_n(\boldsymbol{A}_{-i}^{(2)}) \leq \mu_{k+1}(\boldsymbol{A}_{-i}^{(2)}) \lesssim \operatorname{tr}(\boldsymbol{H}_{\mathbb{J}^c}).$$

*Proof.* Notice that for $\mathbb{K}$ that satisfies $r_{\mathbb{K}^c}(\boldsymbol{H}) \geq bn$, it also holds for every $i$ that $r_{\mathbb{K}^c - \{i\}}(\boldsymbol{H}) \geq bn$, since

$$\begin{aligned}
\operatorname{tr}(\boldsymbol{H}_{\mathbb{K}^c - \{i\}}) &= \operatorname{tr}(\boldsymbol{H}_{\mathbb{K}^c}) - \lambda_i \\
&\geq bn\|\boldsymbol{H}_{\mathbb{K}^c}\| - \|\boldsymbol{H}_{\mathbb{K}^c}\| \\
&\geq b'n\|\boldsymbol{H}_{\mathbb{K}^c - \{i\}}\|.
\end{aligned}$$

As a result, $\operatorname{tr}(\boldsymbol{H}_{\mathbb{J}^c}) \lesssim \mu_n(\boldsymbol{A}_{\mathbb{K}^c - \{i\}}^{(2)}) \leq \mu_1(\boldsymbol{A}_{\mathbb{K}^c - \{i\}}^{(2)}) \lesssim \operatorname{tr}(\boldsymbol{H}_{\mathbb{J}^c})$. Notice that $\boldsymbol{A}_{-i}^{(2)} - \boldsymbol{A}_{\mathbb{K}^c - \{i\}}^{(2)}$ is a PSD matrix with rank at most $|\mathbb{K}| = k$. Therefore, $\mu_n(\boldsymbol{A}_{-i}^{(2)}) \geq \mu_n(\boldsymbol{A}_{\mathbb{K}^c - \{i\}}^{(2)}) \gtrsim \operatorname{tr}(\boldsymbol{H}_{\mathbb{J}^c})$; also, there exists a linear space $\mathcal{L}$ with rank $n - k$ such that for all $\boldsymbol{v} \in \mathcal{L}$, $\boldsymbol{v}^\top \boldsymbol{A}_{-i}^{(2)} \boldsymbol{v} = \boldsymbol{v}^\top \boldsymbol{A}_{\mathbb{K}^c - \{i\}}^{(2)} \boldsymbol{v} \leq \mu_1(\boldsymbol{A}_{\mathbb{K}^c - \{i\}}^{(2)})\|\boldsymbol{v}\|_2^2$, and thus $\mu_{k+1}(\boldsymbol{A}_{-i}^{(2)}) \leq \mu_1(\boldsymbol{A}_{\mathbb{K}^c - \{i\}}^{(2)}) \lesssim \operatorname{tr}(\boldsymbol{H}_{\mathbb{J}^c})$. $\qquad\square$

The next two are commonly used concentration inequalities for random matrices with standard Gaussian rows.

**Lemma 25.** *For $t = 1, 2$, there exist constants $b, c > 0$ such that for any nonempty index set $\mathbb{K}$ that satisfies $|\mathbb{K}| = k \le bn$, then with probability at least $1 - 2e^{-n/c}$,*

$$n\boldsymbol{I}_{\mathbb{K}} \precsim \boldsymbol{Z}_{\mathbb{K}}^{(t)^\top} \boldsymbol{Z}_{\mathbb{K}}^{(t)} \precsim n\boldsymbol{I}_{\mathbb{K}}.$$

*Proof.* According to Theorem 5.39 in Vershynin (2010), for some constants $c_1, c_2$, for every $t \ge 0$, with probability at least $1 - 2\exp(-c_1 t^2)$, one has

$$(\sqrt{n} - c_2\sqrt{|\mathbb{K}|} - t)^2 \le \mu_k(\boldsymbol{Z}_{\mathbb{K}}^{(t)^\top} \boldsymbol{Z}_{\mathbb{K}}^{(t)}) \le \mu_1(\boldsymbol{Z}_{\mathbb{K}}^{(t)^\top} \boldsymbol{Z}_{\mathbb{K}}^{(t)}) \le (\sqrt{n} + c_2\sqrt{|\mathbb{K}|} + t)^2.$$

Substituting $t$ with $\sqrt{n}/c_3$ and we get the result. $\qquad\square$

**Lemma 26.** *Suppose $\boldsymbol{Z}$ is a matrix with $n$ i.i.d. rows in the Hilbert space $\mathbb{H}$ with standard Gaussian entries. Then for any PSD matrix $\boldsymbol{J}$, with probability at least $1 - 2e^{-n/c}$,*

$$\operatorname{tr}(\boldsymbol{Z}\boldsymbol{J}\boldsymbol{Z}^\top) \lesssim n \cdot \operatorname{tr}(\boldsymbol{J}).$$

*Proof.* Denote $\boldsymbol{J} = \boldsymbol{V}\boldsymbol{\Lambda}\boldsymbol{V}^\top$ as its singular value decomposition. Notice that each row $\boldsymbol{z}_i \in \boldsymbol{H}$ of $\boldsymbol{Z}$ is standard Gaussian. As a result,

$$\operatorname{tr}(\boldsymbol{Z}\boldsymbol{J}\boldsymbol{Z}^\top) = \sum_{i=1}^{n} \|\boldsymbol{\Lambda}\boldsymbol{V}^\top \boldsymbol{z}_i\|_2^2,$$

where $\|\boldsymbol{\Lambda}\boldsymbol{V}^\top \boldsymbol{z}_i\|_2^2$ are independent sub-exponential random variables with expectation $\operatorname{tr}(\boldsymbol{\Lambda})$ and sub-exponential norms bounded by $c_1 \operatorname{tr}(\boldsymbol{\Lambda})$. Also note that $\operatorname{tr}(\boldsymbol{\Lambda}) = \operatorname{tr}(\boldsymbol{J})$. Therefore, according to Bernstein's inequality,

$$P\left(\left|\frac{1}{n}\operatorname{tr}(\boldsymbol{Z}\boldsymbol{J}\boldsymbol{Z}^\top) - \operatorname{tr}\boldsymbol{J}\right| \ge t\operatorname{tr}\boldsymbol{J}\right) \le 2\exp(-c_2 n \min(t^2, t)).$$

By substituting $t$ with a constant we get our result. $\qquad\square$

## C.5 UPPER BOUNDS

**Bounding $\boldsymbol{X}^\top \boldsymbol{A}^{-1}\boldsymbol{X}$.**

**Lemma 27.** *There exist constants $b_1, b_2, b_3 > 0$ for which the following holds. Denote two index sets $\mathbb{J}, \mathbb{K}$ which satisfy $|\mathbb{J}| \le b_1 n$, $r_{\mathbb{J}^c}(\boldsymbol{G}) \ge b_2 n$ and $|\mathbb{K}| \le b_1 n$, $r_{\mathbb{K}^c}(\boldsymbol{H}) \ge b_2 n$. Then with probability at least $1 - b_3 e^{-n/c}$,*

$$\boldsymbol{X}_{\mathbb{J}}^{(1)^\top} \boldsymbol{A}^{(1)^{-2}} \boldsymbol{X}_{\mathbb{J}}^{(1)} \precsim \frac{1}{n} \cdot \boldsymbol{G}_{\mathbb{J}}^{-1}, \quad \boldsymbol{X}_{\mathbb{K}}^{(2)^\top} \boldsymbol{A}^{(2)^{-2}} \boldsymbol{X}_{\mathbb{K}}^{(2)} \precsim \frac{1}{n} \cdot \boldsymbol{H}_{\mathbb{K}}^{-1}.$$

*Proof.* We will give the proof for the second task. By Lemma 20,

$$\boldsymbol{X}_{\mathbb{K}}^{(2)^\top} \boldsymbol{A}^{(2)^{-2}} \boldsymbol{X}_{\mathbb{K}}^{(2)}$$

$$= \left(\boldsymbol{I}_{\mathbb{K}} + \boldsymbol{X}_{\mathbb{K}}^{(2)^\top} \boldsymbol{A}_{\mathbb{K}^c}^{(2)^{-1}} \boldsymbol{X}_{\mathbb{K}}^{(2)}\right)^{-1} \boldsymbol{X}_{\mathbb{K}}^{(2)^\top} \boldsymbol{A}_{\mathbb{K}^c}^{(2)^{-2}} \boldsymbol{X}_{\mathbb{K}}^{(2)} \left(\boldsymbol{I}_{\mathbb{K}} + \boldsymbol{X}_{\mathbb{K}}^{(2)^\top} \boldsymbol{A}_{\mathbb{K}^c}^{(2)^{-1}} \boldsymbol{X}_{\mathbb{K}}^{(2)}\right)^{-1}$$

$$= \boldsymbol{H}_{\mathbb{K}}^{-1/2} \left(\boldsymbol{H}_{\mathbb{K}}^{-1} + \boldsymbol{Z}_{\mathbb{K}}^{(2)^\top} \boldsymbol{A}_{\mathbb{K}^c}^{(2)^{-1}} \boldsymbol{Z}_{\mathbb{K}}^{(2)}\right)^{-1} \boldsymbol{Z}_{\mathbb{K}}^{(2)^\top} \boldsymbol{A}_{\mathbb{K}^c}^{(2)^{-2}} \boldsymbol{Z}_{\mathbb{K}}^{(2)} \left(\boldsymbol{H}_{\mathbb{K}}^{-1} + \boldsymbol{Z}_{\mathbb{K}}^{(2)^\top} \boldsymbol{A}_{\mathbb{K}^c}^{(2)^{-1}} \boldsymbol{Z}_{\mathbb{K}}^{(2)}\right)^{-1} \boldsymbol{H}_{\mathbb{K}}^{-1/2}$$

$$\precsim \frac{1}{(\operatorname{tr}\boldsymbol{H}_{\mathbb{K}^c})^2} \boldsymbol{H}_{\mathbb{K}}^{-1/2} \left(\boldsymbol{H}_{\mathbb{K}}^{-1} + \boldsymbol{Z}_{\mathbb{K}}^{(2)^\top} \boldsymbol{A}_{\mathbb{K}^c}^{(2)^{-1}} \boldsymbol{Z}_{\mathbb{K}}^{(2)}\right)^{-1} \boldsymbol{Z}_{\mathbb{K}}^{(2)^\top} \boldsymbol{Z}_{\mathbb{K}}^{(2)} \left(\boldsymbol{H}_{\mathbb{K}}^{-1} + \boldsymbol{Z}_{\mathbb{K}}^{(2)^\top} \boldsymbol{A}_{\mathbb{K}^c}^{(2)^{-1}} \boldsymbol{Z}_{\mathbb{K}}^{(2)}\right)^{-1} \boldsymbol{H}_{\mathbb{K}}^{-1/2}$$

$$\precsim \frac{n}{(\operatorname{tr}\boldsymbol{H}_{\mathbb{K}^c})^2} \boldsymbol{H}_{\mathbb{K}}^{-1/2} \left(\boldsymbol{H}_{\mathbb{K}}^{-1} + \boldsymbol{Z}_{\mathbb{K}}^{(2)^\top} \boldsymbol{A}_{\mathbb{K}^c}^{(2)^{-1}} \boldsymbol{Z}_{\mathbb{K}}^{(2)}\right)^{-2} \boldsymbol{H}_{\mathbb{K}}^{-1/2}$$

$$\preceq \frac{n}{(\operatorname{tr}\boldsymbol{H}_{\mathbb{K}^c})^2} \boldsymbol{H}_{\mathbb{K}}^{-1/2} \left(\boldsymbol{Z}_{\mathbb{K}}^{(2)^\top} \boldsymbol{A}_{\mathbb{K}^c}^{(2)^{-1}} \boldsymbol{Z}_{\mathbb{K}}^{(2)}\right)^{-2} \boldsymbol{H}_{\mathbb{K}}^{-1/2}$$

$$\preceq \frac{n}{(\operatorname{tr}\boldsymbol{H}_{\mathbb{K}^c})^2} \cdot \frac{(\operatorname{tr}\boldsymbol{H}_{\mathbb{K}^c})^2}{n^2} \boldsymbol{H}_{\mathbb{K}}^{-1}$$

$$= \frac{1}{n}\boldsymbol{H}_{\mathbb{K}}^{-1}.$$

The proof goes the same for the first task. $\qquad\square$

**Lemma 28.** *There exist constants $b_1, b_2, b_3 > 0$ for which the following holds. Denote two index sets $\mathbb{J}, \mathbb{K}$ which satisfy $|\mathbb{J}| \leq b_1 n$, $r_{\mathbb{J}^c}(\boldsymbol{G}) \geq b_2 n$ and $|\mathbb{K}| \leq b_1 n$, $r_{\mathbb{K}^c}(\boldsymbol{H}) \geq b_2 n$. Then for any PSD matrix $\boldsymbol{J}$, with probability at least $1 - b_3 e^{-n/c}$,*

$$\langle \boldsymbol{J},\ \boldsymbol{X}_{\mathbb{K}^c}^{(1)}{}^{\top} \boldsymbol{A}^{(1)}{}^{-2} \boldsymbol{X}_{\mathbb{K}^c}^{(1)} \rangle \lesssim \frac{n}{(\operatorname{tr} \boldsymbol{G}_{\mathbb{J}^c})^2} \cdot \langle \boldsymbol{J},\ \boldsymbol{G}_{\mathbb{J}^c} \rangle,$$

$$\langle \boldsymbol{J},\ \boldsymbol{X}_{\mathbb{K}^c}^{(2)}{}^{\top} \boldsymbol{A}^{(2)}{}^{-2} \boldsymbol{X}_{\mathbb{K}^c}^{(2)} \rangle \lesssim \frac{n}{(\operatorname{tr} \boldsymbol{H}_{\mathbb{K}^c})^2} \cdot \langle \boldsymbol{J},\ \boldsymbol{H}_{\mathbb{K}^c} \rangle.$$

*Proof.* We will give the proof for the second task.

$$
\begin{aligned}
\langle \boldsymbol{J},\ \boldsymbol{X}_{\mathbb{K}^c}^{(2)}{}^{\top} \boldsymbol{A}^{(2)}{}^{-2} \boldsymbol{X}_{\mathbb{K}^c}^{(2)} \rangle &= \langle \boldsymbol{H}_{\mathbb{K}^c}^{1/2} \boldsymbol{J} \boldsymbol{H}_{\mathbb{K}^c}^{1/2},\ \boldsymbol{Z}_{\mathbb{K}^c}^{(2)}{}^{\top} \boldsymbol{A}^{(2)}{}^{-2} \boldsymbol{Z}_{\mathbb{K}^c}^{(2)} \rangle \\
&\lesssim \frac{1}{(\operatorname{tr} \boldsymbol{H}_{\mathbb{K}^c})^2} \operatorname{tr}(\boldsymbol{Z}_{\mathbb{K}^c}^{(2)} \boldsymbol{H}_{\mathbb{K}^c}^{1/2} \boldsymbol{J} \boldsymbol{H}_{\mathbb{K}^c}^{1/2} \boldsymbol{Z}_{\mathbb{K}^c}^{(2)}{}^{\top}) \\
&\lesssim \frac{n}{(\operatorname{tr} \boldsymbol{H}_{\mathbb{K}^c})^2} \operatorname{tr}(\boldsymbol{H}_{\mathbb{K}^c}^{1/2} \boldsymbol{J} \boldsymbol{H}_{\mathbb{K}^c}^{1/2}) \\
&= \frac{n}{(\operatorname{tr} \boldsymbol{H}_{\mathbb{K}^c})^2} \cdot \langle \boldsymbol{J},\ \boldsymbol{H}_{\mathbb{K}^c} \rangle.
\end{aligned}
$$

The proof goes the same for the first task. $\qquad\square$

**Bounding $\mathcal{P} G \mathcal{P}$.** We first introduce this well-known lemma.

**Lemma 29.** *For any PSD matrix $\begin{pmatrix} \boldsymbol{A} & \boldsymbol{B} \\ \boldsymbol{B}^{\top} & \boldsymbol{C} \end{pmatrix} \succeq \boldsymbol{0}$, we have*

$$\begin{pmatrix} \boldsymbol{A} & \boldsymbol{B} \\ \boldsymbol{B}^{\top} & \boldsymbol{C} \end{pmatrix} \preceq 2 \begin{pmatrix} \boldsymbol{A} & \boldsymbol{0} \\ \boldsymbol{0} & \boldsymbol{C} \end{pmatrix}.$$

*Proof.* Note that the RHS minus the LHS is $\begin{pmatrix} \boldsymbol{A} & -\boldsymbol{B} \\ -\boldsymbol{B}^{\top} & \boldsymbol{C} \end{pmatrix}$. By Schur's Lemma, this matrix is PSD if and only if

$$\boldsymbol{A} - \boldsymbol{B} \boldsymbol{C}^{-1} \boldsymbol{B}^{\top} \succeq \boldsymbol{0},$$

which holds since $\begin{pmatrix} \boldsymbol{A} & \boldsymbol{B} \\ \boldsymbol{B}^{\top} & \boldsymbol{C} \end{pmatrix} \succeq \boldsymbol{0}$ because of Schur's Lemma. $\qquad\square$

**Lemma 30.** *There exist constants $b_1, b_2, b_3 > 0$ for which the following holds. Denote two index sets $\mathbb{J}, \mathbb{K}$ which satisfy $|\mathbb{J}| \leq b_1 n$, $r_{\mathbb{J}^c}(\boldsymbol{G}) \geq b_2 n$ and $|\mathbb{K}| \leq b_1 n$, $r_{\mathbb{K}^c}(\boldsymbol{H}) \geq b_2 n$. Then for any PSD matrix $\boldsymbol{J}$, with probability at least $1 - b_3 e^{-n/c}$,*

$$\langle \boldsymbol{J},\ \mathcal{P}^{(1)} \boldsymbol{G} \mathcal{P}^{(1)} \rangle \lesssim \left\langle \boldsymbol{J},\ \left( \frac{\operatorname{tr} \boldsymbol{G}_{\mathbb{J}^c}}{n} \right)^2 \boldsymbol{G}_{\mathbb{J}}^{-1} + \boldsymbol{G}_{\mathbb{J}^c} \right\rangle,$$

$$\langle \boldsymbol{J},\ \mathcal{P}^{(2)} \boldsymbol{H} \mathcal{P}^{(2)} \rangle \lesssim \left\langle \boldsymbol{J},\ \left( \frac{\operatorname{tr} \boldsymbol{H}_{\mathbb{K}^c}}{n} \right)^2 \boldsymbol{H}_{\mathbb{K}}^{-1} + \boldsymbol{H}_{\mathbb{K}^c} \right\rangle.$$

*Proof.* We will give the proof for the second task. Recall that according to Lemma 21,

$$\mathcal{P}^{(t)} := \begin{pmatrix} \mathcal{P}_{\mathbb{K}}^{(t)} & -\mathcal{Q}^{(t)} \\ -\mathcal{Q}^{(t)}{}^{\top} & \mathcal{P}_{\mathbb{K}^c}^{(t)} \end{pmatrix} = \begin{pmatrix} (\boldsymbol{I}_{\mathbb{K}} + \boldsymbol{X}_{\mathbb{K}}^{(t)}{}^{\top} \boldsymbol{A}_{\mathbb{K}^c}^{(t)}{}^{-1} \boldsymbol{X}_{\mathbb{K}}^{(t)})^{-1} & -\boldsymbol{X}_{\mathbb{K}}^{(t)}{}^{\top} \boldsymbol{A}^{(t)}{}^{-1} \boldsymbol{X}_{\mathbb{K}^c}^{(t)} \\ -\boldsymbol{X}_{\mathbb{K}^c}^{(t)}{}^{\top} \boldsymbol{A}^{(t)}{}^{-1} \boldsymbol{X}_{\mathbb{K}}^{(t)} & \boldsymbol{I}_{\mathbb{K}^c} - \boldsymbol{X}_{\mathbb{K}^c}^{(t)}{}^{\top} \boldsymbol{A}^{(t)}{}^{-1} \boldsymbol{X}_{\mathbb{K}^c}^{(t)} \end{pmatrix}.$$

As a result,

$$
\begin{aligned}
\mathcal{P}^{(2)} \boldsymbol{H} \mathcal{P}^{(2)} &= \begin{pmatrix} \mathcal{P}_{\mathbb{K}}^{(2)} \boldsymbol{H}_{\mathbb{K}} \mathcal{P}_{\mathbb{K}}^{(2)} + \mathcal{Q}^{(2)} \boldsymbol{H}_{\mathbb{K}^c} \mathcal{Q}^{(2)}{}^{\top} & -\mathcal{P}_{\mathbb{K}}^{(2)} \boldsymbol{H}_{\mathbb{K}} \mathcal{Q}^{(2)} - \mathcal{Q}^{(2)} \boldsymbol{H}_{\mathbb{K}^c} \mathcal{P}_{\mathbb{K}^c}^{(2)} \\ -\mathcal{P}_{\mathbb{K}^c}^{(2)} \boldsymbol{H}_{\mathbb{K}^c} \mathcal{Q}^{(2)}{}^{\top} - \mathcal{Q}^{(2)}{}^{\top} \boldsymbol{H}_{\mathbb{K}} \mathcal{P}_{\mathbb{K}^c}^{(2)} & \mathcal{P}_{\mathbb{K}^c}^{(2)} \boldsymbol{H}_{\mathbb{K}^c} \mathcal{P}_{\mathbb{K}^c}^{(2)} + \mathcal{Q}^{(2)}{}^{\top} \boldsymbol{H}_{\mathbb{K}} \mathcal{Q}^{(2)} \end{pmatrix} \\
&\preceq 2 \begin{pmatrix} \mathcal{P}_{\mathbb{K}}^{(2)} \boldsymbol{H}_{\mathbb{K}} \mathcal{P}_{\mathbb{K}}^{(2)} + \mathcal{Q}^{(2)} \boldsymbol{H}_{\mathbb{K}^c} \mathcal{Q}^{(2)}{}^{\top} & \boldsymbol{0} \\ \boldsymbol{0} & \mathcal{P}_{\mathbb{K}^c}^{(2)} \boldsymbol{H}_{\mathbb{K}^c} \mathcal{P}_{\mathbb{K}^c}^{(2)} + \mathcal{Q}^{(2)}{}^{\top} \boldsymbol{H}_{\mathbb{K}} \mathcal{Q}^{(2)} \end{pmatrix}.
\end{aligned}
$$

Therefore, in order to bound $\mathcal{P}^{(2)} \boldsymbol{H} \mathcal{P}^{(2)}$, we have four terms to consider:

- $\mathcal{P}_{\mathbb{K}}^{(2)} \boldsymbol{H}_{\mathbb{K}} \mathcal{P}_{\mathbb{K}}^{(2)}$: We note that

$$
\begin{aligned}
\mathcal{P}_{\mathbb{K}}^{(2)} \boldsymbol{H}_{\mathbb{K}} \mathcal{P}_{\mathbb{K}}^{(2)} &= \left(\boldsymbol{I}_{\mathbb{K}} + \boldsymbol{X}_{\mathbb{K}}^{(2)\top} \boldsymbol{A}_{\mathbb{K}^c}^{(2)^{-1}} \boldsymbol{X}_{\mathbb{K}}^{(2)}\right)^{-1} \boldsymbol{H}_{\mathbb{K}} \left(\boldsymbol{I}_{\mathbb{K}} + \boldsymbol{X}_{\mathbb{K}}^{(2)\top} \boldsymbol{A}_{\mathbb{K}^c}^{(2)^{-1}} \boldsymbol{X}_{\mathbb{K}}^{(2)}\right)^{-1} \\
&= \boldsymbol{H}_{\mathbb{K}}^{-1/2} \left(\boldsymbol{H}_{\mathbb{K}}^{-1} + \boldsymbol{Z}_{\mathbb{K}}^{(2)\top} \boldsymbol{A}_{\mathbb{K}^c}^{(2)^{-1}} \boldsymbol{Z}_{\mathbb{K}}^{(2)}\right)^{-2} \boldsymbol{H}_{\mathbb{K}}^{-1/2} \\
&\preceq \boldsymbol{H}_{\mathbb{K}}^{-1/2} \left(\boldsymbol{Z}_{\mathbb{K}}^{(2)\top} \boldsymbol{A}_{\mathbb{K}^c}^{(2)^{-1}} \boldsymbol{Z}_{\mathbb{K}}^{(2)}\right)^{-2} \boldsymbol{H}_{\mathbb{K}}^{-1/2} \\
&\precsim \frac{(\operatorname{tr} \boldsymbol{H}_{\mathbb{K}^c})^2}{n^2} \boldsymbol{H}_{\mathbb{K}}^{-1}.
\end{aligned}
$$

- $\mathcal{Q}^{(2)} \boldsymbol{H}_{\mathbb{K}^c} \mathcal{Q}^{(2)\top}$: Note that

$$
\begin{aligned}
\mathcal{Q}^{(2)} \boldsymbol{H}_{\mathbb{K}^c} \mathcal{Q}^{(2)\top} &= \boldsymbol{X}_{\mathbb{K}}^{(2)\top} \boldsymbol{A}^{(2)^{-1}} \boldsymbol{X}_{\mathbb{K}^c}^{(2)} \boldsymbol{H}_{\mathbb{K}^c} \boldsymbol{X}_{\mathbb{K}^c}^{(2)\top} \boldsymbol{A}^{(2)^{-1}} \boldsymbol{X}_{\mathbb{K}}^{(2)} \\
&= \boldsymbol{X}_{\mathbb{K}}^{(2)\top} \boldsymbol{A}^{(2)^{-1}} \boldsymbol{Z}_{\mathbb{K}^c}^{(2)} \boldsymbol{H}_{\mathbb{K}^c}^2 \boldsymbol{Z}_{\mathbb{K}^c}^{(2)\top} \boldsymbol{A}^{(2)^{-1}} \boldsymbol{X}_{\mathbb{K}}^{(2)}.
\end{aligned}
$$

Notice that according to Lemma 22,

$$
\begin{aligned}
\|\boldsymbol{Z}_{\mathbb{K}^c}^{(2)} \boldsymbol{H}_{\mathbb{K}^c}^2 \boldsymbol{Z}_{\mathbb{K}^c}^{(2)\top}\|_2 &\le c_1 (\operatorname{tr}(\boldsymbol{H}_{\mathbb{K}^c}^2) + n \|\boldsymbol{H}_{\mathbb{K}^c}^2\|_2) \\
&\le c_1 (\|\boldsymbol{H}_{\mathbb{K}^c}\|_2 \operatorname{tr}(\boldsymbol{H}_{\mathbb{K}^c}) + n \|\boldsymbol{H}_{\mathbb{K}^c}\|_2^2) \\
&\le \left(\frac{c_1}{b_2} + \frac{c_1}{b_2^2}\right) \frac{(\operatorname{tr} \boldsymbol{H}_{\mathbb{K}^c})^2}{n}
\end{aligned}
$$

Therefore,

$$
\begin{aligned}
\mathcal{Q}^{(2)} \boldsymbol{H}_{\mathbb{K}^c} \mathcal{Q}^{(2)\top} &= \boldsymbol{X}_{\mathbb{K}}^{(2)\top} \boldsymbol{A}^{(2)^{-1}} \boldsymbol{Z}_{\mathbb{K}^c}^{(2)} \boldsymbol{H}_{\mathbb{K}^c}^2 \boldsymbol{Z}_{\mathbb{K}^c}^{(2)\top} \boldsymbol{A}^{(2)^{-1}} \boldsymbol{X}_{\mathbb{K}}^{(2)} \\
&\precsim \frac{(\operatorname{tr} \boldsymbol{H}_{\mathbb{K}^c})^2}{n} \boldsymbol{X}_{\mathbb{K}}^{(2)\top} \boldsymbol{A}^{(2)^{-2}} \boldsymbol{X}_{\mathbb{K}}^{(2)} \\
&\preceq \frac{(\operatorname{tr} \boldsymbol{H}_{\mathbb{K}^c})^2}{n^2} \boldsymbol{H}_{\mathbb{K}}^{-1}.
\end{aligned}
$$

- $\mathcal{P}_{\mathbb{K}^c}^{(2)} \boldsymbol{H}_{\mathbb{K}^c} \mathcal{P}_{\mathbb{K}^c}^{(2)}$: Note that

$$
\begin{aligned}
\mathcal{P}_{\mathbb{K}^c}^{(2)} \boldsymbol{H}_{\mathbb{K}^c} \mathcal{P}_{\mathbb{K}^c}^{(2)} &= \left(\boldsymbol{I}_{\mathbb{K}^c} - \boldsymbol{X}_{\mathbb{K}^c}^{(2)\top} \boldsymbol{A}^{(2)^{-1}} \boldsymbol{X}_{\mathbb{K}^c}^{(2)}\right) \boldsymbol{H}_{\mathbb{K}^c} \left(\boldsymbol{I}_{\mathbb{K}^c} - \boldsymbol{X}_{\mathbb{K}^c}^{(2)\top} \boldsymbol{A}^{(2)^{-1}} \boldsymbol{X}_{\mathbb{K}^c}^{(2)}\right) \\
&\preceq 2 \cdot \left(\boldsymbol{H}_{\mathbb{K}^c} + \boldsymbol{X}_{\mathbb{K}^c}^{(2)\top} \boldsymbol{A}^{(2)^{-1}} \boldsymbol{X}_{\mathbb{K}^c}^{(2)} \boldsymbol{H}_{\mathbb{K}^c} \boldsymbol{X}_{\mathbb{K}^c}^{(2)\top} \boldsymbol{A}^{(2)^{-1}} \boldsymbol{X}_{\mathbb{K}^c}^{(2)}\right),
\end{aligned}
$$

and that

$$
\begin{aligned}
\boldsymbol{X}_{\mathbb{K}^c}^{(2)\top} \boldsymbol{A}^{(2)^{-1}} \boldsymbol{X}_{\mathbb{K}^c}^{(2)} \boldsymbol{H}_{\mathbb{K}^c} \boldsymbol{X}_{\mathbb{K}^c}^{(2)\top} \boldsymbol{A}^{(2)^{-1}} \boldsymbol{X}_{\mathbb{K}^c}^{(2)} &= \boldsymbol{X}_{\mathbb{K}^c}^{(2)\top} \boldsymbol{A}^{(2)^{-1}} \boldsymbol{Z}_{\mathbb{K}^c}^{(2)} \boldsymbol{H}_{\mathbb{K}^c}^2 \boldsymbol{Z}_{\mathbb{K}^c}^{(2)\top} \boldsymbol{A}^{(2)^{-1}} \boldsymbol{X}_{\mathbb{K}^c}^{(2)} \\
&\precsim \frac{(\operatorname{tr} \boldsymbol{H}_{\mathbb{K}^c})^2}{n} \boldsymbol{X}_{\mathbb{K}^c}^{(2)\top} \boldsymbol{A}^{(2)^{-2}} \boldsymbol{X}_{\mathbb{K}^c}^{(2)}
\end{aligned}
$$

Therefore, for any PSD matrix $\boldsymbol{J}$,

$$
\begin{aligned}
\langle \boldsymbol{J},\ \mathcal{P}_{\mathbb{K}^c}^{(2)} \boldsymbol{H}_{\mathbb{K}^c} \mathcal{P}_{\mathbb{K}^c}^{(2)} \rangle &\lesssim \langle \boldsymbol{J},\ \boldsymbol{H}_{\mathbb{K}^c} \rangle + \frac{(\operatorname{tr} \boldsymbol{H}_{\mathbb{K}^c})^2}{n} \langle \boldsymbol{J},\ \boldsymbol{X}_{\mathbb{K}^c}^{(2)\top} \boldsymbol{A}^{(2)^{-2}} \boldsymbol{X}_{\mathbb{K}^c}^{(2)} \rangle \\
&\lesssim \langle \boldsymbol{J},\ \boldsymbol{H}_{\mathbb{K}^c} \rangle.
\end{aligned}
$$

- $\mathcal{Q}^{(2)}{}^\top \boldsymbol{H}_{\mathbb{K}} \mathcal{Q}^{(2)}$: Note that

$$
\begin{aligned}
\mathcal{Q}^{(2)}{}^\top \boldsymbol{H}_{\mathbb{K}} \mathcal{Q}^{(2)} &= \boldsymbol{X}_{\mathbb{K}^c}^{(2)}{}^\top \boldsymbol{A}^{(2)}{}^{-1} \boldsymbol{X}_{\mathbb{K}}^{(2)} \boldsymbol{H}_{\mathbb{K}} \boldsymbol{X}_{\mathbb{K}}^{(2)}{}^\top \boldsymbol{A}^{(2)}{}^{-1} \boldsymbol{X}_{\mathbb{K}^c}^{(2)} \\
&= \boldsymbol{X}_{\mathbb{K}^c}^{(2)}{}^\top \boldsymbol{A}_{\mathbb{K}^c}^{(2)}{}^{-1} \boldsymbol{X}_{\mathbb{K}}^{(2)} \cdot \big(\boldsymbol{I}_{\mathbb{K}} + \boldsymbol{X}_{\mathbb{K}}^{(2)}{}^\top \boldsymbol{A}_{\mathbb{K}^c}^{(2)}{}^{-1} \boldsymbol{X}_{\mathbb{K}}^{(2)}\big)^{-1} \boldsymbol{H}_{\mathbb{K}} \big(\boldsymbol{I}_{\mathbb{K}} + \boldsymbol{X}_{\mathbb{K}}^{(2)}{}^\top \boldsymbol{A}_{\mathbb{K}^c}^{(2)}{}^{-1} \boldsymbol{X}_{\mathbb{K}}^{(2)}\big)^{-1} \\
&\quad\cdot \boldsymbol{X}_{\mathbb{K}}^{(2)}{}^\top \boldsymbol{A}_{\mathbb{K}^c}^{(2)}{}^{-1} \boldsymbol{X}_{\mathbb{K}^c}^{(2)} \\
&\precsim \frac{(\operatorname{tr} \boldsymbol{H}_{\mathbb{K}^c})^2}{n^2} \cdot \boldsymbol{X}_{\mathbb{K}^c}^{(2)}{}^\top \boldsymbol{A}_{\mathbb{K}^c}^{(2)}{}^{-1} \boldsymbol{X}_{\mathbb{K}}^{(2)} \cdot \boldsymbol{H}_{\mathbb{K}}^{-1} \cdot \boldsymbol{X}_{\mathbb{K}}^{(2)}{}^\top \boldsymbol{A}_{\mathbb{K}^c}^{(2)}{}^{-1} \boldsymbol{X}_{\mathbb{K}^c}^{(2)} \\
&= \frac{(\operatorname{tr} \boldsymbol{H}_{\mathbb{K}^c})^2}{n^2} \cdot \boldsymbol{X}_{\mathbb{K}^c}^{(2)}{}^\top \boldsymbol{A}_{\mathbb{K}^c}^{(2)}{}^{-1} \boldsymbol{Z}_{\mathbb{K}}^{(2)} \boldsymbol{Z}_{\mathbb{K}}^{(2)}{}^\top \boldsymbol{A}_{\mathbb{K}^c}^{(2)}{}^{-1} \boldsymbol{X}_{\mathbb{K}^c}^{(2)} \\
&\precsim \frac{(\operatorname{tr} \boldsymbol{H}_{\mathbb{K}^c})^2}{n} \cdot \boldsymbol{X}_{\mathbb{K}^c}^{(2)}{}^\top \boldsymbol{A}_{\mathbb{K}^c}^{(2)}{}^{-2} \boldsymbol{X}_{\mathbb{K}^c}^{(2)} \\
&\precsim \frac{1}{n} \cdot \boldsymbol{X}_{\mathbb{K}^c}^{(2)}{}^\top \boldsymbol{X}_{\mathbb{K}^c}^{(2)}.
\end{aligned}
$$

Therefore, for any PSD matrix $\boldsymbol{J}$,

$$
\begin{aligned}
\langle \boldsymbol{J}, \ \mathcal{Q}^{(2)}{}^\top \boldsymbol{H}_{\mathbb{K}} \mathcal{Q}^{(2)} \rangle &\lesssim \frac{1}{n} \langle \boldsymbol{J}, \ \boldsymbol{X}_{\mathbb{K}^c}^{(2)}{}^\top \boldsymbol{X}_{\mathbb{K}^c}^{(2)} \rangle \\
&\lesssim \frac{1}{n} \operatorname{tr}(\boldsymbol{Z}_{\mathbb{K}^c}^{(2)} \boldsymbol{H}_{\mathbb{K}^c}^{1/2} \boldsymbol{J} \boldsymbol{H}_{\mathbb{K}^c}^{1/2} \boldsymbol{Z}_{\mathbb{K}^c}^{(2)}{}^\top) \\
&\lesssim \operatorname{tr}(\boldsymbol{H}_{\mathbb{K}^c}^{1/2} \boldsymbol{J} \boldsymbol{H}_{\mathbb{K}^c}^{1/2}) \\
&= \langle \boldsymbol{J}, \ \boldsymbol{H}_{\mathbb{K}^c} \rangle.
\end{aligned}
$$

Combine these four terms and we get the result. $\qquad\square$

**Lemma 31.** *There exist constants $b_1, b_2, b_3 > 0$ for which the following holds. Denote two index sets $\mathbb{J}, \mathbb{K}$ which satisfy $|\mathbb{J}| \le b_1 n$, $r_{\mathbb{J}^c}(\boldsymbol{G}) \ge b_2 n$ and $|\mathbb{K}| \le b_1 n$, $r_{\mathbb{K}^c}(\boldsymbol{H}) \ge b_2 n$. Then for any PSD matrix $\boldsymbol{J}$, with probability at least $1 - b_3 e^{-n/c}$,*

$$
\langle \boldsymbol{J}, \ \mathcal{P}^{(2)} \boldsymbol{G} \mathcal{P}^{(2)} \rangle \lesssim \left\langle \boldsymbol{J}, \boldsymbol{G}_{\mathbb{K}^c} + \left( \|\boldsymbol{G}_{\mathbb{K}} \boldsymbol{H}_{\mathbb{K}}^{-1}\|_2 + \frac{n(\operatorname{tr}(\boldsymbol{G}_{\mathbb{K}^c} \boldsymbol{H}_{\mathbb{K}^c}) + n\|\boldsymbol{G}_{\mathbb{K}^c} \boldsymbol{H}_{\mathbb{K}^c}\|_2)}{(\operatorname{tr} \boldsymbol{H}_{\mathbb{K}^c})^2} \right) \cdot \left( \frac{(\operatorname{tr} \boldsymbol{H}_{\mathbb{K}^c})^2}{n^2} \boldsymbol{H}_{\mathbb{K}}^{-1} + \boldsymbol{H}_{\mathbb{K}^c} \right) \right\rangle.
$$

*Proof.* We will give the proof for the second task. Recall that according to Lemma 21,

$$
\mathcal{P}^{(t)} := \begin{pmatrix} \mathcal{P}_{\mathbb{K}}^{(t)} & -\mathcal{Q}^{(t)} \\ -\mathcal{Q}^{(t)}{}^\top & \mathcal{P}_{\mathbb{K}^c}^{(t)} \end{pmatrix} = \begin{pmatrix} \big(\boldsymbol{I}_{\mathbb{K}} + \boldsymbol{X}_{\mathbb{K}}^{(t)}{}^\top \boldsymbol{A}_{\mathbb{K}^c}^{(t)}{}^{-1} \boldsymbol{X}_{\mathbb{K}}^{(t)}\big)^{-1} & -\boldsymbol{X}_{\mathbb{K}}^{(t)}{}^\top \boldsymbol{A}^{(t)}{}^{-1} \boldsymbol{X}_{\mathbb{K}^c}^{(t)} \\ -\boldsymbol{X}_{\mathbb{K}^c}^{(t)}{}^\top \boldsymbol{A}^{(t)}{}^{-1} \boldsymbol{X}_{\mathbb{K}}^{(t)} & \boldsymbol{I}_{\mathbb{K}^c} - \boldsymbol{X}_{\mathbb{K}^c}^{(t)}{}^\top \boldsymbol{A}^{(t)}{}^{-1} \boldsymbol{X}_{\mathbb{K}^c}^{(t)} \end{pmatrix}.
$$

As a result,

$$
\begin{aligned}
\mathcal{P}^{(2)} \boldsymbol{G} \mathcal{P}^{(2)} &= \begin{pmatrix} \mathcal{P}_{\mathbb{K}}^{(2)} \boldsymbol{G}_{\mathbb{K}} \mathcal{P}_{\mathbb{K}}^{(2)} + \mathcal{Q}^{(2)} \boldsymbol{G}_{\mathbb{K}^c} \mathcal{Q}^{(2)}{}^\top & -\mathcal{P}_{\mathbb{K}}^{(2)} \boldsymbol{G}_{\mathbb{K}} \mathcal{Q}^{(2)} - \mathcal{Q}^{(2)} \boldsymbol{G}_{\mathbb{K}^c} \mathcal{P}_{\mathbb{K}^c}^{(2)} \\ -\mathcal{P}_{\mathbb{K}^c}^{(2)} \boldsymbol{G}_{\mathbb{K}^c} \mathcal{Q}^{(2)}{}^\top - \mathcal{Q}^{(2)}{}^\top \boldsymbol{G}_{\mathbb{K}} \mathcal{P}_{\mathbb{K}^c}^{(2)} & \mathcal{P}_{\mathbb{K}^c}^{(2)} \boldsymbol{G}_{\mathbb{K}^c} \mathcal{P}_{\mathbb{K}^c}^{(2)} + \mathcal{Q}^{(2)}{}^\top \boldsymbol{G}_{\mathbb{K}} \mathcal{Q}^{(2)} \end{pmatrix} \\
&\preceq 2 \begin{pmatrix} \mathcal{P}_{\mathbb{K}}^{(2)} \boldsymbol{G}_{\mathbb{K}} \mathcal{P}_{\mathbb{K}}^{(2)} + \mathcal{Q}^{(2)} \boldsymbol{G}_{\mathbb{K}^c} \mathcal{Q}^{(2)}{}^\top & \boldsymbol{0} \\ \boldsymbol{0} & \mathcal{P}_{\mathbb{K}^c}^{(2)} \boldsymbol{G}_{\mathbb{K}^c} \mathcal{P}_{\mathbb{K}^c}^{(2)} + \mathcal{Q}^{(2)}{}^\top \boldsymbol{G}_{\mathbb{K}} \mathcal{Q}^{(2)} \end{pmatrix}.
\end{aligned}
$$

Therefore, in order to bound $\mathcal{P}^{(2)} \boldsymbol{G} \mathcal{P}^{(2)}$, we have four terms to consider:

- $\mathcal{P}_{\mathbb{K}}^{(2)} \boldsymbol{G}_{\mathbb{K}} \mathcal{P}_{\mathbb{K}}^{(2)}$: We note that

$$
\begin{aligned}
&\mathcal{P}_{\mathbb{K}}^{(2)} \boldsymbol{G}_{\mathbb{K}} \mathcal{P}_{\mathbb{K}}^{(2)} \\
&= \big(\boldsymbol{I}_{\mathbb{K}} + \boldsymbol{X}_{\mathbb{K}}^{(2)}{}^\top \boldsymbol{A}_{\mathbb{K}^c}^{(2)}{}^{-1} \boldsymbol{X}_{\mathbb{K}}^{(2)}\big)^{-1} \boldsymbol{G}_{\mathbb{K}} \big(\boldsymbol{I}_{\mathbb{K}} + \boldsymbol{X}_{\mathbb{K}}^{(2)}{}^\top \boldsymbol{A}_{\mathbb{K}^c}^{(2)}{}^{-1} \boldsymbol{X}_{\mathbb{K}}^{(2)}\big)^{-1} \\
&= \boldsymbol{H}_{\mathbb{K}}^{-1/2} \big(\boldsymbol{H}_{\mathbb{K}}^{-1} + \boldsymbol{Z}_{\mathbb{K}}^{(2)}{}^\top \boldsymbol{A}_{\mathbb{K}^c}^{(2)}{}^{-1} \boldsymbol{Z}_{\mathbb{K}}^{(2)}\big)^{-1} \boldsymbol{H}_{\mathbb{K}}^{-1/2} \boldsymbol{G}_{\mathbb{K}} \boldsymbol{H}_{\mathbb{K}}^{-1/2} \big(\boldsymbol{H}_{\mathbb{K}}^{-1} + \boldsymbol{Z}_{\mathbb{K}}^{(2)}{}^\top \boldsymbol{A}_{\mathbb{K}^c}^{(2)}{}^{-1} \boldsymbol{Z}_{\mathbb{K}}^{(2)}\big)^{-1} \boldsymbol{H}_{\mathbb{K}}^{-1/2} \\
&\preceq \|\boldsymbol{H}_{\mathbb{K}}^{-1/2} \boldsymbol{G}_{\mathbb{K}} \boldsymbol{H}_{\mathbb{K}}^{-1/2}\|_2 \cdot \boldsymbol{H}_{\mathbb{K}}^{-1/2} \big(\boldsymbol{H}_{\mathbb{K}}^{-1} + \boldsymbol{Z}_{\mathbb{K}}^{(2)}{}^\top \boldsymbol{A}_{\mathbb{K}^c}^{(2)}{}^{-1} \boldsymbol{Z}_{\mathbb{K}}^{(2)}\big)^{-2} \boldsymbol{H}_{\mathbb{K}}^{-1/2} \\
&\preceq \|\boldsymbol{G}_{\mathbb{K}} \boldsymbol{H}_{\mathbb{K}}^{-1}\|_2 \cdot \boldsymbol{H}_{\mathbb{K}}^{-1/2} \big(\boldsymbol{Z}_{\mathbb{K}}^{(2)}{}^\top \boldsymbol{A}_{\mathbb{K}^c}^{(2)}{}^{-1} \boldsymbol{Z}_{\mathbb{K}}^{(2)}\big)^{-2} \boldsymbol{H}_{\mathbb{K}}^{-1/2} \\
&\precsim \|\boldsymbol{G}_{\mathbb{K}} \boldsymbol{H}_{\mathbb{K}}^{-1}\|_2 \cdot \frac{(\operatorname{tr} \boldsymbol{H}_{\mathbb{K}^c})^2}{n^2} \boldsymbol{H}_{\mathbb{K}}^{-1}.
\end{aligned}
$$

- $\mathcal{Q}^{(2)} \boldsymbol{G}_{\mathbb{K}^c} \mathcal{Q}^{(2)\top}$: Note that

$$
\begin{aligned}
\mathcal{Q}^{(2)} \boldsymbol{G}_{\mathbb{K}^c} \mathcal{Q}^{(2)\top} &= \boldsymbol{X}_{\mathbb{K}}^{(2)\top} \boldsymbol{A}^{(2)-1} \boldsymbol{X}_{\mathbb{K}^c}^{(2)} \boldsymbol{G}_{\mathbb{K}^c} \boldsymbol{X}_{\mathbb{K}^c}^{(2)\top} \boldsymbol{A}^{(2)-1} \boldsymbol{X}_{\mathbb{K}}^{(2)} \\
&= \boldsymbol{X}_{\mathbb{K}}^{(2)\top} \boldsymbol{A}^{(2)-1} \boldsymbol{Z}_{\mathbb{K}^c}^{(2)} \boldsymbol{H}_{\mathbb{K}^c}^{1/2} \boldsymbol{G}_{\mathbb{K}^c} \boldsymbol{H}_{\mathbb{K}^c}^{1/2} \boldsymbol{Z}_{\mathbb{K}^c}^{(2)\top} \boldsymbol{A}^{(2)-1} \boldsymbol{X}_{\mathbb{K}}^{(2)}.
\end{aligned}
$$

Notice that according to Lemma 22,

$$
\begin{aligned}
\boldsymbol{Z}_{\mathbb{K}^c}^{(2)} \boldsymbol{H}_{\mathbb{K}^c}^{1/2} \boldsymbol{G}_{\mathbb{K}^c} \boldsymbol{H}_{\mathbb{K}^c}^{1/2} \boldsymbol{Z}_{\mathbb{K}^c}^{(2)\top} &\le c_1(\operatorname{tr}(\boldsymbol{H}_{\mathbb{K}^c}^{1/2} \boldsymbol{G}_{\mathbb{K}^c} \boldsymbol{H}_{\mathbb{K}^c}^{1/2}) + n\|\boldsymbol{H}_{\mathbb{K}^c}^{1/2} \boldsymbol{G}_{\mathbb{K}^c} \boldsymbol{H}_{\mathbb{K}^c}^{1/2}\|_2) \\
&\le c_1(\operatorname{tr}(\boldsymbol{G}_{\mathbb{K}^c} \boldsymbol{H}_{\mathbb{K}^c}) + n\|\boldsymbol{G}_{\mathbb{K}^c} \boldsymbol{H}_{\mathbb{K}^c}\|_2).
\end{aligned}
$$

Therefore,

$$
\begin{aligned}
\mathcal{Q}^{(2)} \boldsymbol{H}_{\mathbb{K}^c} \mathcal{Q}^{(2)\top} &= \boldsymbol{X}_{\mathbb{K}}^{(2)\top} \boldsymbol{A}^{(2)-1} \boldsymbol{Z}_{\mathbb{K}^c}^{(2)} \boldsymbol{H}_{\mathbb{K}^c}^2 \boldsymbol{Z}_{\mathbb{K}^c}^{(2)\top} \boldsymbol{A}^{(2)-1} \boldsymbol{X}_{\mathbb{K}}^{(2)} \\
&\precsim (\operatorname{tr}(\boldsymbol{G}_{\mathbb{K}^c} \boldsymbol{H}_{\mathbb{K}^c}) + n\|\boldsymbol{G}_{\mathbb{K}^c} \boldsymbol{H}_{\mathbb{K}^c}\|_2) \boldsymbol{X}_{\mathbb{K}}^{(2)\top} \boldsymbol{A}^{(2)-2} \boldsymbol{X}_{\mathbb{K}}^{(2)} \\
&\precsim \frac{(\operatorname{tr}(\boldsymbol{G}_{\mathbb{K}^c} \boldsymbol{H}_{\mathbb{K}^c}) + n\|\boldsymbol{G}_{\mathbb{K}^c} \boldsymbol{H}_{\mathbb{K}^c}\|_2)}{n} \boldsymbol{H}_{\mathbb{K}}^{-1}.
\end{aligned}
$$

- $\mathcal{P}_{\mathbb{K}^c}^{(2)} \boldsymbol{G}_{\mathbb{K}^c} \mathcal{P}_{\mathbb{K}^c}^{(2)}$: Note that

$$
\begin{aligned}
\mathcal{P}_{\mathbb{K}^c}^{(2)} \boldsymbol{G}_{\mathbb{K}^c} \mathcal{P}_{\mathbb{K}^c}^{(2)} &= \big(\boldsymbol{I}_{\mathbb{K}^c} - \boldsymbol{X}_{\mathbb{K}^c}^{(2)\top} \boldsymbol{A}^{(2)-1} \boldsymbol{X}_{\mathbb{K}^c}^{(2)}\big) \boldsymbol{G}_{\mathbb{K}^c} \big(\boldsymbol{I}_{\mathbb{K}^c} - \boldsymbol{X}_{\mathbb{K}^c}^{(2)\top} \boldsymbol{A}^{(2)-1} \boldsymbol{X}_{\mathbb{K}^c}^{(2)}\big) \\
&\preceq 2 \cdot \big(\boldsymbol{G}_{\mathbb{K}^c} + \boldsymbol{X}_{\mathbb{K}^c}^{(2)\top} \boldsymbol{A}^{(2)-1} \boldsymbol{X}_{\mathbb{K}^c}^{(2)} \boldsymbol{G}_{\mathbb{K}^c} \boldsymbol{X}_{\mathbb{K}^c}^{(2)\top} \boldsymbol{A}^{(2)-1} \boldsymbol{X}_{\mathbb{K}^c}^{(2)}\big),
\end{aligned}
$$

and that

$$
\begin{aligned}
\boldsymbol{X}_{\mathbb{K}^c}^{(2)\top} \boldsymbol{A}^{(2)-1} \boldsymbol{X}_{\mathbb{K}^c}^{(2)} \boldsymbol{H}_{\mathbb{K}^c} \boldsymbol{X}_{\mathbb{K}^c}^{(2)\top} \boldsymbol{A}^{(2)-1} \boldsymbol{X}_{\mathbb{K}^c}^{(2)} &= \boldsymbol{X}_{\mathbb{K}^c}^{(2)\top} \boldsymbol{A}^{(2)-1} \boldsymbol{Z}_{\mathbb{K}^c}^{(2)} \boldsymbol{H}_{\mathbb{K}^c}^{1/2} \boldsymbol{G}_{\mathbb{K}^c} \boldsymbol{H}_{\mathbb{K}^c}^{1/2} \boldsymbol{Z}_{\mathbb{K}^c}^{(2)\top} \boldsymbol{A}^{(2)-1} \boldsymbol{X}_{\mathbb{K}^c}^{(2)} \\
&\precsim (\operatorname{tr}(\boldsymbol{G}_{\mathbb{K}^c} \boldsymbol{H}_{\mathbb{K}^c}) + n\|\boldsymbol{G}_{\mathbb{K}^c} \boldsymbol{H}_{\mathbb{K}^c}\|_2) \boldsymbol{X}_{\mathbb{K}^c}^{(2)\top} \boldsymbol{A}^{(2)-2} \boldsymbol{X}_{\mathbb{K}^c}^{(2)}
\end{aligned}
$$

Therefore, for any PSD matrix $\boldsymbol{J}$,

$$
\begin{aligned}
\langle \boldsymbol{J}, \ \mathcal{P}_{\mathbb{K}^c}^{(2)} \boldsymbol{G}_{\mathbb{K}^c} \mathcal{P}_{\mathbb{K}^c}^{(2)} \rangle &\lesssim \langle \boldsymbol{J}, \ \boldsymbol{G}_{\mathbb{K}^c} \rangle + (\operatorname{tr}(\boldsymbol{G}_{\mathbb{K}^c} \boldsymbol{H}_{\mathbb{K}^c}) + n\|\boldsymbol{G}_{\mathbb{K}^c} \boldsymbol{H}_{\mathbb{K}^c}\|_2)\langle \boldsymbol{J}, \ \boldsymbol{X}_{\mathbb{K}^c}^{(2)\top} \boldsymbol{A}^{(2)-2} \boldsymbol{X}_{\mathbb{K}^c}^{(2)} \rangle \\
&\lesssim \langle \boldsymbol{J}, \ \boldsymbol{G}_{\mathbb{K}^c} \rangle + (\operatorname{tr}(\boldsymbol{G}_{\mathbb{K}^c} \boldsymbol{H}_{\mathbb{K}^c}) + n\|\boldsymbol{G}_{\mathbb{K}^c} \boldsymbol{H}_{\mathbb{K}^c}\|_2) \cdot \frac{n}{(\operatorname{tr} \boldsymbol{H}_{\mathbb{K}^c})^2}\langle \boldsymbol{J}, \ \boldsymbol{H}_{\mathbb{K}^c} \rangle.
\end{aligned}
$$

- $\mathcal{Q}^{(2)\top} \boldsymbol{G}_{\mathbb{K}} \mathcal{Q}^{(2)}$: Note that

$$
\begin{aligned}
&\mathcal{Q}^{(2)\top} \boldsymbol{G}_{\mathbb{K}} \mathcal{Q}^{(2)} \\
&= \boldsymbol{X}_{\mathbb{K}^c}^{(2)\top} \boldsymbol{A}^{(2)-1} \boldsymbol{X}_{\mathbb{K}}^{(2)} \boldsymbol{G}_{\mathbb{K}} \boldsymbol{X}_{\mathbb{K}}^{(2)\top} \boldsymbol{A}^{(2)-1} \boldsymbol{X}_{\mathbb{K}^c}^{(2)} \\
&= \boldsymbol{X}_{\mathbb{K}^c}^{(2)\top} \boldsymbol{A}_{\mathbb{K}^c}^{(2)-1} \boldsymbol{X}_{\mathbb{K}}^{(2)} \cdot \big(\boldsymbol{I}_{\mathbb{K}} + \boldsymbol{X}_{\mathbb{K}}^{(2)\top} \boldsymbol{A}_{\mathbb{K}^c}^{(2)-1} \boldsymbol{X}_{\mathbb{K}}^{(2)}\big)^{-1} \boldsymbol{G}_{\mathbb{K}} \big(\boldsymbol{I}_{\mathbb{K}} + \boldsymbol{X}_{\mathbb{K}}^{(2)\top} \boldsymbol{A}_{\mathbb{K}^c}^{(2)-1} \boldsymbol{X}_{\mathbb{K}}^{(2)}\big)^{-1} \\
&\quad \cdot \boldsymbol{X}_{\mathbb{K}}^{(2)\top} \boldsymbol{A}_{\mathbb{K}^c}^{(2)-1} \boldsymbol{X}_{\mathbb{K}^c}^{(2)} \\
&\precsim \|\boldsymbol{G}_{\mathbb{K}} \boldsymbol{H}_{\mathbb{K}}^{-1}\|_2 \cdot \frac{(\operatorname{tr} \boldsymbol{H}_{\mathbb{K}^c})^2}{n^2} \cdot \boldsymbol{X}_{\mathbb{K}^c}^{(2)\top} \boldsymbol{A}_{\mathbb{K}^c}^{(2)-1} \boldsymbol{X}_{\mathbb{K}}^{(2)} \cdot \boldsymbol{H}_{\mathbb{K}}^{-1} \cdot \boldsymbol{X}_{\mathbb{K}}^{(2)\top} \boldsymbol{A}_{\mathbb{K}^c}^{(2)-1} \boldsymbol{X}_{\mathbb{K}^c}^{(2)} \\
&= \|\boldsymbol{G}_{\mathbb{K}} \boldsymbol{H}_{\mathbb{K}}^{-1}\|_2 \cdot \frac{(\operatorname{tr} \boldsymbol{H}_{\mathbb{K}^c})^2}{n^2} \cdot \boldsymbol{X}_{\mathbb{K}^c}^{(2)\top} \boldsymbol{A}_{\mathbb{K}^c}^{(2)-1} \boldsymbol{Z}_{\mathbb{K}}^{(2)} \boldsymbol{Z}_{\mathbb{K}}^{(2)\top} \boldsymbol{A}_{\mathbb{K}^c}^{(2)-1} \boldsymbol{X}_{\mathbb{K}^c}^{(2)} \\
&\precsim \|\boldsymbol{G}_{\mathbb{K}} \boldsymbol{H}_{\mathbb{K}}^{-1}\|_2 \cdot \frac{(\operatorname{tr} \boldsymbol{H}_{\mathbb{K}^c})^2}{n} \cdot \boldsymbol{X}_{\mathbb{K}^c}^{(2)\top} \boldsymbol{A}_{\mathbb{K}^c}^{(2)-2} \boldsymbol{X}_{\mathbb{K}^c}^{(2)} \\
&\precsim \|\boldsymbol{G}_{\mathbb{K}} \boldsymbol{H}_{\mathbb{K}}^{-1}\|_2 \cdot \frac{1}{n} \cdot \boldsymbol{X}_{\mathbb{K}^c}^{(2)\top} \boldsymbol{X}_{\mathbb{K}^c}^{(2)}.
\end{aligned}
$$

Therefore, for any PSD matrix $\boldsymbol{J}$,

$$
\begin{aligned}
\langle \boldsymbol{J}, \ \mathcal{Q}^{(2)\top} \boldsymbol{G}_{\mathbb{K}} \mathcal{Q}^{(2)} \rangle &\lesssim \|\boldsymbol{G}_{\mathbb{K}} \boldsymbol{H}_{\mathbb{K}}^{-1}\|_2 \cdot \frac{1}{n}\langle \boldsymbol{J}, \ \boldsymbol{X}_{\mathbb{K}^c}^{(2)\top} \boldsymbol{X}_{\mathbb{K}^c}^{(2)} \rangle \\
&\lesssim \|\boldsymbol{G}_{\mathbb{K}} \boldsymbol{H}_{\mathbb{K}}^{-1}\|_2 \cdot \frac{1}{n} \operatorname{tr}(\boldsymbol{Z}_{\mathbb{K}^c}^{(2)} \boldsymbol{H}_{\mathbb{K}^c}^{1/2} \boldsymbol{J} \boldsymbol{H}_{\mathbb{K}^c}^{1/2} \boldsymbol{Z}_{\mathbb{K}^c}^{(2)\top}) \\
&\lesssim \|\boldsymbol{G}_{\mathbb{K}} \boldsymbol{H}_{\mathbb{K}}^{-1}\|_2 \cdot \operatorname{tr}(\boldsymbol{H}_{\mathbb{K}^c}^{1/2} \boldsymbol{J} \boldsymbol{H}_{\mathbb{K}^c}^{1/2}) \\
&= \|\boldsymbol{G}_{\mathbb{K}} \boldsymbol{H}_{\mathbb{K}}^{-1}\|_2 \cdot \langle \boldsymbol{J}, \ \boldsymbol{H}_{\mathbb{K}^c} \rangle.
\end{aligned}
$$

Combine these four terms and we get the result. □

Now we can prove our main theorem.

*Proof of Theorem 19.* According to (23) and (24),

$$\mathbb{E}_\varepsilon[\mathcal{R}_1(\boldsymbol{w}^{(2)}) - \min \mathcal{R}_1] = \langle \boldsymbol{G}, \mathbb{E}_\varepsilon(\boldsymbol{w}^{(2)} - \boldsymbol{w}^*)(\boldsymbol{w}^{(2)} - \boldsymbol{w}^*)^\top \rangle$$

$$= \langle \boldsymbol{G}, \mathcal{P}^{(2)}\mathbb{E}_\varepsilon(\boldsymbol{w}^{(1)} - \boldsymbol{w}^*)(\boldsymbol{w}^{(1)} - \boldsymbol{w}^*)^\top \mathcal{P}^{(2)} \rangle + \sigma^2 \langle \boldsymbol{G}, \boldsymbol{X}^{(2)^\top}\boldsymbol{A}^{(2)^{-2}}\boldsymbol{X}^{(2)} \rangle$$

$$= \langle \boldsymbol{G}, \mathcal{P}^{(2)}\mathcal{P}^{(1)}\boldsymbol{w}^*\boldsymbol{w}^{*\top}\mathcal{P}^{(1)}\mathcal{P}^{(2)} \rangle$$

$$\quad + \sigma^2\big(\langle \boldsymbol{G}, \mathcal{P}^{(2)}\boldsymbol{X}^{(1)^\top}\boldsymbol{A}^{(1)^{-2}}\boldsymbol{X}^{(1)}\mathcal{P}^{(2)} \rangle + \langle \boldsymbol{G}, \boldsymbol{X}^{(2)^\top}\boldsymbol{A}^{(2)^{-2}}\boldsymbol{X}^{(2)} \rangle\big)$$

$$= \mathtt{bias} + \mathtt{variance}.$$

For the bias part, with probability at least $1 - b_3 e^{-n/c}$,

$$\mathtt{bias} = \langle \boldsymbol{G}, \mathcal{P}^{(2)}\mathcal{P}^{(1)}\boldsymbol{w}^*\boldsymbol{w}^{*\top}\mathcal{P}^{(1)}\mathcal{P}^{(2)} \rangle$$

$$\leq \langle \mathcal{P}^{(1)}\boldsymbol{G}\mathcal{P}^{(1)}, \boldsymbol{w}^*\boldsymbol{w}^{*\top} \rangle$$

$$\lesssim \left\langle \left(\frac{\operatorname{tr}\boldsymbol{G}_{\mathbb{J}^c}}{n}\right)^2 \boldsymbol{G}_{\mathbb{J}}^{-1} + \boldsymbol{G}_{\mathbb{J}^c}, \boldsymbol{w}^*\boldsymbol{w}^{*\top} \right\rangle$$

$$= \frac{(\operatorname{tr}\boldsymbol{G}_{\mathbb{J}^c})^2}{n^2}\|\boldsymbol{w}^*\|^2_{\boldsymbol{G}_{\mathbb{J}}^{-1}} + \|\boldsymbol{w}^*\|^2_{\boldsymbol{G}_{\mathbb{J}^c}}.$$

For the variance part, with probability at least $1 - b_3 e^{-n/c}$,

$$\mathtt{variance} = \sigma^2\big(\langle \mathcal{P}^{(2)}\boldsymbol{G}\mathcal{P}^{(2)}, \boldsymbol{X}^{(1)^\top}\boldsymbol{A}^{(1)^{-2}}\boldsymbol{X}^{(1)} \rangle + \langle \boldsymbol{G}, \boldsymbol{X}^{(2)^\top}\boldsymbol{A}^{(2)^{-2}}\boldsymbol{X}^{(2)} \rangle\big)$$

$$\lesssim \sigma^2\bigg(\bigg\langle \boldsymbol{G}_{\mathbb{K}^c} + \bigg(\|\boldsymbol{G}_{\mathbb{K}}\boldsymbol{H}_{\mathbb{K}}^{-1}\|_2 + \frac{n(\operatorname{tr}(\boldsymbol{G}_{\mathbb{K}^c}\boldsymbol{H}_{\mathbb{K}^c}) + n\|\boldsymbol{G}_{\mathbb{K}^c}\boldsymbol{H}_{\mathbb{K}^c}\|_2)}{(\operatorname{tr}\boldsymbol{H}_{\mathbb{K}^c})^2}\bigg) \cdot \bigg(\frac{(\operatorname{tr}\boldsymbol{H}_{\mathbb{K}^c})^2}{n^2}\boldsymbol{H}_{\mathbb{K}}^{-1} + \boldsymbol{H}_{\mathbb{K}^c}\bigg),$$

$$\frac{1}{n}\boldsymbol{G}_{\mathbb{J}}^{-1} + \frac{n}{(\operatorname{tr}\boldsymbol{G}_{\mathbb{J}^c})^2}\cdot\boldsymbol{G}_{\mathbb{J}^c} \bigg\rangle + \bigg\langle \boldsymbol{G}, \frac{1}{n}\boldsymbol{H}_{\mathbb{K}}^{-1} + \frac{n}{(\operatorname{tr}\boldsymbol{H}_{\mathbb{K}^c})^2}\cdot\boldsymbol{H}_{\mathbb{K}^c} \bigg\rangle\bigg)$$

$$= \frac{\sigma^2}{n}\bigg(\bigg(\|\boldsymbol{G}_{\mathbb{K}}\boldsymbol{H}_{\mathbb{K}}^{-1}\|_2 + \frac{n((\operatorname{tr}\boldsymbol{G}_{\mathbb{K}^c}\boldsymbol{H}_{\mathbb{K}^c}) + n\|\boldsymbol{G}_{\mathbb{K}^c}\boldsymbol{H}_{\mathbb{K}^c}\|_2)}{(\operatorname{tr}\boldsymbol{H}_{\mathbb{K}^c})^2}\bigg)\cdot\bigg\langle\bigg(\frac{(\operatorname{tr}\boldsymbol{H}_{\mathbb{K}^c})^2}{n^2}\boldsymbol{H}_{\mathbb{K}}^{-1} + \boldsymbol{H}_{\mathbb{K}^c}\bigg),$$

$$\boldsymbol{G}_{\mathbb{J}}^{-1} + \frac{n^2}{(\operatorname{tr}\boldsymbol{G}_{\mathbb{J}^c})^2}\cdot\boldsymbol{G}_{\mathbb{J}^c} \bigg\rangle + \bigg\langle \boldsymbol{G}_{\mathbb{J}} + \boldsymbol{G}_{\mathbb{J}^c}, \boldsymbol{H}_{\mathbb{K}}^{-1} + \frac{n^2}{(\operatorname{tr}\boldsymbol{H}_{\mathbb{K}^c})^2}\cdot\boldsymbol{H}_{\mathbb{K}^c} \bigg\rangle + |\mathbb{J}\cap\mathbb{K}| + n^2\frac{\operatorname{tr}(\boldsymbol{G}_{\mathbb{J}^c\cap\mathbb{K}^c}^2)}{(\operatorname{tr}\boldsymbol{G}_{\mathbb{J}^c})^2}\bigg)$$

$$\leq \frac{\sigma^2}{n}\bigg(\bigg(\|\boldsymbol{G}_{\mathbb{K}}\boldsymbol{H}_{\mathbb{K}}^{-1}\|_2 + \frac{n(\operatorname{tr}(\boldsymbol{G}_{\mathbb{K}^c}\boldsymbol{H}_{\mathbb{K}^c}) + n\|\boldsymbol{G}_{\mathbb{K}^c}\boldsymbol{H}_{\mathbb{K}^c}\|_2)}{(\operatorname{tr}\boldsymbol{H}_{\mathbb{K}^c})^2} + \frac{(\operatorname{tr}\boldsymbol{G}_{\mathbb{J}^c})^2}{(\operatorname{tr}\boldsymbol{H}_{\mathbb{K}^c})^2}\bigg)$$

$$\cdot\bigg\langle \frac{(\operatorname{tr}\boldsymbol{H}_{\mathbb{K}^c})^2}{n^2}\boldsymbol{H}_{\mathbb{K}}^{-1} + \boldsymbol{H}_{\mathbb{K}^c}, \boldsymbol{G}_{\mathbb{J}}^{-1} + \frac{n^2}{(\operatorname{tr}\boldsymbol{G}_{\mathbb{J}^c})^2}\cdot\boldsymbol{G}_{\mathbb{J}^c} \bigg\rangle$$

$$+ \bigg\langle \boldsymbol{G}_{\mathbb{J}}, \boldsymbol{H}_{\mathbb{K}}^{-1} + \frac{n^2}{(\operatorname{tr}\boldsymbol{H}_{\mathbb{K}^c})^2}\cdot\boldsymbol{H}_{\mathbb{K}^c} \bigg\rangle + |\mathbb{J}\cap\mathbb{K}| + n^2\frac{\operatorname{tr}(\boldsymbol{G}_{\mathbb{J}^c\cap\mathbb{K}^c}^2)}{(\operatorname{tr}\boldsymbol{G}_{\mathbb{J}^c})^2}\bigg)$$

$$= \frac{\sigma^2}{n}\bigg(\operatorname{tr}(\boldsymbol{G}_{\mathbb{J}}\boldsymbol{H}_{\mathbb{K}}^{-1}) + n^2\frac{\operatorname{tr}(\boldsymbol{G}_{\mathbb{J}}\boldsymbol{H}_{\mathbb{K}^c})}{(\operatorname{tr}\boldsymbol{H}_{\mathbb{K}^c})^2} + |\mathbb{J}\cap\mathbb{K}| + n^2\frac{\operatorname{tr}(\boldsymbol{G}_{\mathbb{J}^c\cap\mathbb{K}^c}^2)}{(\operatorname{tr}\boldsymbol{G}_{\mathbb{J}^c})^2}$$

$$+ \bigg(\|\boldsymbol{G}_{\mathbb{K}}\boldsymbol{H}_{\mathbb{K}}^{-1}\|_2 + \frac{n(\operatorname{tr}(\boldsymbol{G}_{\mathbb{K}^c}\boldsymbol{H}_{\mathbb{K}^c}) + n\|\boldsymbol{G}_{\mathbb{K}^c}\boldsymbol{H}_{\mathbb{K}^c}\|_2)}{(\operatorname{tr}\boldsymbol{H}_{\mathbb{K}^c})^2} + \frac{(\operatorname{tr}\boldsymbol{G}_{\mathbb{J}^c})^2}{(\operatorname{tr}\boldsymbol{H}_{\mathbb{K}^c})^2}\bigg)$$

$$\cdot\bigg\langle \frac{(\operatorname{tr}\boldsymbol{H}_{\mathbb{K}^c})^2}{n^2}\boldsymbol{H}_{\mathbb{K}}^{-1} + \boldsymbol{H}_{\mathbb{K}^c}, \boldsymbol{G}_{\mathbb{J}}^{-1} + \frac{n^2}{(\operatorname{tr}\boldsymbol{G}_{\mathbb{J}^c})^2}\cdot\boldsymbol{G}_{\mathbb{J}^c} \bigg\rangle\bigg)$$

We have finished the proof. □

## C.6 Lower Bounds

**Lemma 32.** *There exist constants $b_1, b_2, c > 0$ for which the following holds. Denote two index sets $\mathbb{J}, \mathbb{K}$ which satisfy $|\mathbb{J}| \leq b_1 n$, $r_{\mathbb{J}^c}(\boldsymbol{G}) \geq b_2 n$ and $|\mathbb{K}| \leq b_1 n$, $r_{\mathbb{K}^c}(\boldsymbol{H}) \geq b_2 n$. Then, in expectation*

to $\boldsymbol{X}^{(2)}$, $\boldsymbol{X}^{(2)\top}\boldsymbol{A}^{(2)^{-2}}\boldsymbol{X}^{(2)}$ is a diagonal matrix. In particular, for every $n > c$,

$$(\mathbb{E}\boldsymbol{X}^{(1)\top}\boldsymbol{A}^{(1)^{-2}}\boldsymbol{X}^{(1)})_{ii} \gtrsim \frac{\mu_i}{n} \cdot \big(\frac{\operatorname{tr}\boldsymbol{G}_{\mathbb{J}^c}}{n} + \mu_i\big)^{-2},$$

$$(\mathbb{E}\boldsymbol{X}^{(2)\top}\boldsymbol{A}^{(2)^{-2}}\boldsymbol{X}^{(2)})_{ii} \gtrsim \frac{\lambda_i}{n} \cdot \big(\frac{\operatorname{tr}\boldsymbol{H}_{\mathbb{K}^c}}{n} + \lambda_i\big)^{-2}.$$

*Proof.* Recall that $\boldsymbol{X}^{(2)} = \boldsymbol{Z}^{(2)}\boldsymbol{H}^{(2)^{1/2}}$. Therefore,

$$\boldsymbol{X}^{(2)\top}\boldsymbol{A}^{(2)^{-2}}\boldsymbol{X}^{(2)} = \boldsymbol{H}^{(2)^{1/2}}\boldsymbol{Z}^{(2)\top}\boldsymbol{A}^{(2)^{-2}}\boldsymbol{Z}^{(2)}\boldsymbol{H}^{(2)^{1/2}}.$$

Consider the off-diagonal terms of $\boldsymbol{Z}^{(2)\top}\boldsymbol{A}^{(2)^{-2}}\boldsymbol{Z}^{(2)}$:

$$\begin{aligned}(\mathbb{E}\boldsymbol{Z}^{(2)\top}\boldsymbol{A}^{(2)^{-2}}\boldsymbol{Z}^{(2)})_{ij} &= \boldsymbol{e}_i^\top \boldsymbol{Z}^{(2)\top}\boldsymbol{A}^{(2)^{-2}}\boldsymbol{Z}^{(2)}\boldsymbol{e}_j \\ &= \boldsymbol{z}_i^{(2)\top}\boldsymbol{A}^{(2)^{-2}}\boldsymbol{z}_j^{(2)} \\ &= \boldsymbol{z}_i^{(2)\top}\sum_l \lambda_l \boldsymbol{z}_l^{(2)}(\boldsymbol{z}_l^{(2)})^{\top^{-2}}\boldsymbol{z}_j^{(2)} =: f(\boldsymbol{z}_i^{(2)}).\end{aligned}$$

Observe that $f(-\boldsymbol{z}_i^{(2)}) = -f(\boldsymbol{z}_i^{(2)})$ and that $\boldsymbol{z}_i^{(2)}$ follows the standard Gaussian distribution, which is symmetric across 0. Therefore $\mathbb{E}f(\boldsymbol{z}_i^{(2)}) = 0$, which kills all off-diagonal terms of $\boldsymbol{Z}^{(2)\top}\boldsymbol{A}^{(2)^{-2}}\boldsymbol{Z}^{(2)}$.

For the diagonal terms, applying Lemma 20 on the single-element index set $\{i\}$ and we get

$$\begin{aligned}(\boldsymbol{Z}^{(2)\top}\boldsymbol{A}^{(2)^{-2}}\boldsymbol{Z}^{(2)})_{ii} &= \boldsymbol{e}_i^\top \boldsymbol{Z}^{(2)\top}\boldsymbol{A}^{(2)^{-2}}\boldsymbol{Z}^{(2)}\boldsymbol{e}_i \\ &= \boldsymbol{z}_i^{(2)\top}\boldsymbol{A}^{(2)^{-2}}\boldsymbol{z}_i^{(2)} \\ &= \lambda_i^{-1}\cdot(\lambda_i^{-1} + \boldsymbol{z}_i^{(2)\top}\boldsymbol{A}_{-i}^{(2)^{-1}}\boldsymbol{z}_i^{(2)})^{-1}\boldsymbol{z}_i^{(2)\top}\boldsymbol{A}_{-i}^{(2)^{-2}}\boldsymbol{z}_i^{(2)}(\lambda_i^{-1} + \boldsymbol{z}_i^{(2)\top}\boldsymbol{A}_{-i}^{(2)^{-1}}\boldsymbol{z}_i^{(2)})^{-1}. \\ &= \frac{\boldsymbol{z}_i^{(2)\top}\boldsymbol{A}_{-i}^{(2)^{-2}}\boldsymbol{z}_i^{(2)}}{(1 + \lambda_i \boldsymbol{z}_i^{(2)\top}\boldsymbol{A}_{-i}^{(2)^{-1}}\boldsymbol{z}_i^{(2)})^2}.\end{aligned}$$

According to Cauchy-Schwarz, we have

$$\|\boldsymbol{z}_i^{(2)}\|_2^2 \cdot \boldsymbol{z}_i^{(2)\top}\boldsymbol{A}_{-i}^{(2)^{-2}}\boldsymbol{z}_i^{(2)} \geq (\boldsymbol{z}_i^{(2)\top}\boldsymbol{A}_{-i}^{(2)^{-1}}\boldsymbol{z}_i^{(2)})^2.$$

Recall that $\|\boldsymbol{z}_i^{(2)}\|_2^2 = \boldsymbol{z}_i^{(2)\top}\boldsymbol{z}_i^{(2)} \asymp n$ with probability $1 - 2\exp(-n/c)$ according to Lemma 25. Therefore with probability $1 - 2\exp(-n/c)$,

$$\begin{aligned}(\boldsymbol{Z}^{(2)\top}\boldsymbol{A}^{(2)^{-2}}\boldsymbol{Z}^{(2)})_{ii} &\geq \frac{(\boldsymbol{z}_i^{(2)\top}\boldsymbol{A}_{-i}^{(2)^{-1}}\boldsymbol{z}_i^{(2)})^2}{n(1 + \lambda_i \boldsymbol{z}_i^{(2)\top}\boldsymbol{A}_{-i}^{(2)^{-1}}\boldsymbol{z}_i^{(2)})^2} \\ &= \frac{1}{n}\cdot\big(\frac{1}{\boldsymbol{z}_i^{(2)\top}\boldsymbol{A}_{-i}^{(2)^{-1}}\boldsymbol{z}_i^{(2)}} + \lambda_i\big)^{-2}.\end{aligned}$$

$\square$

Now we examine $\boldsymbol{z}_i^{(2)\top}\boldsymbol{A}_{-i}^{(2)^{-1}}\boldsymbol{z}_i^{(2)}$. Let $\mathcal{L}$ be the top $(n-k)$ dimension subspace of $\boldsymbol{A}_{-i}^{(2)^{-1}}$. Then with probability at least $1 - 4\exp(-n/c)$,

$$\begin{aligned}\boldsymbol{z}_i^{(2)\top}\boldsymbol{A}_{-i}^{(2)^{-1}}\boldsymbol{z}_i^{(2)} &\geq \|\Pi_{\mathcal{L}}\boldsymbol{z}_i^{(2)}\|_2^2 \cdot \mu_{n-k}(\boldsymbol{A}_{-i}^{(2)^{-1}}) \\ &\gtrsim (n-k)\cdot\frac{1}{\mu_{k+1}(\boldsymbol{A}_{-i}^{(2)})} \\ &\gtrsim \frac{n}{\operatorname{tr}\boldsymbol{H}_{\mathbb{K}^c}}.\end{aligned}\qquad(25)$$

Since $(\frac{1}{x} + \lambda_i)^{-2}$ is an increasing function, now we know that with probability at least $1 - 6 \exp(-n/c)$,

$$(\boldsymbol{Z}^{(2)^\top} \boldsymbol{A}^{(2)^{-2}} \boldsymbol{Z}^{(2)})_{ii} \gtrsim \frac{1}{n} \cdot \big( \frac{\operatorname{tr} \boldsymbol{H}_{\mathbb{K}^c}}{n} + \lambda_i \big)^{-2}.$$

Since $(\boldsymbol{Z}^{(2)^\top} \boldsymbol{A}^{(2)^{-2}} \boldsymbol{Z}^{(2)})_{ii} \geq 0$, by taking expectation and multiply by $\lambda_i$ on each side we get the result.

**Lemma 33.** *There exist constants $b_1, b_2, c > 0$ for which the following holds. Denote two index sets $\mathbb{J}, \mathbb{K}$ which satisfy $|\mathbb{J}| \leq b_1 n$, $r_{\mathbb{J}^c}(\boldsymbol{G}) \geq b_2 n$ and $|\mathbb{K}| \leq b_1 n$, $r_{\mathbb{K}^c}(\boldsymbol{H}) \geq b_2 n$. Then for any PSD diagonal matrix $\boldsymbol{J}$, for every $n > c$,*

$$\mathbb{E}(\mathcal{P}^{(1)} \boldsymbol{J} \mathcal{P}^{(1)}) \succsim \boldsymbol{J} \cdot \left( \frac{(\operatorname{tr} \boldsymbol{G}_{\mathbb{J}^c})^2}{n^2} \boldsymbol{G}_{\mathbb{J}}^{-2} + \boldsymbol{I}_{\mathbb{J}^c} \right),$$

$$\mathbb{E}(\mathcal{P}^{(2)} \boldsymbol{J} \mathcal{P}^{(2)}) \succsim \boldsymbol{J} \cdot \left( \frac{(\operatorname{tr} \boldsymbol{H}_{\mathbb{K}^c})^2}{n^2} \boldsymbol{H}_{\mathbb{K}}^{-2} + \boldsymbol{I}_{\mathbb{K}^c} \right).$$

*Proof.* Suppose $\boldsymbol{J} = \operatorname{diag}(\phi_i)_{i=1}^d$. We examine all diagonal elements of $\mathcal{P}^{(2)} \boldsymbol{J} \mathcal{P}^{(2)}$.

$$(\mathcal{P}^{(2)} \boldsymbol{J} \mathcal{P}^{(2)})_{ii} = \sum_j \phi_i \mathcal{P}_{ij}^{(2)^2} \geq \phi_i \mathcal{P}_{ii}^{(2)^2}.$$

As a result, we only need to analyze the lower bound of $\mathcal{P}_{ii}^{(2)}$. According to Lemma 21,

$$\mathcal{P}_{ii}^{(2)} = (1 + \boldsymbol{x}_i^{(2)^\top} \boldsymbol{A}_{-i}^{(2)^{-1}} \boldsymbol{x}_i^{(2)})^{-1}$$
$$= (1 + \lambda_i \boldsymbol{z}_i^{(2)^\top} \boldsymbol{A}_{-i}^{(2)^{-1}} \boldsymbol{x}_i^{(2)})^{-1}.$$

Recall that in (25) we have that with probability at least $1 - 4 \exp(-n/c)$,

$$\boldsymbol{z}_i^{(2)^\top} \boldsymbol{A}_{-i}^{(2)^{-1}} \boldsymbol{x}_i^{(2)} \gtrsim \frac{n}{\operatorname{tr} \boldsymbol{H}_{\mathbb{K}^c}}.$$

Therefore, with probability at least $1 - 4 \exp(-n/c)$,

$$(\mathcal{P}^{(2)} \boldsymbol{J} \mathcal{P}^{(2)})_{ii} \gtrsim \phi_i \cdot (1 + \frac{n \lambda_i}{\operatorname{tr} \boldsymbol{H}_{\mathbb{K}^c}})^{-2}$$
$$\gtrsim \phi_i \cdot \big( 1 + \frac{(\operatorname{tr} \boldsymbol{H}_{\mathbb{K}^c})^2}{n^2 \lambda_i^2} \big).$$

Since $(\mathcal{P}^{(2)} \boldsymbol{J} \mathcal{P}^{(2)})_{ii} \geq 0$, by taking expectation we get the result. $\square$

**Theorem 34.** *There exist constants $b_1, b_2, c > 0$ for which the following holds. Denote index sets $\mathbb{J}, \mathbb{K}$ that are defined as follows. Let $\mathbb{J}_\mu = i : \mu_i \geq \mu$. Let $\mu^* = \max\{\mu : r_{\mathbb{J}_\mu^c}(\boldsymbol{G}) \geq b_2 n\}$, and define $\mathbb{J} := \mathbb{J}_{\mu^*}$. Similarly, let $\mathbb{K}_\lambda = i : \lambda_i \geq \lambda$, $\lambda^* = \max\{\lambda : r_{\mathbb{K}_\lambda^c}(\boldsymbol{H}) \geq b_2 n\}$, and define $\mathbb{K} := \mathbb{K}_{\lambda^*}$. Then if $|\mathbb{J}| \leq b_1 n$ and $|\mathbb{K}| \leq b_1 n$, for every $n > c$,*

$$\mathbb{E}_\varepsilon[\mathcal{R}_1(\boldsymbol{w}^{(2)}) - \min \mathcal{R}_1] \gtrsim \frac{\sigma^2}{n} \left( \Big\langle \frac{(\operatorname{tr} \boldsymbol{H}_{\mathbb{K}^c})^2}{n^2} \boldsymbol{H}_{\mathbb{K}}^{-2} + \boldsymbol{I}_{\mathbb{K}^c}, \, \boldsymbol{I}_{\mathbb{J}} + \frac{n^2}{(\operatorname{tr} \boldsymbol{G}_{\mathbb{J}^c})^2} \cdot \boldsymbol{G}_{\mathbb{J}^c}^2 \Big\rangle + \Big\langle \boldsymbol{G}, \, \boldsymbol{H}_{\mathbb{K}}^{-1} + \frac{n^2}{(\operatorname{tr} \boldsymbol{H}_{\mathbb{K}^c})^2} \cdot \boldsymbol{H}_{\mathbb{K}^c} \Big\rangle \right).$$

*Proof.* According to (23) and (24),

$$\mathbb{E}_\varepsilon[\mathcal{R}_1(\boldsymbol{w}^{(2)}) - \min \mathcal{R}_1] = \langle \boldsymbol{G}, \, \mathbb{E}_\varepsilon(\boldsymbol{w}^{(2)} - \boldsymbol{w}^*)(\boldsymbol{w}^{(2)} - \boldsymbol{w}^*)^\top \rangle$$
$$= \langle \boldsymbol{G}, \, \mathcal{P}^{(2)} \mathbb{E}_\varepsilon(\boldsymbol{w}^{(1)} - \boldsymbol{w}^*)(\boldsymbol{w}^{(1)} - \boldsymbol{w}^*)^\top \mathcal{P}^{(2)} \rangle + \sigma^2 \langle \boldsymbol{G}, \, \boldsymbol{X}^{(2)^\top} \boldsymbol{A}^{(2)^{-2}} \boldsymbol{X}^{(2)} \rangle$$
$$= \langle \boldsymbol{G}, \, \mathcal{P}^{(2)} \mathcal{P}^{(1)} \boldsymbol{w}^* \boldsymbol{w}^{*\top} \mathcal{P}^{(1)} \mathcal{P}^{(2)} \rangle$$
$$\quad + \sigma^2 \big( \langle \boldsymbol{G}, \, \mathcal{P}^{(2)} \boldsymbol{X}^{(1)^\top} \boldsymbol{A}^{(1)^{-2}} \boldsymbol{X}^{(1)} \mathcal{P}^{(2)} \rangle + \langle \boldsymbol{G}, \, \boldsymbol{X}^{(2)^\top} \boldsymbol{A}^{(2)^{-2}} \boldsymbol{X}^{(2)} \rangle \big)$$
$$= \texttt{bias} + \texttt{variance}.$$

We give the bias lower bound in the expectation form. For the bias part,

$$
\begin{aligned}
\mathbb{E}\texttt{bias} &= \mathbb{E}\langle \boldsymbol{G},\ \mathcal{P}^{(2)}\mathcal{P}^{(1)}\boldsymbol{w}^*\boldsymbol{w}^{*\top}\mathcal{P}^{(1)}\mathcal{P}^{(2)}\rangle \\
&= \mathbb{E}_{\boldsymbol{X}^{(1)}}\langle \mathbb{E}_{\boldsymbol{X}^{(2)}}\mathcal{P}^{(2)}\boldsymbol{G}\mathcal{P}^{(2)},\ \mathcal{P}^{(1)}\boldsymbol{w}^*\boldsymbol{w}^{*\top}\mathcal{P}^{(1)}\rangle \\
&\gtrsim \left\langle \mathbb{E}_{\boldsymbol{X}^{(1)}}\mathcal{P}^{(1)}\left(\frac{(\operatorname{tr}\boldsymbol{H}_{\mathbb{K}^c})^2}{n^2}\boldsymbol{H}_{\mathbb{K}}^{-2}+\boldsymbol{I}_{\mathbb{K}^c}\right)\mathcal{P}^{(1)},\ \boldsymbol{w}^*\boldsymbol{w}^{*\top}\right\rangle \\
&\gtrsim \left\langle \left(\frac{(\operatorname{tr}\boldsymbol{G}_{\mathbb{J}^c})^2}{n^2}\boldsymbol{G}_{\mathbb{J}}^{-2}+\boldsymbol{I}_{\mathbb{J}^c}\right)\cdot\left(\frac{(\operatorname{tr}\boldsymbol{H}_{\mathbb{K}^c})^2}{n^2}\boldsymbol{H}_{\mathbb{K}}^{-2}+\boldsymbol{I}_{\mathbb{K}^c}\right),\ \boldsymbol{w}^*\boldsymbol{w}^{*\top}\right\rangle \\
&= \left\|\left(\frac{(\operatorname{tr}\boldsymbol{G}_{\mathbb{J}^c})^2}{n^2}\boldsymbol{G}_{\mathbb{J}}^{-2}+\boldsymbol{I}_{\mathbb{J}^c}\right)^{\frac12}\cdot\left(\frac{(\operatorname{tr}\boldsymbol{H}_{\mathbb{K}^c})^2}{n^2}\boldsymbol{H}_{\mathbb{K}}^{-2}+\boldsymbol{I}_{\mathbb{K}^c}\right)^{\frac12}\boldsymbol{w}^*\right\|_{\boldsymbol{G}}^2.
\end{aligned}
$$

We also give the variance lower bound in the expectation form.

$$
\begin{aligned}
\mathbb{E}\texttt{variance} &= \sigma^2\big(\langle \mathbb{E}\mathcal{P}^{(2)}\boldsymbol{G}\mathcal{P}^{(2)},\ \mathbb{E}\boldsymbol{X}^{(1)\top}\boldsymbol{A}^{(1)-2}\boldsymbol{X}^{(1)}\rangle + \langle \boldsymbol{G},\ \mathbb{E}\boldsymbol{X}^{(2)\top}\boldsymbol{A}^{(2)-2}\boldsymbol{X}^{(2)}\rangle\big) \\
&\gtrsim \frac{\sigma^2}{n}\left(\left\langle \boldsymbol{G}\cdot\left(\frac{(\operatorname{tr}\boldsymbol{H}_{\mathbb{K}^c})^2}{n^2}\boldsymbol{H}_{\mathbb{K}}^{-2}+\boldsymbol{I}_{\mathbb{K}^c}\right),\ \boldsymbol{G}_{\mathbb{J}}^{-1}+\frac{n^2}{(\operatorname{tr}\boldsymbol{G}_{\mathbb{J}^c})^2}\cdot\boldsymbol{G}_{\mathbb{J}^c}\right\rangle + \left\langle \boldsymbol{G},\ \boldsymbol{H}_{\mathbb{K}}^{-1}+\frac{n^2}{(\operatorname{tr}\boldsymbol{H}_{\mathbb{K}^c})^2}\cdot\boldsymbol{H}_{\mathbb{K}^c}\right\rangle\right) \\
&= \frac{\sigma^2}{n}\left(\left\langle \frac{(\operatorname{tr}\boldsymbol{H}_{\mathbb{K}^c})^2}{n^2}\boldsymbol{H}_{\mathbb{K}}^{-2}+\boldsymbol{I}_{\mathbb{K}^c},\ \boldsymbol{I}_{\mathbb{J}}+\frac{n^2}{(\operatorname{tr}\boldsymbol{G}_{\mathbb{J}^c})^2}\cdot\boldsymbol{G}_{\mathbb{J}^c}^2\right\rangle + \left\langle \boldsymbol{G},\ \boldsymbol{H}_{\mathbb{K}}^{-1}+\frac{n^2}{(\operatorname{tr}\boldsymbol{H}_{\mathbb{K}^c})^2}\cdot\boldsymbol{H}_{\mathbb{K}^c}\right\rangle\right)
\end{aligned}
$$

We have finished the proof. $\qquad\square$

### C.7 PROOF OF EXAMPLES

*Proof of Example 9.* We examine $\mathbb{E}\left[\mathcal{R}_1(\boldsymbol{w}^{(2)})-\mathcal{R}_1(\boldsymbol{w}^*)\right]$ to represent the joint risk and forgetting.

1. By Theorem 18 we have

$$
\texttt{variance} \geq \frac{\sigma^2}{n}\langle \boldsymbol{G},\ \boldsymbol{H}_{\mathbb{K}}^{-1}\rangle \geq \frac{\sigma^2}{n}\cdot\frac{\mu_1}{\lambda_1}=1.
$$

As a result, $\mathbb{E}\left[\mathcal{R}_1(\boldsymbol{w}^{(2)})-\mathcal{R}_1(\boldsymbol{w}^*)\right]=\Omega(1)$.

2. By Theorem 18 we have

$$
\texttt{variance} \geq \frac{\sigma^2}{n}\langle \boldsymbol{G},\ \frac{n^2}{(\operatorname{tr}\boldsymbol{H}_{\mathbb{K}^c})^2}\boldsymbol{H}_{\mathbb{K}^c}\rangle,
$$

where $\mathbb{K}=\{i:\lambda_i>\lambda^*\}$ and the choice of $\lambda^*$ let $\mathbb{K}$ satisfies $\operatorname{tr}(\boldsymbol{H}_{\mathbb{K}^c})\asymp n\|\boldsymbol{H}_{\mathbb{K}^c}\|_2$. By definition of $\boldsymbol{H}$, $\mathbb{K}=\{i:i\geq i^*\}$ where $i^*$ satisfies $i^*\log i^*\asymp n$. Therefore,

$$
\begin{aligned}
\texttt{variance} &\geq \frac{\sigma^2}{n}\langle \boldsymbol{G},\ \frac{n^2}{(\operatorname{tr}\boldsymbol{H}_{\mathbb{K}^c})^2}\boldsymbol{H}_{\mathbb{K}^c}\rangle = n\cdot\frac{\sum_{i\geq i^*}\mu_i\lambda_i}{\left(\sum_{i\geq i^*}\lambda_i\right)^2} \\
&= n\cdot\frac{\sum_{i\geq i^*}i^{-2}\log^{-\alpha-\beta}i}{\left(\sum_{i\geq i^*}i^{-1}\log^{-\beta}i\right)^2} \\
&\gtrsim n\cdot\frac{i^{*-1}\log^{-\alpha-\beta}i^*}{\left(\log^{-\beta+1}i^*\right)^2}\asymp \log^{\beta-\alpha-1}i^* \gtrsim \log^{\beta-\alpha-1}n.
\end{aligned}
$$

As a result, $\mathbb{E}\left[\mathcal{R}_1(\boldsymbol{w}^{(2)})-\mathcal{R}_1(\boldsymbol{w}^*)\right]=\Omega\left(\log^{\beta-\alpha-1}n\right)$.

$\qquad\square$

## D  PROOF OF EXTENSIONS

### D.1  PROOF OF CL IN THE NTK REGIME

We set up a 2-task CL setting in the NTK regime. Consider two data distributions $\mathcal{D}_1, \mathcal{D}_2$. For each task, $n$ individual samples $(\boldsymbol{x}_i^{(1)}, y_i^{(1)})_{i=1}^n, (\boldsymbol{x}_i^{(2)}, y_i^{(2)})_{i=1}^n$ are sampled independently from $\mathcal{D}_1, \mathcal{D}_2$. The goal is to learn a neural network model $f_{\boldsymbol{w}}(\boldsymbol{x})$ specified by parameter $\boldsymbol{w} \in \mathcal{R}^d$. Lee et al. (2019) proved that under the NTK regime, neural networks evolve as a linear model:

$$f_{\boldsymbol{w}}(\boldsymbol{x}) = f^{(0)}(\boldsymbol{x}) + \langle \nabla_{\boldsymbol{w}} f^{(0)}(x), \boldsymbol{w} - \boldsymbol{w}^{(0)} \rangle,$$

where $f^{(0)}(\cdot)$ denotes the model with $\boldsymbol{w}^{(0)} = \boldsymbol{0}$ as the initial weight. This formulation implies that the feature map $\phi(\boldsymbol{x}) = \nabla_{\boldsymbol{w}} f^{(0)}(x) \in \mathcal{R}^d$ is constant over time.

Under this setting, we consider the *convergence of GRCL with gradient descent (GD)*, consistent with the setting of Section 3.2 in Bennani et al. (2020). In this algorithm, we optimize the following training loss for tasks $t = 1, 2$:

$$L_1(\boldsymbol{w}_\tau^{(1)}) = \frac{1}{n} \sum_{i=1}^n (f_\tau^{(1)}(\boldsymbol{x}_i^{(1)}) - y_i^{(1)})^2, \tag{26}$$

$$L_2(\boldsymbol{w}_\tau^{(2)}) = \frac{1}{n} \sum_{i=1}^n (f_\tau^{(2)}(\boldsymbol{x}_i^{(2)}) - y_i^{(2)})^2 + \|\boldsymbol{w}_\tau^{(2)} - \boldsymbol{w}^{(1)}\|_{\boldsymbol{\Sigma}}^2, \tag{27}$$

where $\boldsymbol{\Sigma}$ is the regularization matrix, $\tau$ is the GD iteration number, and $\boldsymbol{w}^{(1)}, \boldsymbol{w}^{(2)}$ is the limit of $\boldsymbol{w}_\tau^{(1)}, \boldsymbol{w}_\tau^{(2)}$, respectively. The update rule of GD for tasks $t = 1, 2$ is that

$$\boldsymbol{w}_{\tau+1}^{(t)} = \boldsymbol{w}_\tau^{(t)} - \eta \nabla_{\boldsymbol{w}} L_t(\boldsymbol{w}_\tau^{(1)}).$$

We have the following theorem to connect GRCL in this NTK regime with our linear model:

**Theorem 35** (Continual Learning in NTK as a linear GRCL). *In the NTK setting, with an infinitesimally small $\eta$, GRCL outputs $\boldsymbol{w}^{(2)}$ such that:*

$$\boldsymbol{w}^{(1)} = \left(\phi(\boldsymbol{X}^{(1)})^\top \phi(\boldsymbol{X}^{(1)})\right)^{-1} \phi(\boldsymbol{X}^{(1)})^\top (\boldsymbol{y}^{(1)} - f^{(0)}(\boldsymbol{X}^{(1)}));$$

$$\boldsymbol{w}^{(2)} = \arg\min_{\boldsymbol{w}} \frac{1}{n} \|\boldsymbol{y}^{(2)} - f^{(1)}(\boldsymbol{X}^{(2)}) - \phi(\boldsymbol{X}^{(1)})^\top (\boldsymbol{w} - \boldsymbol{w}^{(1)})\|_2^2 + \|\boldsymbol{w} - \boldsymbol{w}^{(1)}\|_{\boldsymbol{\Sigma}}^2.$$

*Proof.* By putting (12) into (26) and (27), and computing the first-order optimality condition of (26), we get the result. □

This theorem essentially reduces the GRCL in the NTK setting to the linear GRCL algorithm in (3), providing that we give the following inputs $\boldsymbol{x}', y'$ to the linear model:

$$\boldsymbol{x}'^{(1)} = \phi(\boldsymbol{x}^{(1)}), \ \boldsymbol{x}'^{(2)} = \phi(\boldsymbol{x}^{(2)}), \ y'^{(1)} = y^{(1)} - f^{(0)}(\boldsymbol{x}^{(1)}), \ y'^{(2)} = y^{(2)} - f^{(1)}(\boldsymbol{x}^{(2)}). \tag{28}$$

**Assumption 5** (NTK data distribution condition). We assume that $\boldsymbol{x}'^{(1)}, \boldsymbol{x}'^{(2)}, y'^{(1)}, y'^{(2)}$ satisfy Assumptions 1, 2, 3.

Note that in this case, the covariance matrices $\boldsymbol{G} := \mathbb{E}_{\mathcal{D}^{(1)}}[\phi(\boldsymbol{x}^{(1)})\phi(\boldsymbol{x}^{(1)})^\top], \boldsymbol{H} := \mathbb{E}_{\mathcal{D}^{(2)}}[\phi(\boldsymbol{x}^{(2)})\phi(\boldsymbol{x}^{(2)})^\top]$ are actually the Hessian matrices of the model. Now we have our main results in the NTK regime:

**Theorem 36.** *Suppose Assumption 5 holds. Then for the GRCL output $\boldsymbol{w}^{(2)}$ in the NTK regime, it holds that*

$$\mathbb{E}\Delta(\boldsymbol{w}^{(2)}) = \texttt{bias} + \texttt{variance},$$

*where the bias and variance satisfy* (6) *defined in Theorem 2.*

Therefore, our main results in Theorem 2 and its collollaries and messages still hold in the NTK regime and can be applied in general neural networks.

## D.2 PROOF OF MULTI-TASK CL

We set up a multi-task CL setting with linear regression problems. For tasks $t = 1, \ldots, T$ where $T \geq 3$ is a constant, consider the data distributions $(\mathcal{D}^{(t)})_{t=1}^T$. For each task $t$, $n$ samples $(\boldsymbol{x}_i^{(t)}, y_i^{(t)})_{i=1}^n$ are drawn independently from the task distribution. The goal is to learn a model $\boldsymbol{w}$ to minimize the joint population risk $\mathcal{R}(\boldsymbol{w}) = \sum_t \mathcal{R}_t(w)$, where $\mathcal{R}_t(\boldsymbol{w}) := \mathbb{E}_{\mathcal{D}^{(t)}}(y - \boldsymbol{w}^\top \boldsymbol{x})^2$ is defined in the same way as Definition 2 in the paper. We retain the shared-optimal assumption in Definition 2. For the covariance condition in Definition 1, we denote:

**Definition 3** (Covariance conditions). $\boldsymbol{H}^{(t)} := \mathbb{E}_{\mathcal{D}^{(t)}}[\boldsymbol{x}\boldsymbol{x}^\top] = \mathrm{diag}(\lambda_i^{(2)}), \boldsymbol{G}^{(t)} := \sum_{\tau=1}^t \boldsymbol{H}^{(t)} = \mathrm{diag}(\mu_i^{(2)})$.

One can verify that $\boldsymbol{G}^{(1)} = \boldsymbol{G}$ and $\boldsymbol{H}^{(2)} = \boldsymbol{H}$ is consistent with the original Definition 1. Under this multi-task setting, we consider the GRCL algorithm, in which the regularization matrix series $\boldsymbol{\Sigma}^{(t)}$ can adapt as the number of task $t$ grows:

$$\boldsymbol{w}^{(1)} = (\boldsymbol{X}^{(1)}{}^\top \boldsymbol{X}^{(1)})^{-1} \boldsymbol{X}^{(1)}{}^\top \boldsymbol{y}^{(1)};$$

$$\boldsymbol{w}^{(t)} = \arg\min_{\boldsymbol{w}} \frac{1}{n}\|\boldsymbol{y}^{(t)} - \boldsymbol{X}^{(t)}\boldsymbol{w}\|_2^2 + \|\boldsymbol{w} - \boldsymbol{w}^{(t-1)}\|_{\boldsymbol{\Sigma}^{(t)}}^2, \ 2 \leq t \leq T.$$

**Assumption 6** (Multi-task data distribution condition). We assume that Assumptions 1, 2, 3 hold for $t = 1, \ldots, T$.

*Proof of Corollary 11.* We use induction to prove this corollary. Consider $E\Delta(w^{(t)})$. For $t = 2$, (14) already holds due to the original Corollary 6. Now we assume (14) holds for task $t-1$ such that $3 \leq t \leq T$. In this case, according to (18), we have

$$\mathbb{E}\Delta(\boldsymbol{w}^{(t-1)}) = \langle \boldsymbol{G}^{(t-1)}, \mathbb{E}(\boldsymbol{w}^{(t-1)} - \boldsymbol{w}^*)(\boldsymbol{w}^{(t-1)} - \boldsymbol{w}^*)^\top \rangle \lesssim \mathbb{E}\Delta(\boldsymbol{w}_{\texttt{joint}}).$$

Then for task $t$, reall that $\boldsymbol{P}_{\boldsymbol{\Sigma}}^{(t)} = (\boldsymbol{X}^{(t)}{}^\top \boldsymbol{X}^{(t)} + n\boldsymbol{\Sigma}^{(t)})^{-1} n\boldsymbol{\Sigma}^{(t)}$. According to Eqs. (12) and (15),

$$\begin{aligned}
\mathbb{E}\Delta(\boldsymbol{w}^{(t)}) &= \langle \boldsymbol{G}^{(t-1)} + \boldsymbol{H}^{(t)}, \mathbb{E}(\boldsymbol{w}^{(t)} - \boldsymbol{w}^*)(\boldsymbol{w}^{(t)} - \boldsymbol{w}^*)^\top \rangle \\
&= \mathbb{E}_t \langle \boldsymbol{G}^{(t-1)} + \boldsymbol{H}^{(t)}, \boldsymbol{P}_{\boldsymbol{\Sigma}}^{(t)} \mathbb{E}(\boldsymbol{w}^{(t-1)} - \boldsymbol{w}^*)(\boldsymbol{w}^{(t-1)} - \boldsymbol{w}^*)^\top \boldsymbol{P}_{\boldsymbol{\Sigma}}^{(t)} \rangle \\
&\quad + \sigma^2 \langle \boldsymbol{G}^{(t-1)} + \boldsymbol{H}^{(t)}, \mathbb{E}_t(\boldsymbol{X}^{(t)}{}^\top \boldsymbol{X}^{(t)} + n\boldsymbol{\Sigma}^{(t)})^{-2} \boldsymbol{X}^{(t)}{}^\top \boldsymbol{X}^{(t)} \rangle.
\end{aligned}$$

For the first term, notice that according to Lemma 13,

$$\mathbb{E}_t(\boldsymbol{G}^{(t-1)} + \boldsymbol{H}^{(t)})\boldsymbol{P}_{\boldsymbol{\Sigma}}^{(t)}{}^2 \leq (\boldsymbol{G}^{(t-1)} + \boldsymbol{H}^{(t)})[\boldsymbol{G}^{(t-1)}{}^2(\boldsymbol{G}^{(t-1)} + \boldsymbol{H}^{(t)})^{-2} + (\boldsymbol{I}_{\mathbb{J}} - \boldsymbol{H}_{\mathbb{J}}^{(t)})^n] \lesssim \boldsymbol{G}^{(t-1)}.$$

Therefore,

$$\begin{aligned}
&\mathbb{E}_t \langle \boldsymbol{G}^{(t-1)} + \boldsymbol{H}^{(t)}, \boldsymbol{P}_{\boldsymbol{\Sigma}}^{(t)} \mathbb{E}(\boldsymbol{w}^{(t-1)} - \boldsymbol{w}^*)(\boldsymbol{w}^{(t-1)} - \boldsymbol{w}^*)^\top \boldsymbol{P}_{\boldsymbol{\Sigma}}^{(t)} \rangle \\
&\lesssim \langle \boldsymbol{G}^{(t-1)}, \mathbb{E}(\boldsymbol{w}^{(t-1)} - \boldsymbol{w}^*)(\boldsymbol{w}^{(t-1)} - \boldsymbol{w}^*)^\top \rangle \\
&\lesssim \mathbb{E}\Delta(\boldsymbol{w}_{\texttt{joint}}).
\end{aligned}$$

For the second term, following (22), it is also bounded by variance$_{\texttt{joint}}$, thus bounded by $\mathbb{E}\Delta(\boldsymbol{w}_{\texttt{joint}})$. Adding them up and we get $\mathbb{E}\Delta(\boldsymbol{w}^{(t)}) \lesssim \mathbb{E}\Delta(\boldsymbol{w}_{\texttt{joint}})$.

By induction, we get the result. $\qquad\square$

Corollary 11 extends the message in the paper that with sufficient memory and appropriate regularization, GRCL can match the performance of joint training in multi-task CL. Combining with the existing Example 7, where low-memory CL behaves poorly, we deliver the message that there is a provable memory-statistics trade-off in multi-task CL.

# E  EXPERIMENTS

## E.1  NUMERICAL EXPERIMENT SETTINGS

We consider a CL problem instance adapted from Wu et al. (2022) and Li et al. (2023). Specifically, the eigenvalues for $\boldsymbol{G}$ and $\boldsymbol{H}$ are $\{1/2^{i-1}\}_{i=1}^{d}$. For a given integer $k > 0$ and a small value $0 < \delta < 1$, the CL problem $\mathsf{P}(k) = (\boldsymbol{w}^*, \sigma^2, \boldsymbol{G}, \boldsymbol{H})$ is specified as follows:

$$\boldsymbol{G} = \mathrm{diag}\left(1, \frac{1}{2}, \frac{1}{2^2}, \ldots, \frac{1}{2^{k-1}}, \frac{1}{2^k}, \cdots\right), \quad \boldsymbol{H} = \mathrm{diag}\left(\frac{1}{2^{k-1}}, \frac{1}{2^{k-2}}, \ldots, 1, \frac{1}{2^k}, \cdots\right), \quad (29)$$
$$\boldsymbol{w}^* = (1, 2^{-1}, 3^{-1}, \ldots, k^{-1}, \ldots)^\top, \qquad \sigma^2 = 1,$$

We choose $\mathsf{P}(15)$ for our illustration. It is evident that $\boldsymbol{G}$ and $\boldsymbol{H}$ differ in their top-15 eigenvalues. For the GRCL algorithm, the regularization matrix $\boldsymbol{\Sigma}$ is specified by the rank-$k$ approximation of the empirical covariance matrix of the first task $\frac{1}{n}\boldsymbol{X}^{(1)}{}^\top \boldsymbol{X}^{(1)}$. We then test this problem instance and compare the GRCL risk convergence with OCL and joint learning results.

## E.2  NEURAL NETWORK EXPERIMENTS

In this section, we further verify our theoretical results with experiments on practical CL datasets using neural networks, complementing the numerical experiments in Section 4.2. We use Permuted MNIST and Rotated MNIST, two CL benchmark datasets widely used in the literature (Kirkpatrick et al., 2017; Farajtabar et al., 2020). In both experiments, we consider a two-task problem: the first consists of standard MNIST digits, while the input of the second task is transformed from NMINST. In Permuted MNIST, a random shuffle of pixels is applied on the $28 \times 28$ images of the original MNIST hand-written digits to create the second task; in Rotated MNIST, instead of the random shuffle, a rotation of an unknown fixed angle is applied.

In analogy to the OCL and the GRCL algorithms that we theoretically analyzed in Section 4, we examine three algorithms: vanilla training without regularization, full regularization, and PCA-based low-rank regularization. In vanilla training, standard optimization is performed with the Adam optimizer during the second task training. In the full regularization algorithm, the Hessian is computed at the end of the first task training and is used as the regularization matrix in training the second task. In the low-rank regularization algorithm, instead of the full Hessian, PCA with a predetermined rank is performed on the saved regularization matrix. We examine different network architectures for the two datasets due to their different properties. For Permuted MNIST, we use an MLP-based model with 4 residual blocks, each consisting of 2 MLP layers with a hidden size of 40. The model input dimension is 196 (the MNIST figures are resized to $14 \times 14$), and the output dimension is 10. For Rotated MNIST, we use a CNN-based network with a LeNet5 structure.

The experiment results are shown in Figure 2. We make the following observation: In both settings, the vanilla algorithm suffers from a significant drop in average accuracy due to catastrophic forgetting. As the memory size increases, the regularized algorithm performance improves and achieves a reasonable empirical result. This result corresponds to the memory-statistics tradeoff demonstrated in our theory and numerical experiments.

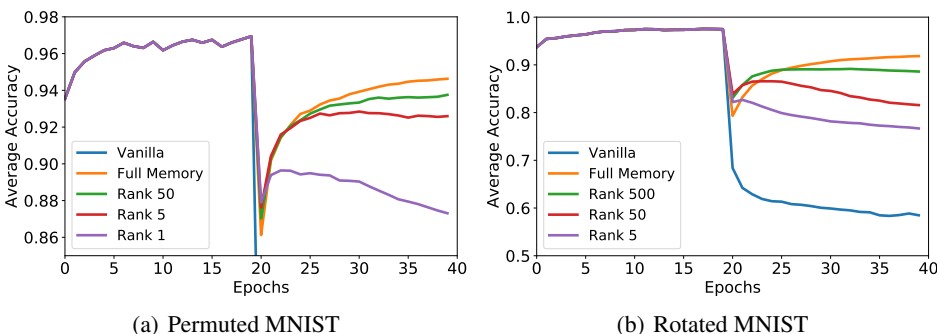

(a) Permuted MNIST

(b) Rotated MNIST

Figure 2: Average accuracy across previously learned tasks on (a) Permuted MNIST and (b) Rotated MNIST after each epoch of training for the vanilla algorithm without regularization, the regularization-based method with full Hessian, and with low-rank regularization. In both experiments, we use the Adam optimizer with a learning rate of $10^{-4}$. The moving average parameter is $\alpha = 0.25$ and the regularization coefficient is $10^4$ for all algorithms.

