# OpenReview forum: "Memory-Statistics Tradeoff in Continual Learning with Structural Regularization"
_ICLR.cc/2026/Conference — ICLR 2026 Poster_

### Official Review · Reviewer_GXcu · 2025-10-26

**Soundness:** 3
**Presentation:** 4
**Contribution:** 3
**Rating:** 6
**Confidence:** 4

**Summary:**

The paper studies continual learning (CL) for two linear regression tasks under random design and covariate shift. It analyzes a generalized $\ell_2$-regularized CL algorithm (GRCL) that, during the second task, penalizes deviation from the first-task solution using a user-chosen positive semidefinite matrix $\Sigma$. The work derives matching upper and lower bounds on the joint excess risk and shows a quantitative trade-off between memory and statistics: richer $\Sigma$ (higher memory, measured by the number of stored vectors/eigen-components) yields lower excess risk. It proves that ordinary CL (OCL) and isotropic $\ell_2$ regularization can suffer catastrophic forgetting, while an appropriately chosen $\Sigma$ can match joint training performance. Extensions include lower/upper bounds for Gaussian design, a memory-risk curve with top-$k$ regularization, a discussion of NTK-regime neural networks, and small-scale experiments illustrating the theory.

**Strengths:**

The paper makes a clear and useful move: it treats continual learning through a memory-statistics lens and asks how much structure one must store to avoid forgetting. This framing is fresh and concrete. The theory then backs it up with matched upper and lower bounds. The main theorem decomposes risk into bias and variance terms that depend on the regularization matrix. This gives readers a direct way to see why naive OCL or an isotropic ridge can fail, and why a structured $\ell_2$ penalty helps. The "top-$k$" view of the matrix is especially appealing. It turns memory into a dial that one can set and shows that risk falls toward joint training as the rank grows. The examples are well chosen: they show catastrophic forgetting when the method ignores key directions, and they show how the proposed structure fixes that. The writing of the core results is also clear. Assumptions are stated, quantities are interpretable, and the logic builds in a steady way.

**Weaknesses:**

1. Strength of assumptions and scope: The shared optimal parameter and commuting covariances (diagonalizable in the same basis) simplify analysis but may be restrictive in practice.

2. From one-hot to Gaussian (and beyond): Main sharp results are for one-hot design; Gaussian design coverage is partial (OCL bounds, phenomena via Example 9), while GRCL bounds are deferred.

3.  Model selection for $k$ (or $\Sigma$): The theory shows performance as a function of $k$, but there is no principled selection rule.


4. Positioning vs. prior theory: The paper cites related work, but the novelty over Li et al. (2023) and Zhao et al. (2024) could be crisper, especially regarding random design + memory-explicit lower bounds.


5. Memory accounting and implementability: Memory is "number of vectors to store" or "rank of $\Sigma$, but concrete byte/parameter costs are not spelled out, especially for neural nets where curvature is layer-structured.

**Questions:**

1. Strength of assumptions and scope.

- Regarding the optimal parameter, can you add a proposition or appendix note quantifying the error when optimal parameters differ (the text hints at triangle-inequality arguments) and provide an explicit bound to show robustness to small misspecification?
- Can you relax "commuting $G, H$" via a perturbation result? For example, if $||G H-H G||$ is small, how do constants in the bounds inflate? Even a qualitative theorem would help.

2. From one-hot to Gaussian (and beyond).

- Can you provide at least an upper bound for GRCL in Gaussian design under mild spectral decay (e.g., polynomial/log-corrected tails)? Even if looser, it would connect the top-k message to the more common model. I think you can use Theorem 4.2 of Zhao et al. (2024, ICML) to do this.
- Can you include a short "what breaks" paragraph? For example, I was curious which steps in the one-hot proof fail for Gaussian, and what tools are perhaps needed?

3.  Model selection for $k$ (or $\Sigma$).

- Can the authors propose a simple criterion?
- The authors mention CountSketch can be used to specify the memory utilized in GRCL. Can the authors state a data-driven estimator for $\Sigma$ with guarantees? (Formalize an estimation step and propagate its error into the risk bound).

4. Positioning vs. prior theory.

- Can the authors add a comparative lemma or table? For example, which settings (fixed vs random; isotropic vs structured $\Sigma$; with/without matching lower bounds) each prior paper covers, and what this paper adds.

5. Memory accounting and implementability.

- Can the authors first define “memory” precisely, and then provide worked examples that show how to compute and compare storage costs for different choices of examples? For example, diagonal $\Sigma$, block-diagonal by layer, K-FAC with factor sizes, and a sketch-based $\Sigma$ showing CountSketch dimension versus error. I think this effort would close this gap and make the proposed trade-offs operational.

---

> ### Author Response · Authors · 2025-11-23
>
> Thank you for your constructive comments, which are very helpful for us to further improve the paper. We address your concerns as follows.
>
> ---
> Q1. Strength of assumptions and scope:
> Can you add a proposition or appendix note quantifying the error when optimal parameters differ (the text hints at triangle-inequality arguments) and provide an explicit bound to show robustness to small misspecification?
> Can you relax "commuting" via a perturbation result?
>
> A1.
> We provide an extension of out theoretical result when the optimal parameters differ via the following Proposition.
>
> Proposition 2’: Under the same condition in Theorem 2 without shared optimal parameter in Assumption 1, denote ${w^{(1)}}^\*$ and ${w^{(2)}}^\*$ as the optimal parameter for task 1 and 2. Then the expectation of excess risk satisfies
> $$ E\Delta(w^{(2)})\leq bias+variance+||{w^{(1)}}^\*-{w^{(2)}}^\*||_2^2, $$
> where the variance stays the same with Theorem 2, and the bias is the same with Theorem 2 with $w^\*$ replaced with ${w^{(2)}}^\*$ .
>
> Also, we provide an extension of our theoretical result when the commutativity assumption is slightly violated.
>
> Proposition 2’’: Under the same condition in Theorem 2 without commutativity in Assumption 2, suppose $||G-G’||_2<\alpha$ where $G’$ is commutable with $H$. Then the expectation of excess risk satisfies
> $$ E\Delta(w^{(2)})\leq bias+variance, $$
> where in both bias and variance, the $G+H$ terms in Eq. (6) are replaced with $G+H+2\alpha I$.
>
> This will be enough for small misspecification. We will make sure to include a polished version of these in the revision.
>
> ---
>
> Q2. From one-hot to Gaussian (and beyond):
> Can you provide an upper bound for GRCL in Gaussian design?... I think you can use Theorem 4.2 of Zhao et al. (2024, ICML) to do this.
> Which steps in the one-hot proof fail for Gaussian, and what tools are perhaps needed?
>
> A2. The mathematical tools used in Gaussian design is fundamentally different with that in one-hot setting. In particular, in computing the expectation of projection matrix $P_1 H P_1$ and $P_2 G P_2$ (related to Lemmas 14-16 in one-hot proofs in Appendix B.2, and Lemmas 29-31 in Gausssian proofs in Appendix C.5), different coordinates cannot be easily splitted in Gaussian design. Tools listed in Appendices C.3, C.4 are used to partially solve the problem in OCL, and more complex statistical tools on the fourth-order of high-dimensional Inverse Gaussian distribution are likely needed to fully address the problem.
>
> We note that the analysis in [Zhao et al. 2024] is not compatible with Gaussian design: it is based on the fixed design where the inputs are fixed. We would like to emphasize that even with OCL, there is no such kind of result before our work on analyzing CL on Gaussian distribution with non-identical arbitrary covariance. As our work is the first work on studying memory-statistics tradeoff for CL, we believe we have taken an important step towards this direction and the contribution of our work is significant.
>
> ---
>
> Q3. Model selection:
> Can the authors propose a simple criterion?
> The authors mention CountSketch can be used to specify the memory utilized in GRCL. Can the authors state a data-driven estimator with guarantees?
>
> A3. We have provided an illustrative criterion in selecting $\Sigma$ and $k$ and in Example 6 and 8, respectively. We show that the optimal choice of $k$ is closely related to the rank of dominant eigenspace of data covariance, and $\Sigma$ should approximate that data covariance.
>
> For a data-driven estimator by CountSketch with estimation guarantees, please refer to [1]. Propagate the estimation error of such algorithms into the risk bound is an interesting future direction, which we will mention in the revision.
>
> [1] Li et al. Lifelong Learning with Sketched Structural Regularization. ACML 2021.
>
> ---
>
> Q4. Positioning vs. prior theory:
> Can the authors add a comparative lemma or table?
>
> A4. Sure. We have such a table in previous versions but have removed it due to conciseness and page limits. We provide the concise version of it below due to Markdown constraints. We will put the full version back to the appendix in the revision.
>
> | | [Li et al. 2023] | Our result | Our result | |
> |---|---|---|---|---|
> | Input distribution | Fixed design | One-hot | Gaussian|
> | Dominant feature sets | | $J=\\{i:\mu_i>\mu^\*\\}, K=\\{i:\lambda_i>\lambda^*\\}$ | |
> | Dominant feature conditions| $\lambda^*=0$ | $\mu^\*=\lambda^\*=\frac{1}{n}$ | $\mu^\*: tr(G_{J^c}) \gtrsim n{\\\|G_{J^c}\\\|}_2,$ similar for $ \lambda^\*$  |
>
> [2] Li et al. Fixed design analysis of regularization-based continual learning. CoLLAs 2023.

---

> ### Author Response · Authors · 2025-11-23
>
> ---
>
> Q5. Memory accounting and implementability.
> Can the authors first define “memory” precisely, and then provide worked examples that show how to compute and compare storage costs for different choices of examples? For example, diagonal, block-diagonal by layer, K-FAC with factor sizes, and a sketch-based showing CountSketch dimension versus error.
>
> A5. We provided the memory definition of regularization-based CL algorithms in linear models in the paper with a size of “$k$ vectors of dimension $d$”; please refer to Ln 196-197. In this definition, CountSketch has $k$ vectors of dimension $d$, where $k$ is the sketch size. Diagonal regularization in a linear model has a size of a single vector of dimension $d$; however, it is known that diagonal regularization performs poorly in linear models.
>
> The other type of regularization, including block-diagonal and K-FAC, is dependent on multi-layer neural network. The theoretical analysis of generalization error of multi-layer neural network is an interesting research question and a well-known unresolved open problem, thus is out of the scope of this paper. We can generally quantify the memory of these CL algorithms by the additional parameters they save.

---

> > ### Comment · Reviewer_GXcu · 2025-11-23
> > **Comment**
> >
> > Thanks for the careful revisions to Q1–Q3. For Q4, I was acutally wondering why you analyze the memory-statistics trade-off under a random design, and what would change (if anything) under a fixed design? For Q5, my suggestion is a formal definition of memory complexity in the main paper. In Lines 196--197, the authors wrote "Compared to OCL, GRCL takes an additional memory of Σ, which can be expressed with k vectors of dimension d." Also, in the abstract, the authors wrote "Our analysis reveals a fundamental trade-off between memory complexity and statistical efficiency, where memory complexity is measured by the number of vectors needed to define the structural regularization." These are descriptive rather than a definition. This matters because different implementations can have very different storage needs. For example, a low-rank (p\times p) matrix costs ($O(p^2)$) numbers to store directly, but only ($2rp+r$) via an SVD, and $p(c+r)+cr$ for CUR decompositions, when (rank=r). Since the authors studied only on GRCL, it would be easy to give a formal, unifying definition of memory cost for different special cases of GRCL.

---

> > > ### Author Response · Authors · 2025-11-28
> > >
> > > Thanks for your questions.
> > >
> > > For Q4, the random setting is considered more “realistic” than the fixed design, since the training inputs in machine learning applications are commonly seen as random. Note that the widely used practical datasets (e.g. CIFAR, ImageNet, etc.) are collected randomly from the underlying environment distribution. Compared to the fixed design, the random design is closer to the typical input statistics in machine learning applications: the risk in random design is measured on the population distribution, which is analogous to test performance in real applications, rather than on the training inputs in the fixed design.
> > >
> > > The risk behaviour changes under fixed design in that the tail of the data covariance affect risk in a different way. We update the table in the rebuttal for a clearer look.
> > >
> > > For Q5, we agree with the reviewer that there are numerous ways to store a low-rank matrix in applications. Specifically, it costs at least $r \times p$ number of parameters to store an arbitrary $p\times p$ matrix $\Sigma$ with rank $r$, in the form of $\Sigma = W^\top W$. We can define GRCL memory cost as the minimum number of parameters needed to store the class of regularizer in this particular CL algorithm. Please feel free to raise opinions on this definition.

---

### Official Review · Reviewer_RwWP · 2025-10-30

**Soundness:** 3
**Presentation:** 3
**Contribution:** 3
**Rating:** 6
**Confidence:** 4

**Summary:**

This paper studies the trade-off between statistical performance and memory cost in Continual Learning (CL) with structural regularization methods. Focusing on a two-task linear regression model under a random design setting with covariate shift, the authors propose a Generalized ℓ₂-regularized Continual Learning (GRCL) algorithm and establish upper and lower bounds for its joint excess risk. The theoretical analysis reveals that Ordinary Continual Learning (OCL) without regularization and simple ℓ₂-regularized CL (ℓ₂-RCL) suffer from catastrophic forgetting in certain tasks, but a well-designed structural regularization (GRCL) with sufficient memory can mitigate forgetting and achieve statistical performance comparable to joint learning. The theoretical results are validated through numerical experiments and extended to neural networks in the Neural Tangent Kernel (NTK) regime.

**Strengths:**

First to provide rigorous theoretical formulation of the memory-performance trade-off in continual learning, with a unifying GRCL framework.

Quality:

Solid theoretical analysis with matching bounds and well-constructed examples. Clean extension to neural networks. Experiments properly validate theory.

Clarity:

Well-structured and logically coherent. Core ideas are clearly communicated despite some heavy notation.

Significance:

Establishes valuable theoretical foundation for continual learning. Provides practical guidance for algorithm design and opens directions for future work.

**Weaknesses:**

(1)	Limited Empirical Validation

The experimental validation, while supporting the theoretical findings, remains somewhat limited in scope and practical impact:
Neural network experiments are conducted only on small-scale MNIST variants (Permuted/Rotated MNIST). To better demonstrate real-world relevance, experiments on more challenging benchmarks (e.g., CIFAR-10, Split CIFAR-100) would strengthen the claims.
The comparison focuses primarily on methods within the structural regularization family. Including strong baselines from other CL paradigm would provide a more comprehensive assessment of where GRCL stands in the current landscape.

(2)	Strong Theoretical Assumptions

The analysis relies on several strong assumptions that may limit practical applicability. For example, the commutativity assumption of covariance matrices (Assumption 2), while mathematically convenient, rarely holds in real-world datasets. The authors could discuss how violations of this assumption might affect their conclusions, or provide empirical studies showing the theory's robustness to approximate commutativity.

**Questions:**

(1)	Could you discuss the practical implications of the commutativity assumption (Assumption 2)? How might your conclusions change when this assumption is violated in real-world datasets?

(2)	Would you consider adding experiments on more challenging benchmarks (e.g., CIFAR-100, ImageNet subsets) to better demonstrate the practical relevance of your theory?

(3)	Have you compared GRCL with strong baselines from other CL paradigms (e.g., experience replay, expansion methods)? Such comparisons would help position your method in the broader CL landscape.

---

> ### Author Response · Authors · 2025-11-23
>
> Thank you for your constructive comments, which are very helpful for us to further improve the paper. We address your concerns as follows.
>
> ---
>
> Q1. Could you discuss the practical implications of the commutativity assumption (Assumption 2)? How might your conclusions change when this assumption is violated in real-world datasets?
>
> A1. In practice, the commutativity assumption of a CL dataset is satisfied when the CL tasks share a same group of features. A particular example of a commutative CL dataset is Permuted-MNIST, in which the pixel features between tasks are only shuffled; it is straightforward to show in math that shuffling retains the eigenspace (and thus commutativity) of dataset covariance.
>
> In the scenario when the commutativity assumption is slightly violated, we provide an extension of our theoretical result.
>
> Proposition 2’’: Under the same condition in Theorem 2 without commutativity in Assumption 2, suppose $||G-G’||_2<\alpha$ where $G’$ is commutable with $H$. Then the expectation of excess risk satisfies
> $$ E\Delta(w^{(2)})\leq bias+variance, $$
> where in both bias and variance, the $G+H$ terms in Eq. (6) are replaced with $G+H+2\alpha I$.
>
> This will be enough for small misspecification. We will make sure to include a polished version of these in the revision.
>
> ---
>
> Q2. Would you consider adding experiments on more challenging benchmarks (e.g., CIFAR-100, ImageNet subsets) to better demonstrate the practical relevance of your theory?
>
> A2. We provide a preliminary 2-task Split CIFAR-100 experiment on a low-memory regularization-based practical CL method to demonstrate the practical relevance of your theory; due to computational constraints, even larger experiments are out of the scope of this paper.
>
> | Memory size  | k=2 | k=5 | k=10 | k=20 | k=50 |
> |---|---|---|---|---|---|
> | Avg. accuracy | 85.1% | 88.4% | 90.4% | 89.6% | 90.4% |
>
>
> Table 1: Average accuracy of Sketched EWC with k=2,5,10,20,50 number of sketches
>
> We show that in such a practical setting, the performance initially improves when the memory $k$grows. After $k$ grows to a point, the performance essentially saturates. This is consistent with our theory and other experimental demonstrations.
>
> We will make sure to include a refined version of this CIFAR experiment in the revision.
>
> ---
>
> Q3. Have you compared GRCL with strong baselines from other CL paradigms (e.g., experience replay, expansion methods)? Such comparisons would help position your method in the broader CL landscape.
>
> A3. Our theoretical analysis focuses on the memory-performance relationship of CL methods; comparison between performances of different CL paradigms is not the focus of this paper. Understanding the memory-performance relationship in other CL paradigms like replay-based methods is an interesting question, which we will comment on as a future direction.

---

### Official Review · Reviewer_YoRz · 2025-11-02

**Soundness:** 3
**Presentation:** 3
**Contribution:** 2
**Rating:** 4
**Confidence:** 4

**Summary:**

The paper studies continual learning with two linear-regression tasks under random design. It analyzes three algorithms: OCL,l2-RCL and GRCL and establishes sharp upper/lower bounds on the joint excess risk of three lags. It shows that GRCL algorithm that uses a curvature-aware quadratic penalty (a PSD matrix) derived from the first task to mitigate catastrophic forgetting.

**Strengths:**

1. The paper considers an interesting problem in theoretical continual learning community.
2. The paper is well-written and well-structured.
3. The results driven by the data covariance are novel and interesting.

**Weaknesses:**

1. Several related work about theoretical CL are not discussed, for example:

[C1] Banayeeanzade et al., Theoretical Insights into Overparameterized Models in Multi-Task and Replay-Based Continual Learning, TMLR 2025.

[C2] Zheng et al., Towards Understanding Memory Buffer Based Continual Learning, 2025.

[C3] Ding, et al. Understanding forgetting in continual learning with linear regression. 2024.

[C4] Wen, et al. Information-theoretic generalization bounds of replay-based continual learning. 2025.

[C5] Wan et al. Understanding the Forgetting of (Replay-based) Continual Learning via Feature Learning: Angle Matters. 2025.

2. Two tasks in linear setting is simply and not enough to discuss the paper's findings. If it is only two tasks, what is difference between transfer learning vs continual learning? Based on equation 2, 3, it seems to be hard to extend the paper's results to $M$ tasks setting.

3. How to understand the “tail” of the distributions in equation 5?

4. Can the author decompose their results to show the separate loss evaluations on each tasks? Since the main results are based on joint learning.

5. The results in section 4.2, why only give the example instead of general results. Since the paper investigates the impact of constrained memory while it still assumes that the data covariance ($\mu_i$) is related to the constrained memory setting ($k$). Can the authors explain more?

**Questions:**

Above

---

> ### Author Response · Authors · 2025-11-23
>
> Thank you for your comments. We address your concerns as follows.
>
> ---
>
> Q1. Several related work about theoretical CL are not discussed.
>
> A1. Thank you for raising these works. While these are good works, we believe there are substantial gaps between our work and their papers. We list the differences and comparisons as follows:
>
> - [Banayeeanzade et al.]: The data distribution considered in this paper is vastly different with ours: they consider an isotropic Gaussian distribution of input $x$, while we consider distributions with arbitrary covariance matrices. Also, we theoretically quantify the relation between memory and performance in regularization-based CL.
> - [Zheng et al.], [Wen, et al] and [Wan et al]: They consider the replay-based CL method, while we consider the regularization-based methods.
> - [Ding, et al.]. This paper considers SGD without any regularization, while we consider regularization-based CL and its memory-performance relationship.
>
> We will make sure to reflect these in the revision.
>
> ---
>
> Q2. Two tasks in linear setting is simply and not enough to discuss the paper's findings. If it is only two tasks, what is difference between transfer learning vs continual learning? Based on equation 2, 3, it seems to be hard to extend the paper's results to M tasks setting.
>
> A2. In the paper we provided the extension of our results to multi-task learning: please refer to Ln 453-465 in Section 5 as well as proofs in Appendix D.2. We would like to emphasize that transfer learning is different with continual learning in both two-task and multi-task setting. Transfer learning algorithms are measured by the performance of the target task (i.e., the last task). In comparison, continual learning is measured by the performance of all trained task, therefore forgetting must be considered.
>
> ---
>
> Q3. How to understand the “tail” of the distributions in equation 5?
>
> A3. Thanks for pointing this out. We clarify that the necessary and sufficient condition in Eq. (5) depends on the “head” and “tail” of the eigenspaces of the data covariances $G$ and $H$. The “tail” refers to the small eigenvalues of $G$ and $H$, and Eq. (5) states that it must not have a heavy tail (i.e. the 2-norm of the small eigenvalues is sufficiently small).
>
> We will clarify this in the revision.
>
> ---
>
> Q4. Can the author decompose their results to show the separate loss evaluations on each tasks? Since the main results are based on joint learning.
>
> A4. In the main paper, our result in Theorem 2 presents the joint population risk, which a common measure of CL performance. For details of expression risk for each task, please simply replace the $(G+H)$ terms with $G$ for task 1 (or $H$ for task 2, respectively) at the beginning of each line in Eq. (6). This is because the joint risk can be decomposed in risk of each task, as stated in Eqs. (18-20) in Ln 763-768, Appendix B.1.
>
> Note that we measure the joint population risk of continual learning, which is different with joint learning where tasks are learned all at once.
>
> ---
>
> Q5. The results in Section 4.2, why only give the example instead of general results. Since the paper investigates the impact of constrained memory while it still assumes that the data covariance $(\mu_i)$ is related to the constrained memory setting $(k)$. Can the authors explain more?
>
> A5. We have given the general analytical expression of risk in Theorem 2. The examples are corollaries of Theorem 2, which aims to provide clear theoretical messages, including the relationship between memory and performance. In particular, we did not assume that the data covariance $\mu_i$ is related to the constrained memory $k$ of the CL algorithm. Rather, with Examples 6 and 8, we show that the optimal choice of $k$ (smallest memory for the risk to be negligible) is closely related to the rank of dominant eigenspace of data covariance, and $\Sigma$ should approximate that data covariance.

---

### Author Response · Authors · 2025-12-02

It is unfortunate that the discussion period with reviewers was cut short due to the identity leak incident.
Below we provide the new AC with a summary of the key points and promised revisions in the review-rebuttal period.

---

We would like to thank all the reviewers for their thoughtful suggestions on our submission, and appreciate that the reviewers have multiple positive impressions of our work, including:

- *Interesting/Fresh theoretical CL problem* (YoRz, GXcu): Investigating the trade-off between memory size and statistical performance of CL;
- *Solid/Concrete theoretical analysis* (RwWP, GXcu): Providing the upper and lower bounds for the joint excess risk of GRCL;
- *Well-chosen examples* (GXcu): Showing why naive OCL or an isotropic ridge can fail, why a structured $\ell_2$ penalty helps, and how the "top-$k$" regularization illustrates the trade-off.
- *Clear experiments and extensions to neural nets* (RwWP): Providing extensions to Gaussian design, multi-task setting and NTK-regime neural networks.
- *Clear logic, well-written and well-structured* (YoRz, RwWP, GXcu).

We summarize below the key revisions we will incorporate into the manuscript.

- *Theoretical extension on assumptions*: We will give formal proofs that establish the extensions to the **slightly non-commutative** (RwWP, GXcu) and **non-shared-optimal** (GXcu) scenarios.

- *Practical extension on CIFAR*: We will incorporate **CIFAR experiment** that demonstrates the memory-performance trade-off in a low-memory regularization-based CL algorithm. (RwWP)

- *Related works*: We will include the discussion of the works raised by the reviewer (YoRz).

- *Comparison table to fixed design*: We will incorporate the table that characterizes the difference between the random design results in this work and the fixed design results in previous work. (GXcu)

We have also made multiple clarifications on the reviews. We believe we have answered all raised questions and addressed all perceived weaknesses.

---

Once again, we sincerely thank all reviewers for their detailed feedback and constructive suggestions.
All corresponding clarifications, analyses, and additional results will be integrated into the revised submission.

---

### Meta-Review · Area_Chair_AfyW · 2026-01-11

**Summary:**

This paper studies the trade-off between memory complexity and statistical efficiency in the context of continual learning with two linear regression tasks. It proposes a Generalized $\ell_2$-regularized Continual Learning (GRCL) algorithm and establishes upper and lower bounds on the joint excess risk. The theoretical results are verified through numerical experiments and extensions on NTK-regime neural networks.

The reviews are mixed. Two reviewers consider this work to have solid theoretical analysis on matching bounds and choosing good examples to verify the effectiveness in mitigating catastrophic forgetting. In contrast, one reviewer raises concerns regarding the task setting (e.g., transfer/multitask vs. continual). The rebuttal is detailed and technically solid, successfully addressing the reviewers' concerns on theoretical assumptions and experimental results.

**Reviewer Concerns:**

Reviewer YoRz mainly raised concerns regarding the task setting and asked for further explanations on experimental results.

Reviewer RwWP mainly raised concerns about the practical implications of the proposed assumption and evaluations on more challenging datasets. The authors provided responses with an extension of theoretical results when the commutativity assumption is slightly violated, and provided additional experiments on CIFAR-100 to demonstrate the practical effectiveness.

Reviewer GXcu mainly raised concerns on the strength of assumptions, model selection, and comparison to prior studies. The authors rebutted by providing extensions of theoretical results when the optimal parameters differ and when the commutativity assumption is slightly violated. The authors also presented explanations on the model selection criterion.

In my view, the main concern from Reviewer YoRz is regarding the extension to multiple tasks and the difference between transfer learning and continual learning. The authors' responses are convincing to me, though I did not check the proof.

**Reviewer Scores:**

Reviewer YoRz: Rated the paper marginally below the acceptance threshold but did not participate in the rebuttal. The rebuttal is convincing to me.

Reviewer RwWP: Rated the paper marginally above the acceptance threshold and did not participate in the rebuttal. Although no explicit post-rebuttal update was provided, the rebuttal adequately addressed the raised concerns, and the score is likely to be stable.

Reviewer GXcu: Rated the paper marginally above the acceptance threshold and further discussed prior theory and the formal definition of memory complexity with the authors in the post-rebuttal phase. Based on the comments, the score is unlikely to change.

Overall, although the empirical analysis is relatively weak, I think this work makes a good theoretical contribution to continual learning, and therefore recommend acceptance.

---

### Decision · Program_Chairs · 2026-01-26

Accept (Poster)